# Architecture of the ESCPE-1 membrane coat

Carlos Lopez-Robles[1,7,10], Stefano Scaramuzza[2,10], Elsa N. Astorga-Simon[1,10], Morié Ishida[3,10], Chad D. Williamson[3], Soledad Baños-Mateos[1,7], David Gil-Carton [1,4,8], Miguel Romero-Durana [5,6], Ander Vidaurrazaga[1], Juan Fernandez-Recio [5,6], Adriana L. Rojas [1], Juan S. Bonifacino [3] ✉, Daniel Castaño-Díez [2,9] ✉ & Aitor Hierro [1,4] ✉

Recycling of membrane proteins enables the reuse of receptors, ion channels and transporters. A key component of the recycling machinery is the endosomal sorting complex for promoting exit 1 (ESCPE-1), which rescues transmembrane proteins from the endolysosomal pathway for transport to the *trans*-Golgi network and the plasma membrane. This rescue entails the formation of recycling tubules through ESCPE-1 recruitment, cargo capture, coat assembly and membrane sculpting by mechanisms that remain largely unknown. Herein, we show that ESCPE-1 has a single-layer coat organization and suggest how synergistic interactions between ESCPE-1 protomers, phosphoinositides and cargo molecules result in a global arrangement of amphipathic helices to drive tubule formation. Our results thus define a key process of tubule-based endosomal sorting.

Selective cargo recycling depends on coat protein complexes that guide the incorporation of specific cargo into vesicles and membrane tubules. Generally, short peptide motifs present in the cytoplasmic tail of transmembrane proteins interact with components of coat complexes, leading to cargo clustering and membrane deformation. Although these mechanisms are well characterized for COPI, COPII and clathrin-adaptor-type coats that associate with spherical membrane vesicles[1], they are poorly understood for other protein coats that associate with membrane tubules.

Prominent sorting and recycling machineries at endosomes include members of the sorting nexin (SNX) family of proteins, the retromer complex and the recently identified retriever complex[2,3]. The SNX protein family is defined by the presence of a phox-homology (PX) domain that typically binds phosphatidylinositol phospholipids (PtdInsPs), but may also participate in protein–protein interactions[4]. Twelve out of the 33 annotated mammalian SNXs contain an additional carboxy-terminal Bin/Amphiphysin/Rvs (BAR) domain, and are therefore denoted as the PX-BAR or SNX-BAR subfamily. BAR domains exhibit restricted patterns of homodimerization and heterodimerization and play important roles in membrane remodeling processes such as endosomal sorting, endocytosis and autophagy.

Heterodimeric combinations of SNX1 or SNX2 with SNX5, SNX6 or SNX32 have been designated as the endosomal SNX-BAR sorting complex for promoting exit 1 (ESCPE-1)[5]. Thus far, ESCPE-1 has been associated with the sorting of over 60 cargos with broad functions, including the cation-independent mannose-6-phosphate receptor (CI-MPR), the roundabout homolog 1 receptor (ROBO1), the insulin-like growth factor 1 receptor (IGF1R), the tumor necrosis factor-related apoptosis-inducing ligand receptor (TRAILR1)[5–8], and the multifunctional coreceptor neuropilin-1 involved in cardiovascular and neuronal development, as well as in severe acute respiratory syndrome coronavirus 2 (SARS-CoV-2) infection[9], and in *Chlamydia trachomatis* pathogenesis[10–12].

Despite being a central pillar to multiple trafficking pathways, the precise organization and mechanistic function of ESCPE-1 has been enigmatic. Here, we describe the molecular basis for BAR domain

[1]CIC bioGUNE, Derio, Spain. [2]BioEM Lab, Biozentrum, University of Basel, Basel, Switzerland. [3]Neurosciences and Cellular and Structural Biology Division, Eunice Kennedy Shriver National Institute of Child Health and Human Development, National Institutes of Health, Bethesda, MD, USA. [4]Ikerbasque, Basque Foundation for Science, Bilbao, Spain. [5]Barcelona Supercomputing Center (BSC), Barcelona, Spain. [6]Instituto de Ciencias de la Vid y del Vino (ICVV), CSIC–Universidad de La Rioja–Gobierno de La Rioja, Logroño, Spain. [7]Present address: VIVEbiotech, Donostia, Spain. [8]Present address: BREM Basque Resource for Electron Microscopy, Leioa, Spain. [9]Present address: Instituto Biofisika (UPV/EHU, CSIC), University of the Basque Country, Leioa, Spain. [10]These authors contributed equally: Carlos Lopez-Robles, Stefano Scaramuzza, Elsa N. Astorga-Simon, Morié Ishida. ✉e-mail: juan.bonifacino@nih.gov; daniel.castano@csic.es; ahierro@cicbiogune.es

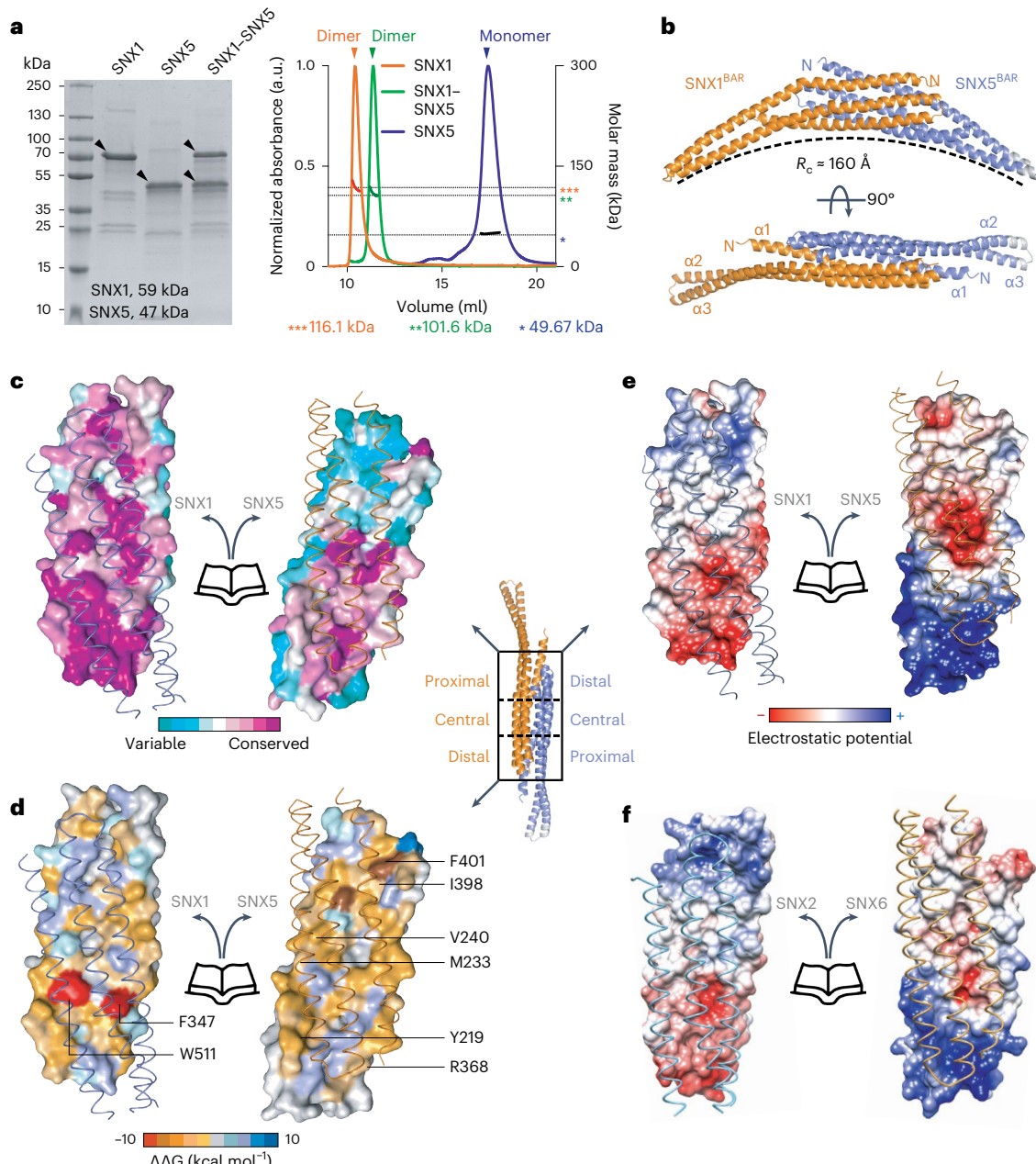

**Fig. 1 | Crystal structure of the SNX1^BAR–SNX5^BAR heterodimer and interface analysis. a**, SDS–PAGE and SEC–MALS analysis of full-length SNX1, SNX5 and SNX1–SNX5 showing the molecular weight difference between species. **b**, Structure of the human SNX1^BAR–SNX5^BAR heterodimer in two orthogonal views. $R_c$ stands for radius of curvature. **c–e**, Close-up views of the SNX1^BAR–SNX5^BAR heterodimer interface illustrating conserved amino-acid residues (**c**), energetic landscape and binding hot-spot prediction (**d**) and electrostatic surface potential from $-5\,kT\,e^{-1}$ (red) to $5\,kT\,e^{-1}$ (blue) (**e**). **f**, Electrostatic surface potential of the SNX2^BAR–SNX6^BAR heterodimerization interface generated by homology modeling. Results in **a** are representative of at least three independent experiments.

heterodimerization and the process that guides cargo capture, coat assembly and tubular protrusion within a single-layer coat.

## Results

### Structure of the SNX1–SNX5 BAR heterodimer

Analysis of ESCPE-1 (SNX1^FL–SNX5^FL, where FL stands for full length; a glossary of protein names used through the paper is shown in Extended Data Table 1) by size-exclusion chromatography coupled with multi-angle light scattering (SEC–MALS) revealed a molecular mass consistent with a single heterodimer in solution (Fig. 1a). Similarly, isolated SNX1^FL formed stable homodimers, whereas SNX5^FL was monomeric;

however, only dimeric SNX1^FL–SNX5^FL and SNX1^FL–SNX1^FL were able to induce membrane tubulation on synthetic liposomes[13] (Extended Data Fig. 1a). To understand the molecular basis for heterodimerization between ESCPE-1 protomers and its membrane tubulation activity, we determined the crystal structure of SNX1^BAR–SNX5^BAR (where BAR stands for the BAR domains of both proteins) at 2.5 Å resolution (Fig. 1b, Table 1 and Extended Data Fig. 1b). The structure forms an antiparallel dimer of three-helix bundles with a curvature that would fit a circle with a diameter of ~32 nm (Extended Data Fig. 1c). Electron density at the tip of SNX5^BAR is very weak, indicating flexibility of this region (Extended Data Fig. 1d). Unlike other SNX-BAR homodimers,

**Table 1 | Data collection, phasing and refinement statistics**

| | Native SNX1–SNX5 | | Crystal 1 Pt derivative | |
|---|---|---|---|---|
| **Data collection** | | | | |
| Space group | P2₁ | | P2 | |
| Cell dimensions | | | | |
| $a, b, c$ (Å) | 49,78, 188.47, 51.60 | | 52.69, 51.47, 192.32 | |
| $α, β, γ$ (°) | 90.0, 89.94, 90.0 | | 90.0, 90.0, 90.0 | |
| | | Peak | Inflection | Remote |
| Wavelength | 0.9793 | 1.07158 | 1.07293 | 1.04473 |
| Resolution (Å) | 50–2.5 (2.65–2.50)[a] | 50–2.8 (2.95–2.80)[a] | 50–2.8 (2.95–2.8)[a] | 50–2.6 (2.74–2.6)[a] |
| $R_{meas}$ (%) | 15.1 (99.8) | 9.5 (118.2) | | |
| $I/σI$ | 7.15 (1.69) | 10.3 (1.7) | 6.7 (1.1) | 8.1 (1.1) |
| Completeness (%) | 98.5 (94.5) | 96.2 (96.1) | 96.0 (96.3) | 95.4 (95.3) |
| Redundancy | 5.17 (4.58) | 6.3 (6.2) | 6.0 (5.8) | 6.2 (6.1) |
| $CC_{1/2}$ (%) | 99.3 (63.2) | 99.9 (70.7) | 99.7 (39.0) | 99.9 (12.3) |
| **Refinement** | | | | |
| Resolution (Å) | 49.7–2.5 (2.58–2.50)[a] | | | |
| No. of unique reflexions | 32,404 (4,992) | 24,815 (3,609) | | |
| $R_{factor}/R_{free}$ | 23.7/27.6 | 25.2/30.5 | | |
| No. of atoms | | | | |
| Protein | 6,742 | 6,648 | | |
| Ligand/ion | 7/0 | 0/8 | | |
| Water | 21 | 4 | | |
| B factors | | | | |
| Protein | 59.03 | 72,19 | | |
| Ligand/ion | 71.01/0 | 0/171 | | |
| Water | 48.02 | 41.25 | | |
| R.m.s. deviations | | | | |
| Bond length (Å) | 0.001 | 0.002 | | |
| Bond angle (°) | 0.293 | 0.398 | | |

$I/σI$, mean of intensity/$σI$ of unique reflections (after merging symmetry-related observations); $CC_{1/2}$, percentage of correlation between intensities from random half-datasets. [a]Values in parentheses are for the highest-resolution shell.

the SNX1^BAR–SNX5^BAR heterodimer exhibits asymmetry on its surface properties (Extended Data Fig. 1e), general shape (Extended Data Fig. 1f), and the associated PX domains located at either side of the BAR heterodimer (Extended Data Fig. 1g).

**Principles of SNX-BAR dimerization**

Next, we carried out a comprehensive characterization of the SNX1^BAR–SNX5^BAR interface residues (Fig. 1c–e). The interface buries a large area of 5,114 Å², of which 78% corresponds to polar surface. Residue conservation, energy profile and electrostatic potential exhibit clear patterns that define the associative behavior (Fig. 1c–e). We estimated the energetic contribution of individual interface residues[14] (Fig. 1d and Extended Data Fig. 2). From this analysis, we found that two clusters formed by F347 and W511 in SNX1, and F273 and F401 in SNX5, have the highest binding energy. Yet, only F347 and W511 in SNX1 are conserved, indicating that these residues likely form a binding hot spot. Additionally, we identified two residues in SNX5 (R388 and Q395) that establish favorable contacts at the SNX1^BAR–SNX5^BAR heterodimer interface, but their electrostatic and solvation energy contribution is unfavorable in a hypothetical SNX5^BAR homodimer (Extended Data Fig. 3). SNX1^BAR and SNX5^BAR exhibit complementary polarity at their distal and proximal interfaces (Fig. 1e). Consistent with previous predictions[13], the central region of SNX5^BAR has a negative patch, whereas the equivalent region

in SNX1^BAR is neutral. Specifically, the conserved E280 in SNX5 is occupied by H381 at the equivalent position in SNX1 (Extended Data Fig. 3). Of note, other retromer-related SNX-BARs such as SNX2 and SNX6 display a comparable electrostatic pattern in their BAR domain interfaces (Fig. 1f, compare to Fig. 1e).

To validate the interface analysis between BAR domains, we generated various mutants influencing SNX1 homodimerization and heterodimerization. From the per-residue energetic analysis, we found that residues F347 and W511 in SNX1 displayed a similar energetic contribution for homodimerization and heterodimerization (Extended Data Fig. 3). Yet, in SEC–MALS and pull-down experiments, a double mutation of F347A and W511A (SNX1†) resulted in disruption of SNX1 homodimers but not heterodimers (Extended Data Fig. 4a,b), indicating that SNX1–SNX5 had more favorable contacts and stability gain. We also found that when SNX1 and SNX5 were incubated in equimolar proportion, the amount of SNX5 retained by SNX1 in pull-down assays was almost equimolar as well, thus demonstrating that SNX1 had a clear preference for heterodimerization (Extended Data Fig. 4b). On the other hand, given that the SNX5 interface does not exhibit clear conserved hot spots, we initially introduced four point mutations, Y219A, M233A, V240A and R368A (SNX5‡), within the central and proximal regions in a compromise between conservation scores and energy contribution (Fig. 1c,d and Extended Data Fig. 2a). We did not contemplate

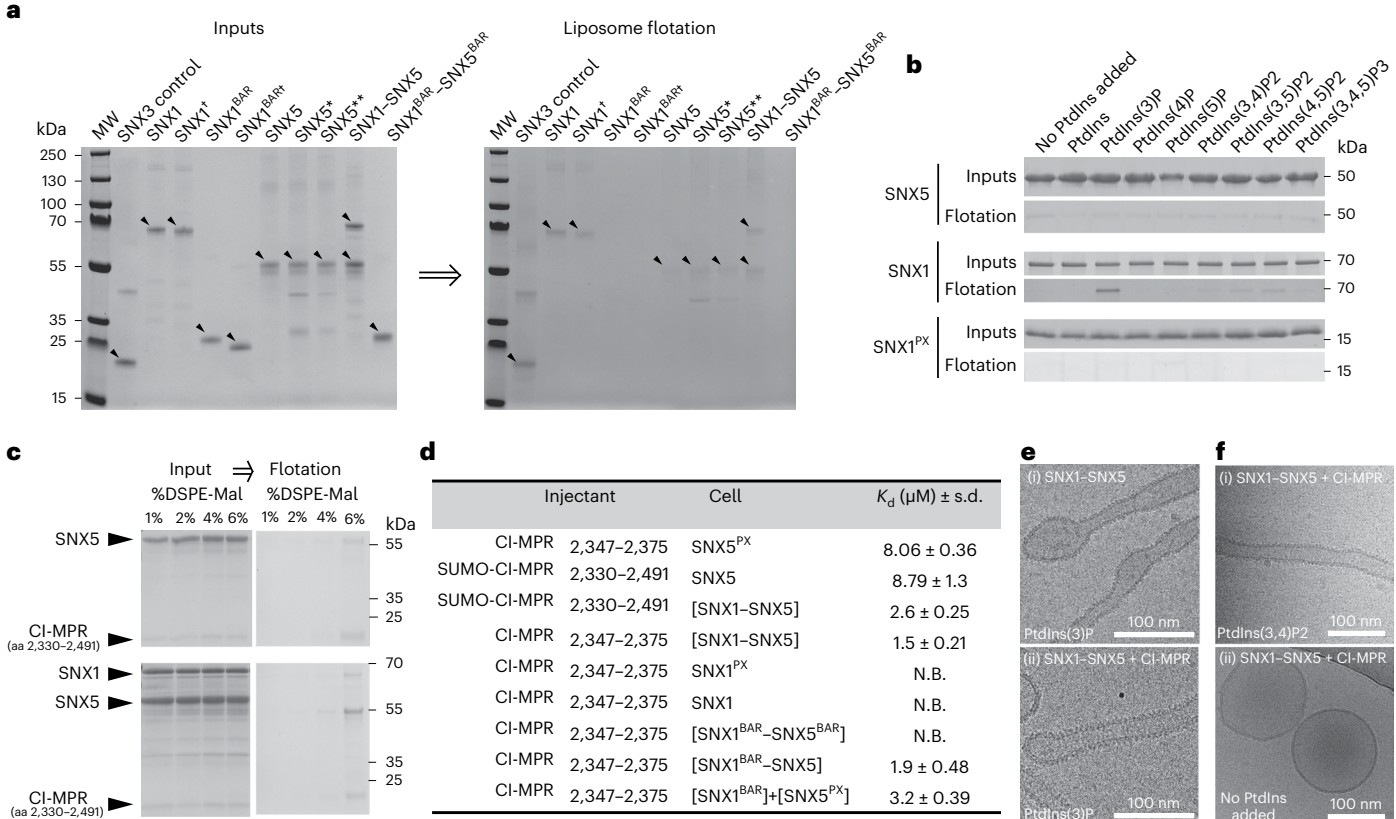

**Fig. 2 | Membrane recruitment and coat organization is influenced by dimerization, phosphoinositides and cooperative interactions with cargo. a**, Effects of SNX1 and SNX5 interface mutations on the association with liposomes (DOPC:DOPE:DOPS:PtdIns(3)P:Liss Rhod PE in a 45:28:20:5:2 molar ratio) by flotation assay. MW, molecular weight marker. Note that association with the membrane was enhanced by dimerization of full-length SNXs but not by heterodimers of BAR domains alone. All SDS–PAGE samples that originated from flotation assays were normalized relative to their Liss Rhod PE content. **b**, Liposome flotation analyses to characterize the binding of SNX5, SNX1 and SNX1$^{PX}$ to specific phosphoinositides. Note that only full-length SNX1 interacts specifically with PtdIns(3)P, and to a minor extent with PtdIns(4,5)P2, PtdIns(3,5) P2 and PtdIns(3,4)P2. **c**, CI-MPR promotes membrane recruitment of SNX5, and this effect is enhanced in the presence of SNX1. Flotation assay of liposomes functionalized with the cytosolic tail of CI-MPR. CI-MPR was conjugated with

increasing concentrations of DSPE-Mal on the surface of liposomes containing no phosphoinositides to exclude their specific interaction with SNX1. aa, amino acids. **d**, SNX1$^{BAR}$ domain enhances the interaction between the PX domain of SNX5 and CI-MPR. Summary of $K_d$ values between CI-MPR and SNX1–SNX5 or various subdomains from the heterodimer. Values are the mean ± s.d. from at least two independent experiments. N.B. no binding. **e**, Representative cryo-transmission electron microscopy (cryo-TEM) images of liposomes (DOPC:DOPE:DOPS:PtdIns(3)P in a 45:30:20:5 molar ratio) incubated with SNX1–SNX5 in the absence (i) or presence (ii) of the cytoplasmic tail of CI-MPR. **f**, Representative cryo-TEM images of liposomes incubated with SNX1–SNX5 and the cytoplasmic tail of CI-MPR in the presence of PtdIns(3,4)P2 (i) or in the absence of PtdIns (ii). Data are representative of three (**a**–**c**) or two (**e**, **f**) independent experiments.

the mutation of S226 in SNX5, as it was not considered an energetically 'hot' residue for the interaction (Fig. 2 and Extended Data Fig. 3) despite the fact that its phosphorylation inhibits heterodimerization with SNX1 and SNX2 (ref. 15). The SNX5$^{‡}$ protein, bearing all four mutations, heterodimerized with SNX1 or SNX1$^{†}$ (Extended Data Fig. 4b). However, the inclusion of two additional mutations, I398A and F401A, in SNX5 (SNX5$^{§}$) abrogated the interaction with SNX1$^{†}$ but not with wild-type (WT) SNX1 (Extended Data Fig. 4b). This observation confirmed that SNX5 uses an evenly distributed energy landscape across a large binding area for interaction with SNX1.

Next, to address the contribution of E280 to preventing SNX5 homodimerization, we introduced the E280H mutation (SNX5*), which mimics H281 in SNX1 (Extended Data Fig. 3), and examined whether this mutation promoted association with SNX5$^{WT}$. Indeed, using sequential affinity capture with separate tags, we confirmed the SNX5–SNX5* complex formation (Extended Data Fig. 4c). Further inclusion of the two residues considered a hot spot in SNX1 (F347 and W511) at the equivalent positions in SNX5, T247F and L394W (SNX5**) (Extended Data Fig. 3), increased the yield of the purified SNX5–SNX5** complex by ~40%, implying a considerable stabilization of the interface

(Extended Data Fig. 4c). These results confirmed that electrostatic repulsive forces within the central region of the SNX5$^{BAR}$ domain preclude homodimerization, whereas residues at the binding hot spot on SNX1 stabilize the interaction.

Lastly, in liposome tubulation assays, mutants that impaired SNX1 homodimerization or SNX1–SNX5 heterodimerization were, as expected, unable to tubulate membranes (Extended Data Fig. 4d). Yet, the SNX5–SNX5**-induced complex did not acquire the ability to tubulate membranes, suggesting that dimerization between BAR domains contributes only partially to the tubulation process.

## Lipids and cargo mediate ESCPE-1 coat assembly

In testing the ability of the above SNX1$^{FL}$ and SNX5$^{FL}$ variants to associate with membranes using liposome flotation assays, we noticed more binding for dimeric SNX1 than monomeric SNX1, suggesting that dimers promote more stable membrane association. SNX5 and the SNX5–SNX5**-induced complex exhibited a similar pattern but with a much weaker association to the membrane (Fig. 2a). Interestingly, the SNX1$^{BAR†}$ monomer, the SNX1$^{BAR}$ homodimer and the SNX1$^{BAR}$–SNX5$^{BAR}$ heterodimer did not interact with membranes, indicating that the sole

BAR domains, even in dimeric conformation, were not sufficient for membrane association.

Previous phosphoinositide interaction studies of SNX1 showed weak-to-moderate or even no interaction with PtdIns(3)P, PtdIns(3,4)P2 and PtdIns(3,5)P2 (refs. 8,16,17), whereas SNX5 was unable to interact with any PtdIns[17,18]. Consistent with these results, we observed negligible levels of SNX5$^{FL}$ association to phosphoinositide-containing liposomes in flotation assays (Fig. 2b). In contrast, SNX1$^{FL}$ showed strong association with PtdIns(3)P and minor association with PtdIns(4,5)P2, PtdIns(3,5)P2 and PtdIns(3,4)P2 (Fig. 2b). Yet, the SNX1$^{PX}$ domain did not exhibit preferential association with any PtdIns under identical experimental conditions (Fig. 2b), suggesting that the sole PX domain of SNX1 has a very weak binding for specific phosphoinositides. Together, these results indicate that SNX5 has negligible, nonspecific association with phospholipid bilayers, whereas SNX1 requires the simultaneous presence of the PX and the BAR domains for its association with PtdIns(3)P and, to a lesser extent, with PtdIns(4,5)P2, PtdIns(3,5)P2 and PtdIns(3,4)P2.

Next, we addressed the effect of the cytoplasmic domain of the CI-MPR (CI-MPR$_{2330-2491}$), which interacts with the PX domain of SNX5 (refs. 5,8), for SNX1–SNX5 recruitment to membranes. To mimic the juxtamembrane position of the cytoplasmic tail of CI-MPR, we anchored the amino terminus to the surface of liposomes by chemical cross-linking with maleimide-functionalized lipids. As expected, the presence of the CI-MPR tail on liposomes lacking any PtdInsP not only triggered the recruitment of SNX5 alone, but also augmented the recruitment of the SNX1–SNX5 heterodimer, suggesting a cooperative action of SNX1 (Fig. 2c). This observation is in line with other cooperative effects mediated by CI-MPR and PtdIns(3)P for the recruitment of SNX1–SNX6 to model membranes[8], or the positive cooperative effect of the DMT1-II cargo in the recruitment of SNX3-retromer[19]. In a similar way, the yeast VPS10 cargo enhanced local clustering of VPS5-VPS17-retromer in membrane microdomains[20].

Next, we aimed to characterize the binding between CI-MPR and ESCPE-1 using isothermal titration calorimetry (ITC). First, we confirmed the binding of the shorter CI-MPR$_{2347-2375}$ tail segment with the SNX5$^{PX}$ domain, which showed a dissociation constant ($K_d$) of ~8 μM (Fig. 2d and Extended Data Fig. 5), consistent with other reported values[5,8]. Similarly, the interaction between the entire CI-MPR$_{2330-2491}$ tail domain and the full-length SNX5 exhibited a $K_d$ of ~8.8 μM (Fig. 2d and Extended Data Fig. 5b), indicating that no additional regions outside the PX domain of SNX5 were involved in the interaction with the cytosolic tail of CI-MPR. Intriguingly, the binding of the tail domain, CI-MPR$_{2330-2491}$, or a shorter fragment containing the bipartite binding motif (CI-MPR$_{2347-2375}$), to the SNX1$^{FL}$–SNX5$^{FL}$ heterodimer showed $K_d$ values of 2.6 μM and 1.5 μM, respectively, which were slightly but significantly higher than those observed with the isolated PX domain (Fig. 2d and Extended Data Fig. 5c,d). These results indicated that the presence of SNX1$^{FL}$ in complex with SNX5$^{FL}$ enhanced the binding between the PX domain of SNX5 and CI-MPR$_{2347-2375}$. To see if the CI-MPR established additional interactions within the ESCPE-1 complex outside the PX domain of SNX5, we evaluated the interaction between CI-MPR$_{2347-2375}$ and SNX1$^{PX}$, SNX1$^{FL}$ or the SNX1$^{BAR}$–SNX5$^{BAR}$ heterodimer and found no binding (Fig. 2d and Extended Data Fig. 5e–g). In contrast, we observed a $K_d$ of ~1.9 μM for the interaction with SNX1$^{BAR}$–SNX5$^{FL}$, which is comparable to that observed for the full-length heterodimer (Fig. 2d and Extended Data Fig. 5h). Of note, SNX1$^{BAR}$ could only contribute to this interaction with its BAR domain, which is far apart from the PX domain of SNX5 and unable to interact with it unless the cooperative effect in cargo binding was promoted by a separate heterodimer. To test this, we titrated CI-MPR$_{2347-2375}$ into a mixture of the two isolated domains, SNX1$^{BAR}$ and SNX5$^{PX}$, and found a $K_d$ of ~3.2 μM, which indicated that the binding was moderately stronger in the presence of SNX1$^{BAR}$ (Fig. 2d and Extended Data Fig. 5i). These results confirmed that the BAR domain of SNX1 allosterically enhanced CI-MPR recognition by the

PX domain of SNX5. Furthermore, when we included the cytoplasmic domain of CI-MPR with SNX1$^{FL}$–SNX5$^{FL}$ in liposome tubulation assays, we observed the formation of more uniformly coated tubes (Fig. 2e), suggesting that the mild allosteric behavior might be reinforced by the cumulative effects of multivalent interactions during coat oligomerization. Similarly, when CI-MPR and SNX1–SNX5 were incubated with liposomes containing PtdIns(3,4)P2, we observed homogeneous tubulation activity, albeit to a lesser extent than with PtdIns(3)P, which is consistent with the lower affinity for PtdIns(3,4)P2 (Fig. 2f). On the contrary, in the absence of PtdIns, there was no tubulation (Fig. 2f).

## Structure of the membrane-assembled SNX1–SNX5 coat

To understand how coat assembly, cargo sorting and membrane deformation are coordinated, we performed cryo-electron tomography (cryo-ET) on the tubulation reaction mediated by SNX1–SNX5 in the presence of CI-MPR$_{2330-2491}$, and solved the three-dimensional (3D) structure by subtomogram averaging (Fig. 3a–d and Table 2). The tube lengths varied from ~100 nm to ~1,600 nm with an average of ~460 nm (Extended Data Fig. 6a). Initial subtomogram averaging of short tube segments revealed right-handed helices around the membrane with a different number of helix starts (rows around the axis) depending on the tubes. Whereas tubes with one start represented only 5.6% of occurrences, tubes with two, three and four helical starts were equally common with occurrences of 31.7%, 27.2% and 36.1%, respectively (Extended Data Fig. 6b).

Following the methodology described in Subtomogram averaging, we characterized the helical behavior of the coating of each filament to perform a particle extraction guided by the lattice geometry determined in each case, leading to two different averages: one average that included the structure of three complete consecutive particles (three-particle average, resolution ~12 Å) (Fig. 3b and Extended Data Fig. 6c), and a finer average that included the structure of individual particles (one-particle average, resolution ~10 Å) (Fig. 3c and Extended Data Fig. 6d). At this resolution, the PX and BAR domains were clearly resolved in the cryo-electron microscopy (cryo-EM) maps revealing a network of interactions that involve the edge of BAR domains (BAR$^{tip}$) and the PX domains, which we denoted as BAR$^{tip}$-to-BAR$^{tip}$ and BAR$^{tip}$-to-PX interactions (Fig. 3c,d). The SNX1–SNX5 heterodimer forms helical rows that are held together mainly through two intertwined BAR$^{tip}$-to-PX contacts between the edge of one BAR domain and the PX domain of the next heterodimer. Considering that cargo binding is enhanced through SNX1$^{BAR}$ and SNX5$^{PX}$ contacts, only successive SNX1–SNX5 heterodimers would enable tip-to-PX contacts between SNX1$^{BAR}$ and SNX5$^{PX}$. Thus, although we could not exclude orientations with contacts between identical protomers, we considered the head-to-tail interlinkage, which is the most plausible scaffold in the presence of cargo. According to this, we created a composite model with the structures of the SNX1$^{BAR}$–SNX5$^{BAR}$ domains (present work), the SNX1$^{PX}$ domain[21], the SNX5$^{PX}$ domain bound to CI-MPR[5] and the predicted AlphaFold[22] structures corresponding to the linker regions between the PX and BAR domains, and then fitted this model to the density map. The lack of high resolution precluded the distinction between the two possible helical-screw directions of heterodimers (Fig. 3e). Using the coordinates of the one-particle average, the geometrical lattice parameters were derived by overlapping the relative positions of all neighboring particles of each particle, generating a so-called neighborhood plot (Fig. 3f). The neighborhood plot showed density peaks corresponding to preferred particle positions. In this sense, particles upstream and downstream of the helical filament that hold the BAR$^{tip}$-to-PX contacts exhibited a more homogeneous distribution (rounded density peaks) compared with lateral neighbors with BAR$^{tip}$-to-BAR$^{tip}$ contacts (Fig. 3f). It is worth mentioning that the lattice parameters ($a$, $b$, $\alpha$ and radius) show only minimal variation between tubes with a different number of helical starts (<1 nm and <1°; Extended Data Fig. 6e–i). Only the relative angle between two consecutive particles varies slightly to adapt to the changing curvature

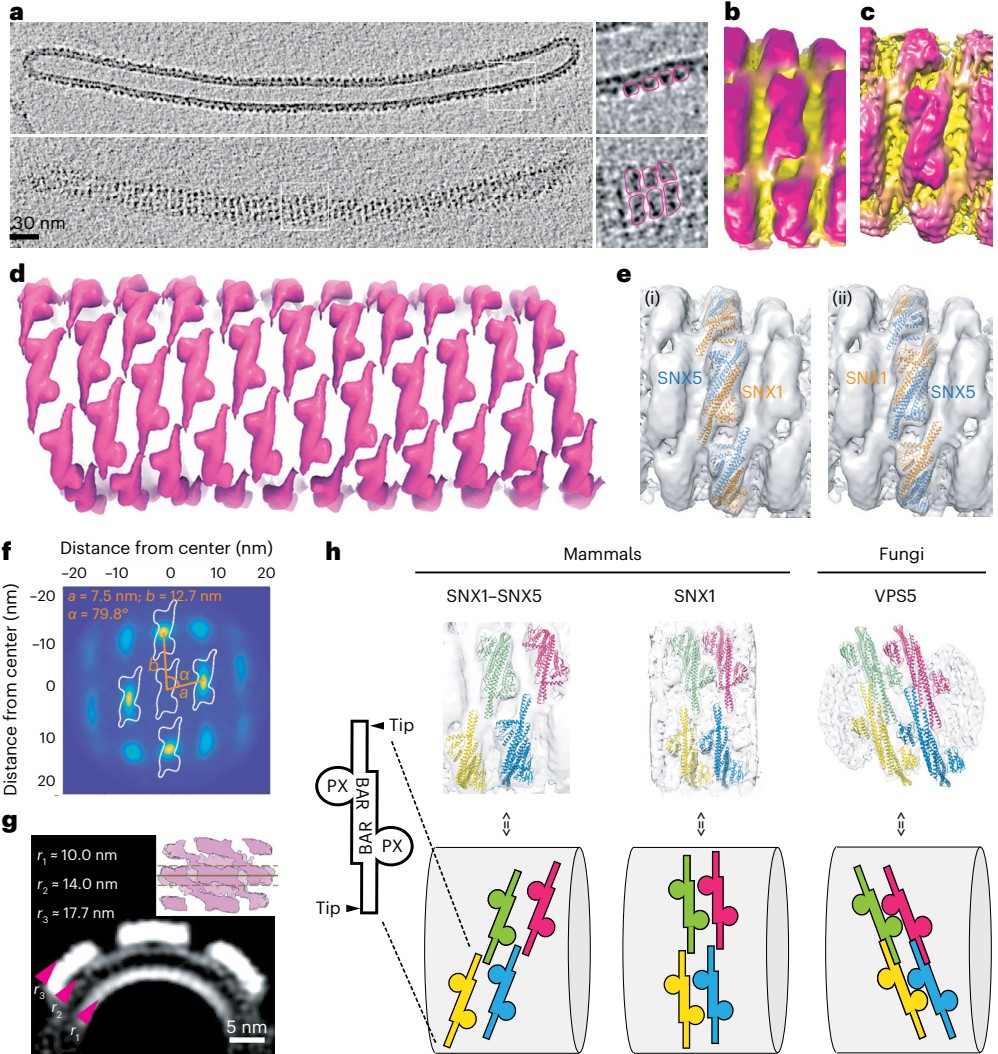

**Fig. 3 | Cryo-ET structure of the membrane-associated ESCPE-1 complex.**
**a**, Sections from a tomogram of a representative tube showing a cut of the tube and its surface. Top and side views are highlighted. **b**, Final average of subtomograms that include three particles. **c**, Average of subtomograms from individual particles. **d**, One-particle averages placed at corresponding coordinates on a representative tube. **e**, Two possible orientations of the SNX1–SNX5 heterodimer are shown relative to the helical filament. **f**, Projection of neighborhood plot using all particles. The distances and angles were computed in 3D (using *k*-means clustering). **g**, Cross-section (averaged over 15 pixels)

through the center of the three-heterodimer map from all particles. Arrowheads denote the radius of curvature for each leaflet of the membrane and the protein coat. The inset shows where the cross-section average was made. **h**, Comparison of membrane lattice scaffolds of the mammalian SNX1–SNX5 heterodimer (current study), the mammalian SNX1 dimer[24] and the fungal VPS5 dimer solved in the context of the retromer complex[23]. Upper row shows four dimers in different colors at equivalent positions on each lattice. Lower row shows a cartoon representation for each lattice.

of the helix caused by different lead angles (Extended Data Fig. 6j). A cut through the three-particle average further shows the bending of the membrane imposed by the SNX1–SNX5 heterodimers (Fig. 3g). The average peripheral membrane diameter was 28 ± 2 nm, which was decorated with a ~3.7-nm-thick protein coat. Although the diameter was slightly smaller than tubules induced by VPS5-retromer (~31 ± 6 nm)[23] or SNX1 homodimers (32–36 nm)[24], all of these protein coats reshaped lipid membranes with comparable curvature values. In contrast, the lattice contacts of these coats differ substantially. In particular, VPS5 and SNX1 homodimers exhibit PX-to-BAR lateral interactions between adjacent rows, whereas SNX1–SNX5 heterodimers are characterized by intertwined BAR^tip-to-PX contacts along the helical row (Fig. 3h and Extended Data Fig. 7a).

The mammalian SNX1/2 and SNX5/6 have been considered potential equivalents of yeast VPS5 and VPS17 (refs. 25,26). Yet, whereas the yeast retromer forms a stable pentameric complex with VPS5–VPS17,

the association of the equivalent mammalian SNXs with retromer has been considered to be labile[27,28] and even to have distinct functional roles[6,7]. Interestingly, the geometric distribution of retromer contacts over the VPS5 lattice in *Chaetomium thermophilum* is not conserved in the mammalian SNX1–SNX5 lattice, suggesting that retromer would not be able to dock in the same configuration (Extended Data Fig. 7b). To test whether retromer could be recruited by SNX1–SNX5, we co-incubated these proteins in the presence of CI-MPR and performed a liposome flotation assay. The results showed that SNX1–SNX5 was unable to recruit retromer (Extended Data Fig. 7c). In contrast, co-incubation of retromer with SNX3 and the DMT1-II_{550–568} sorting motif resulted in retromer recruitment to the membrane (Extended Data Fig. 7c), consistent with their direct interaction[29].

The lipid-binding regions in SNX1–SNX5 include the PX domains, the tips of each BAR domain and an amphipathic helix that connects the PX and BAR domains[13,24] (Fig. 4a–c and Supplementary Video 1).

**Table 2 | Cryo-ET data collection, refinement and validation statistics**

| | SNX1–SNX5 and CI-MPR tail (EMD-15413); (PDB 8AFZ) |
|---|---|
| **Data collection and processing** | |
| Magnification | ×53,000 |
| Voltage (kV) | 300 |
| Electron exposure (e⁻ Å⁻²) | ~120 uniformly distributed over tilt series |
| Defocus range (μm) | −2 to −5 (0.5 steps) |
| Pixel size (Å) | 2.73 |
| Symmetry imposed | C2 |
| Energy filter slit width (eV) | 20 |
| Tilt range (min/max, step) | −60°/+60, 3° |
| Tilt scheme | Dose-symmetrical (Hagen scheme) |
| Movie recording | 10 frames per tilt |
| Electron dose per tilt image (e⁻ Å⁻²) | 2.92 |
| Initial subtomogram (no.) | 77,436 |
| Final subtomogram (no.) | 15,116 |
| Map resolution (Å) | 10 |
| Fourier shell correlation (FSC) threshold | 0.143 |
| Map resolution range (Å) | 10–12 |
| **Refinement** | |
| Initial model used for the BAR domains (PDB) | 8A1G |
| SNX1-PX (PDB) | 2I4K |
| SNX5-PX in complex with CI-MPR (PDB) | 6N5Y |
| Atom inclusion inside the map (%) | 97 |
| Contour level | 0.8 |
| Map sharpening B factor (Å²) | −157.154 |
| Model composition | |
| Nonhydrogen atoms | 6,317 |
| Protein residues | 777 |
| Ligands | 0 |
| B factors (Å²) | |
| Protein | 79.77 |
| Ligand | — |
| R.m.s. deviations | |
| Bond length (Å) | 0.029 |
| Bond angle (°) | 2.766 |
| **Validation** | |
| MolProbity score | 2.85 |
| Clashscore | 64.61 |
| Poor rotamers (%) | 2.47 |
| Ramachandran plot | |
| Favored (%) | 96.10 |
| Allowed (%) | 3.38 |
| Disallowed (%) | 0.52 |

Notably, the lipid-binding regions correlate with the lattice contacts of the coat (Fig. 4b). A closer inspection of the SNX1–SNX5 lattice contacts provided insights of the individual secondary-structure elements involved in the assembly of the coat. BAR$^{tip}$-to-BAR$^{tip}$ contacts occur between the side-tips of the α2 helices from each BAR domain, resembling an SNX1 coat[24], whereas BAR$^{tip}$-to-PX contacts involve the tip of the α3 helix from each BAR domain with the start of the amphipathic helix (SAH) in the adjacent molecule (Fig. 4d and Supplementary Video 2). Interestingly, the recognition site for CI-MPR is in close proximity to the SAH of SNX5 and the BAR$^{tip}$ of SNX1 (Fig. 4e). The spatial proximity between these elements, the increased affinity for CI-MPR in the presence of SNX1 and the induction of more homogeneous tubulation in the presence of cargo suggest that cargo recognition, coat assembly and membrane deformation are integrated through cooperative interactions. Indeed, introducing three Ala mutations in both SAH regions (SNX1-SAH$^{3A}$:SNX5-SAH$^{3A}$), or replacing both BAR$^{tip}$ regions with Gly-Ser linkers (SNX1-BT*:SNX5-BT*), impaired liposomal tubulation (Fig. 4f–h) and mildly reduced the binding of CI-MPR in solution (Extended Data Fig. 8a,b). Importantly, these weak allosteric responses in solution are likely strengthened on the membrane by the cumulative effects of serial binding events during coat formation. It should be noted, however, that despite WT and mutant proteins displaying similar circular dichroism spectra, indicative of unaffected secondary structures (Extended Data Fig. 8c), the SAH$^{3A}$ and BT* mutants exhibited lower association with synthetic liposomes (Extended Data Fig. 8d). In particular, the BT* mutants displayed a major loss in the recruitment of SNX5-BT* (Extended Data Fig. 8d). Although we do not have a mechanistic explanation for the unexpected SNX5-BT* behavior, the BT* mutants clearly affected coat assembly and the binding correlation between protomers in synthetic liposomes.

**ESCPE-1 function requires lattice interactions**

To test the physiological relevance of the BAR$^{tip}$-to-BAR$^{tip}$ and BAR$^{tip}$-to-PX interactions revealed by the structural analyses, we examined the effect of mutating residues involved in those interactions on the membrane recruitment and cargo sorting function of SNX1 and SNX5 in cells. To this end, we performed double knockout (KO) of SNX1 and SNX2 (SNX1-2), and SNX5 and SNX6 (SNX5-6), in the human osteosarcoma cell line HT1080. We observed that SNX1-2 KO not only abolished expression of the target proteins, but also drastically reduced the levels of SNX5 and SNX6 (Fig. 5a), indicating that SNX5/6 requires SNX1/2 for stability. Conversely, SNX5-6 KO abolished expression of the target proteins and reduced the levels of SNX1 and SNX2 (Fig. 5a). However, the reduction of SNX1/2 levels in this case was less drastic, consistent with SNX1/2 forming stable dimers in the absence of SNX5/6.

Next, we substituted nine or ten amino-acid residues from the BAR$^{tip}$ (BT* mutants) and/or SAH (SAH* mutants) structural elements by Gly-Ser linkers in the context of hemagglutinin (HA)-tagged or green fluorescent protein (GFP)-tagged SNX1 and SNX5 (Fig. 5b), and examined the effect of these substitutions on the rescue of SNX functions in SNX KO cells. We found that stable retroviral transduction of all of the HA-tagged SNX mutants restored the levels of their endogenous heterodimeric partners to the same extent as the WT counterparts (Fig. 5c). Live-cell imaging of SNX1-2 KO cells expressing GFP-tagged SNX constructs showed that WT GFP-SNX1 was associated with endosomes (Fig. 5d), as previously shown[7]. The GFP-SNX1-BT* and GFP-SNX1-SAH* mutants exhibited less association with endosomes, and the GFP-SNX1-BT-SAH* mutant was almost completely cytosolic (Fig. 5d,e). Likewise, experiments with SNX5-6 KO cells showed that WT GFP-SNX5 was associated with endosomes, GFP-SNX5-BT* was less associated, and GFP-SNX5-SAH* and GFP-SNX5-BT-SAH* were virtually all cytosolic (Fig. 5d,e). Similar results were obtained by immunofluorescence microscopy of GFP-tagged SNX constructs expressed by transient transfection in WT HT1080 cells (Extended Data Fig. 9); the only difference being that GFP-SNX5-BT* was even more defective, likely due to having to compete with endogenous SNX5/6 for association with endosomes.

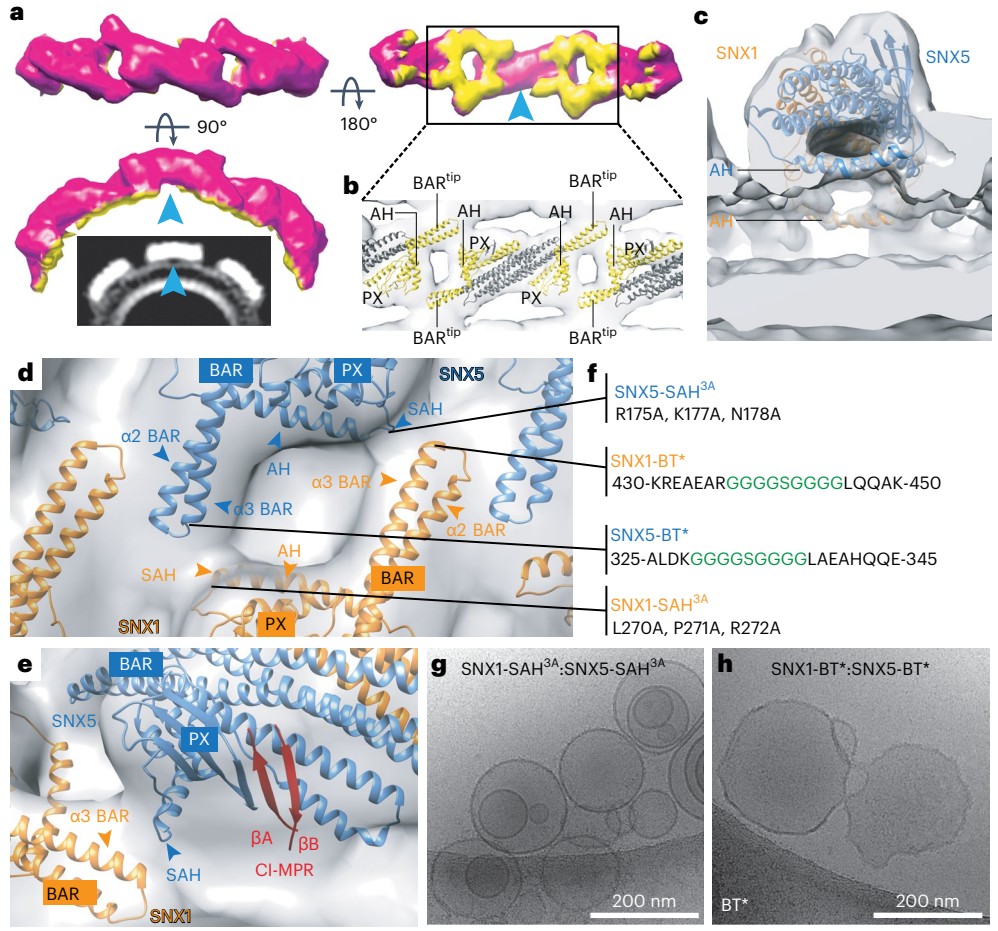

**Fig. 4 | Lattice-forming contacts of the ESCPE-1 coat drive membrane tubulation. a**, Isosurface of the three-particle subtomogram average radially colored based on distance from tube axis to the outer membrane leaflet in yellow. Blue arrowhead indicates the middle section of one BAR heterodimer that does not contact the membrane. **b**, Zoomed-in view of the membrane-associating face with the fitted atomic model of the SNX1–SNX5 heterodimer. Structural elements in contact with the membrane are in yellow. AH, amphipathic helix. **c**, Cross-section through the one-particle subtomogram average illustrates the two amphipathic helices that connect the PX and BAR domains in SNX1 and SNX5. **d**, Overlay of ribbon and the electron-density map showing a top view of the structural elements interconnecting neighboring molecules. **e**, Rotated view from **d** showing the CI-MPR binding site. **f**, Location of introduced mutations. **g,h**, Incubation of the SNX1–SNX5 heterodimer with mutations at both SAH regions (**g**) or at both BAR^tip regions (**h**) is unable to induce tubulation of synthetic liposomes. Data in **g** and **h** are representative of two independent experiments.

Finally, we examined the distribution of endogenous CI-MPR in SNX1-2 KO and SNX5-6 KO HT1080 cells stably transduced with the corresponding WT and double BT-SAH* mutants (the mutants that exhibit the least membrane association) (Fig. 6a). These experiments showed that SNX1-2 KO or SNX5-6 KO increased the colocalization of the CI-MPR with the early-endosomal marker EEA1 (Fig. 6b,c), indicative of reduced export of the CI-MPR from endosomes[7]. Expression of HA-tagged SNX1 (HA-SNX1) or SNX5 (HA-SNX5) in the corresponding KO cells decreased the CI-MPR–EEA1 colocalization, whereas expression of HA-SNX1-BT-SAH* or HA-SNX5-BT-SAH* had little or no effect (Fig. 6b,c), demonstrating that these mutants were functionally inactive.

Taken together, these experiments demonstrated that interactions mediated by the BAR^tip and SAH elements of SNX1 and SNX5 are important for association of these proteins with endosomes and for their function in promoting the export of CI-MPR from endosomes.

## Discussion

We have investigated how ESCPE-1 orchestrates tubular-based cargo sorting, revealing a striking relationship between membrane interaction elements, cargo recognition and coat formation. The crystal structure of the SNX1^BAR–SNX5^BAR complex presented here, together with analytical modeling, revealed that SNX1 comprises a hydrophobic binding hot spot compatible with homodimerization and heterodimerization, whereas SNX5 uses distinct patches of polar residues to preclude homodimerization and enhance heterodimerization. Dimerization can be further modulated by post-translational modifications, as happens with SNX5 phosphorylation at serine 226, which prevents heterodimerization with SNX1 or SNX2, disrupting CI-MPR trafficking and micropinocytosis[15]. Most likely, similar patterns guide the formation of other PX-BAR complexes such as the heterodimers formed by SNX4 with either SNX7 or SNX30 involved in autophagosome biogenesis[30], and by SNX4 with SNX5 involved in cargo sorting from autolysosomes[31].

In higher metazoans, retromer does not form stable complexes with SNX1/2–SNX5/6 (refs. 7,25). The finding that the geometric distribution of retromer contacts over the VPS5 lattice in *C. thermophilum*[23] is not conserved in the mammalian SNX1–SNX5 lattice, together with the lack of direct association in flotation assays, supports the notion of functional diversification between the mammalian and yeast retromer. Indeed, ESCPE-1 can engage SNX27-retromer through the interaction between SNX1 and the SNX27-FERM domain to promote recycling of certain cargos[8,32,33]. However, this SNX27-retromer–ESCPE-1

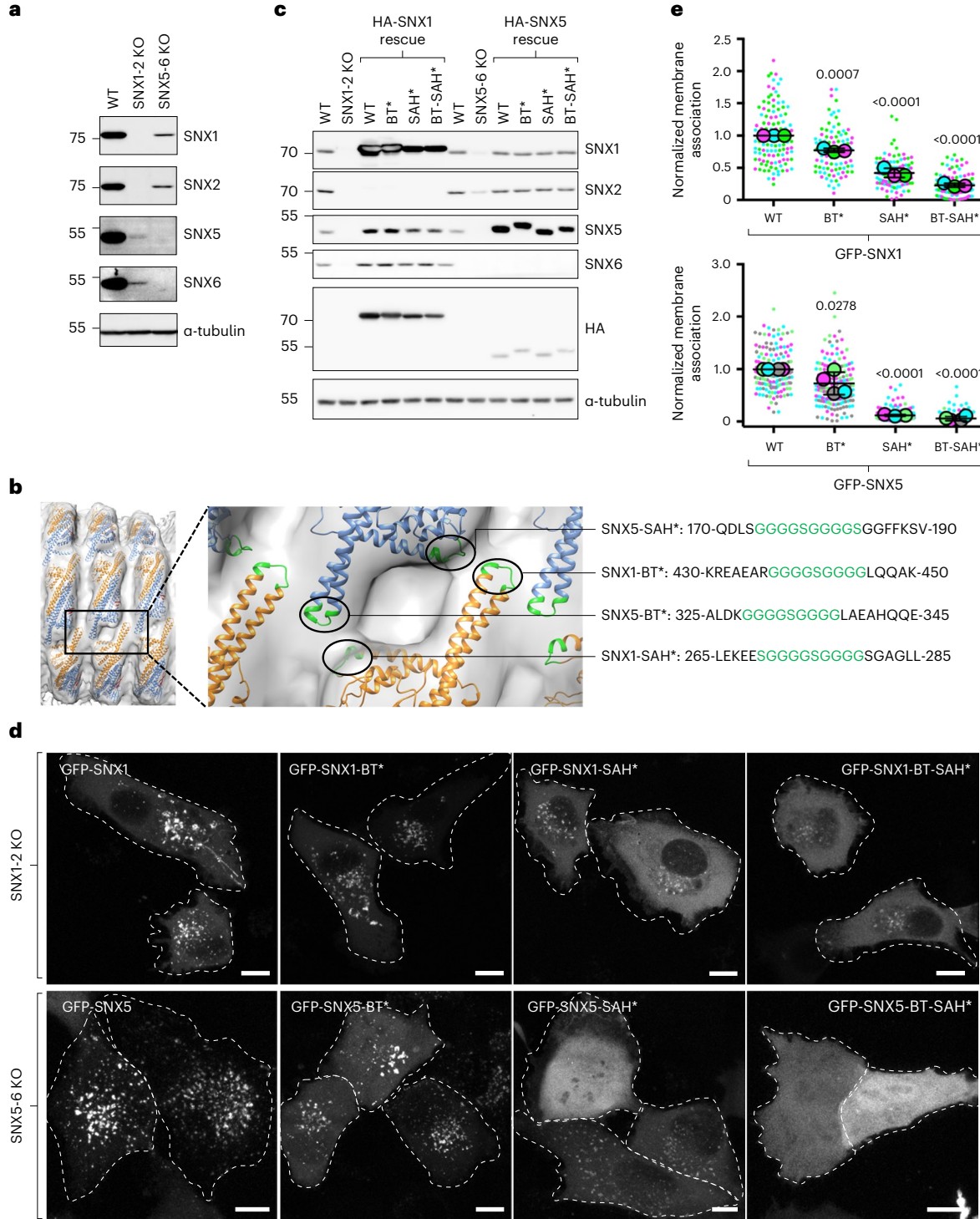

**Fig. 5 | Requirement of BAR^tip-to-BAR^tip and BAR^tip-to-PX interactions for endosomal association of SNX1 and SNX5. a**, Immunoblot analysis of WT, double SNX1-2 KO and double SNX5-6 KO HT1080 cells using antibodies to the proteins indicated on the right. **b**, Mutations introduced in the BAR^tip and PX^SAH regions. **c**, Immunoblot analysis of WT, double SNX1-2 KO and double SNX5-6 KO cells stably transduced with plasmids encoding HA-tagged WT and mutant SNX1 and SNX5 constructs, using antibodies to the proteins indicated on the right. **d**, Double SNX1-2 KO and double SNX5-6 KO HT1080 cells were transiently transfected with plasmids encoding GFP-tagged WT or mutant SNX constructs, as indicated in the figure. Cells were imaged live by confocal microscopy. GFP channels are shown in gray scale, and cell edges are indicated by dashed lines.

Scale bars, 10 μm. **e**, Efficiency of SNX recruitment to punctate intracellular membranes was estimated using the Find Maxima function of ImageJ/Fiji. Fewer local maxima are identified in cells with increased cytosolic GFP signal. Data in **a** and **c** are representative of three independent experiments. For **d** and **e**, the number of local maxima for at least 20 cells per condition was normalized to the average number of local maxima in WT cells and plotted as SuperPlots. In **e**, horizontal lines indicate the mean ± s.d. of the means from three experiments for SNX1 (top panel) and four experiments for SNX5 (bottom panel). Statistical significance was calculated by one-way ANOVA with multiple comparisons to the SNX WT control using Dunnett's test; $P$ values are indicated on the plots.

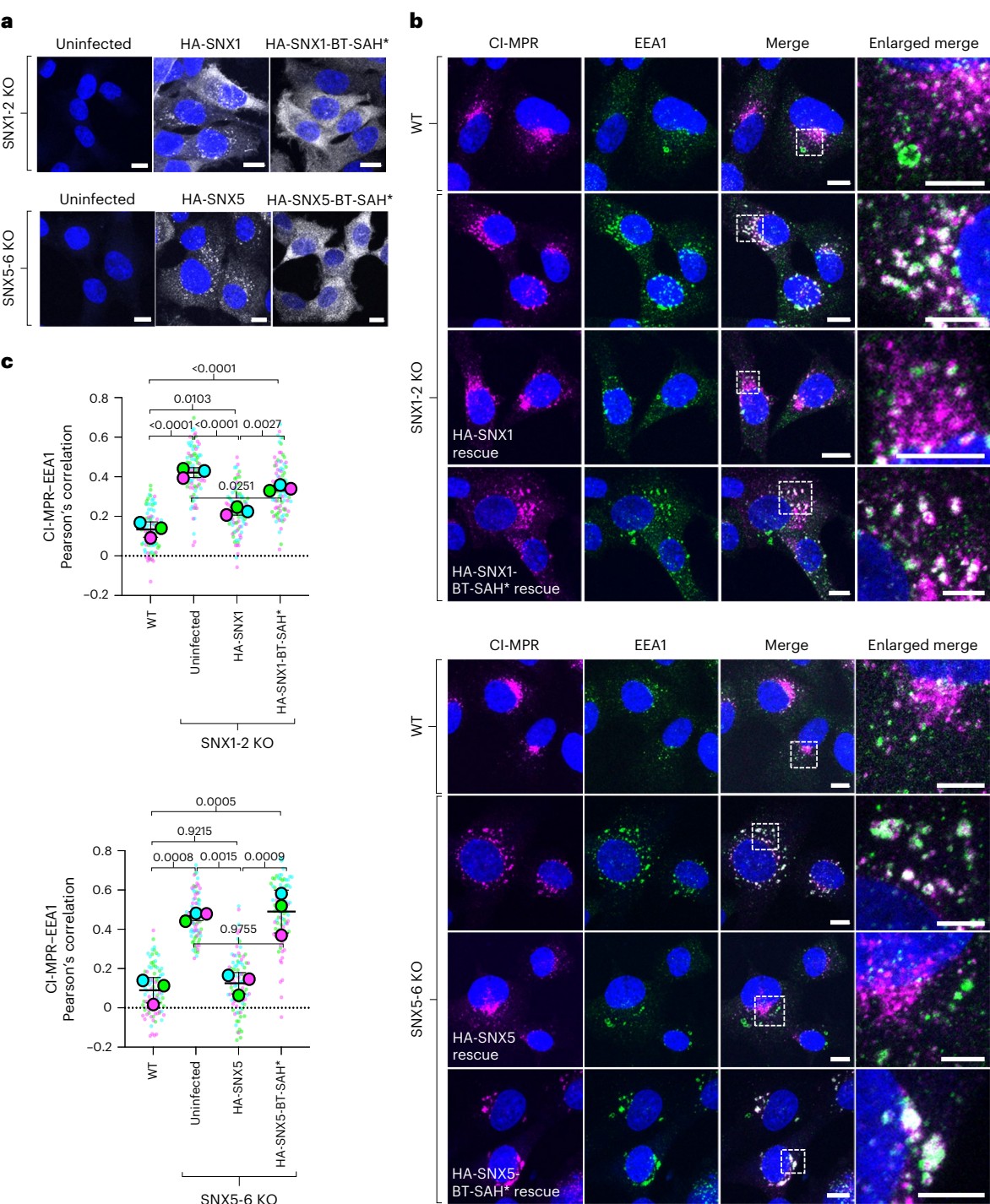

**Fig. 6 | Requirement of BAR^tip-to-BAR^tip and BAR^tip-to-PX interactions for the function of SNX1 and SNX5 in the export of CI-MPR from endosomes.**
**a**, Immunofluorescence microscopy of fixed-permeabilized double SNX1-2 KO and double SNX5-6 KO HT1080 cells stably transduced with HA-tagged WT or BT-SAH* mutant SNX1 or SNX5. Cells were immunostained for the HA epitope and nuclei (DAPI; blue). Scale bars, 10 μm. The experiment was repeated twice with similar results. **b**, WT, untransduced and stably transduced double KO HT1080 cells were immunostained for the CI-MPR (magenta), early endosomes (green) and nuclei (blue), and examined by confocal fluorescence microscopy.

Scale bars, 10 μm. Enlarged views of the boxed areas in the merged images are shown on the right column. Scale bars, 5 μm. **c**, PCC of colocalization between CI-MPR and EEA1 from experiments such as that shown in **b**. PCCs were calculated for 30 cells in each of three independent experiments. Data are represented as SuperPlots showing the individual data points in each experiment, the mean from each experiment and the mean ± s.d. of the means. Statistical significance was analyzed by one-way ANOVA with multiple comparisons using Tukey's test; *P* values are indicated on the plots.

'supercomplex' has been proposed to be of a transient nature at the emerging membrane bud from where cargo is handed to ESCPE-1 (ref. 34). Given that tubular and planar membranes impose distinct spatial restrictions, it is possible that for retrieval certain cargos, retromer

and other factors could associate with ESCPE-1 in pseudoplanar membranes through a different lattice organization.

The ESCPE-1 coat also displays less aerial density and surface coverage on the tube than the SNX1 and VPS5 scaffolds (Extended Data

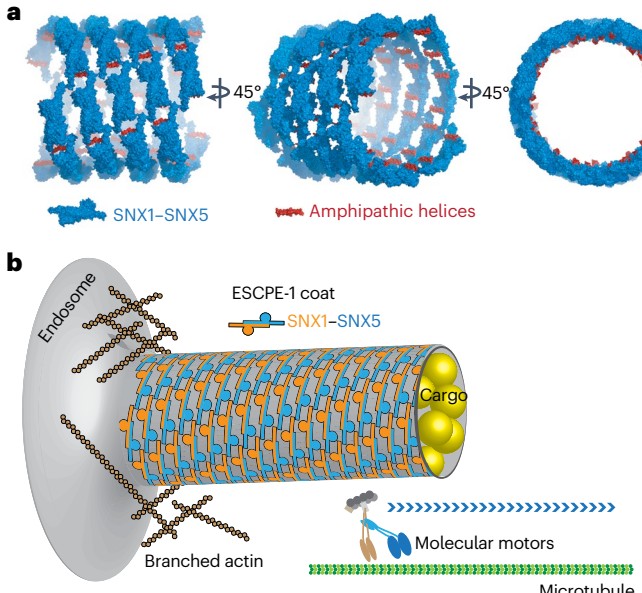

**Fig. 7 | A schematic model for how ESCPE-1 assembles on membranes. a**, 3D reconstruction of the ESCPE-1 coat. The SNX1–SNX5 atomic model has been fitted and symmetrized according to the arrangement calculated from the three-start helix parameters. Amphipathic helices responsible for the lateral expansion of lipids are in red. **b**, Model for the endosomal sorting of ESCPE-1 cargos. The presence of cargo promotes SNX1–SNX5 recruitment and the shaping of a tubular carrier in conjunction with pushing forces produced by actin polymerization and pulling forces produced by molecular motors.

Fig. 7a). This marked difference in the ESCPE-1 lattice organization allows larger exposed protein surfaces along the tube, which for SNX5 or SNX6 would be critical for cargo binding[5] and for the interaction with microtubule motors such as the dynactin component p150^Glued (ref. 35,36), or for WASH-mediated actin polymerization via RME-8 interaction with SNX1 (refs. 37–39). Furthermore, the finding that tubules can be formed using different numbers of helix starts has two important implications. By adding starts, the distance between turns increases, possibly speeding up cargo displacement during the coating process. Furthermore, additional starts could expand the diameter of the tube without altering the helix angle, which might be important for the recruitment of cytosolic factors and/or for keeping contacts with the membrane in an orientation that facilitates tubulation.

Membrane remodeling by SNX1 involves an amphipathic helix between the PX and BAR domains that inserts into the outer leaflet to generate curvature and that is predicted to exist in other SNX-BAR proteins[13,24]. In the present study, we found an interaction between the tips of the BAR heterodimers and the PX domains. In particular, the BAR^tip-to-PX contacts, which involve short amino-acid stretches adjacent to the amphipathic helices in SNX1 and SNX5, are critical for membrane remodeling and CI-MPR transport. In this sense, the BAR^tip-to-PX contacts not only strengthen the interaction with cargo, but also facilitate the alignment of the amphipathic helix parallel to the axis of the tube, thus supporting a membrane-spanning orientation (Fig. 7a and Supplementary Video 2). This cooperative assembly coordinates cargo capture and membrane deformation in a single-layer coat via local interactions that concomitantly promote ESCPE-1 recruitment and lattice contacts, thus creating a scaffolding system that repositions the amphipathic helices for the generation of global curvature. This monolayer organization contrasts sharply with the other canonical double-layer coats like clathrin, COPI and COPII, where the inner layer binds to cargo and lipids, and the outer layer forms scaffolds to bend the membrane. Thus, ESCPE-1 features an all-in-one layer design (Fig. 7b) ideally suited for rapid coating of tubes with variable length.

This structural framework hints at the possibility that other PX-BAR heterodimeric complexes form similar arrays[30,31].

The observation that SNX1–SNX5 heterodimers in solution are more stable than SNX1 homodimers does not exclude the existence of SNX1 homodimers. Indeed, in addition to SNX5 phosphorylation, which could shift the equilibrium toward SNX1 homodimers, the number of SNX1 molecules is slightly higher than that of SNX5 molecules[40]. In this sense, the ability to form two different coat architectures provides evidence for functional specialization in separate sorting pathways or, alternatively, separate roles within the same sorting tube. All of these insights into the diverse PX-BAR coat organizations are just the beginnings of our understanding of how sorting nexins integrate the proper distribution of endosomal cargos to maintain cellular function.

## Online content

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

## Methods

### Recombinant DNA procedures

DNAs encoding SNX1[FL], SNX5[FL], SNX1[BAR], SNX1 (2 mut)[†], SNX1-BT* and SNX1-SAH[3A] were cloned into pET28-SUMO3 vector (European Molecular Biology Laboratory (EMBL)) with an N-terminal cleavable 6xHis-SUMO3 tag. Human DNAs encoding SNX1[PX] and SNX5[PX] were cloned in pHisMBP-Parallel2 (ref. [41]) with an N-terminal cleavable 6xHis-maltose binding protein (MBP) tag. DNAs encoding SNX5[FL], SNX5[BAR], SNX5 (4 mut)[‡], SNX5 (6 mut)[§], SNX5*, SNX5**, SNX5-SAH[3A] and SNX5-BT* were cloned into pGST-Parallel2 vector[41] with a cleavable N-terminal glutathione S-transferase (GST) tag. DNA encoding SNX1[BAR†] was cloned in pDB.GST with a cleavable N-terminal GST tag[42]. The cytosolic tail of CI-MPR with a cysteine at the N terminus for chemical cross-linking was cloned into pET28-SUMO3 vector (EMBL) with an N-terminal cleavable 6xHis-SUMO3 tag. The remaining CI-MPR constructs were cloned with an additional C-terminal GST cleavable tag. Site-directed mutagenesis was carried out using the Phusion Site-Directed Mutagenesis Kit (Thermo Fisher) according to the manufacturer's directions. For cellular assays, plasmids encoding WT GFP-SNX1 and GFP-SNX5 were a gift from D. Gershlick (Eunice Kennedy Shriver National Institute of Child Health and Human Development (NICHD), National Institutes of Health (NIH)). BT*, SAH* and BT-SAH* mutations were introduced into GFP-SNX1 and GFP-SNX5 using QuikChange Lightning Site-Directed Mutagenesis. N-terminally HA-tagged SNX1 (WT, BT*, SAH* or BT-SAH*) and SNX5 (WT, BT*, SAH* or BT-SAH*) were amplified by PCR using the GFP-tagged constructs as templates and the following pairs of oligonucleotides (restriction enzyme sites are in bold): for SNX1, forward primer, TGATC C**GCGG CCGC**G CCACC ATGTA CCCGT ACGAT GTTCC TGACT ATGCG GGCAT GGCGT CGGGT GGTGG TGG, reverse primer, GCG**GA ATTC**T TAGGA GATGG CCTTT GCCT; for SNX5, forward primer, TGATC C**GCGG CCGC**G CCACC ATGTA CCCGT ACGAT GTTCC TGACT ATGCG GGCAT GGCCG CGGTT CCCGA GTTG, reverse primer, GCG**GA ATTC**T CAGTT ATTCT TGAAC AAGTC. The resulting amplicons were subcloned into NotI-EcoRI–digested pQCXIP vector (Takara Bio, S3145). All constructs were verified by DNA sequencing.

### Protein expression and purification

Constructs corresponding to CI-MPR$_{2330-2491}$ were expressed in *Escherichia coli* Rosetta (DE3) cells, and the rest were expressed in *E. coli* BL21 (DE3) cells. Cells were grown in Luria–Bertani (LB) broth at 37 °C until reaching an optical density at 600 nm of 0.8. At that point, protein expression was induced for 16 h at 18 °C with 1 mM isopropyl-β-D-thiogalactopyranoside (IPTG). In the following paragraphs, we have grouped common purification protocols for all of the proteins used in this work.

For the purification of SNX1[PX] and SNX5[PX], cell pellets were lysed by high-pressure homogenization (25 kpsi) in buffer A (300 mM NaCl, 1 mM DTT, 25 mM Tris-HCl pH 8.0) supplemented with 10 mM imidazole, 0.5 mM PMSF and 5 mM benzamidine. After clearing the bacterial lysates by centrifugation for 45 min at 50,000 × *g*, the soluble fraction was incubated in batch for 2 h with Protino Ni-NTA beads (Macherey-Nagel). Beads were extensively washed with buffer A, and the bound protein was then eluted with buffer A complemented with 250 mM imidazole. To remove the N-terminal HisMBP tag, we added tobacco etch virus (TEV) protease to the eluted sample. The mixture was dialyzed overnight against buffer B (150 mM NaCl, 1 mM DTT, 25 mM Tris-HCl pH 7.5). Subsequently, ion-exchange chromatography (HiTrap Q HP, GE Healthcare) was performed using a gradient of 15–1,000 mM NaCl, followed by a size-exclusion chromatography (Superdex 75 16/60, GE Healthcare) in buffer A for SNX5[PX] and in buffer C (500 mM NaCl, 1 mM DTT, 25 mM Tris-HCl pH 7.5) for SNX1[PX].

SNX1[FL] and SNX1 (2 mut)[†] were expressed with a 6xHis-SUMO3 tag. Cell lysis and Ni-NTA affinity chromatography were performed as described for SNX1 and SNX5[PX]. SUMO-specific protease 2 (SENP2) was added to the eluted protein for cleaving the N-terminal 6xHis-Sumo3 tag and dialyzed overnight in buffer D (100 mM NaCl, 1 mM DTT, 25 mM Tris-HCl pH 7.5). This tag-removal step was omitted for certain downstream applications. A second Ni-NTA chromatography step was performed to remove the cleaved tag and uncleaved protein. SNX1[FL] or SNX1 (2 mut)[†] were subsequently purified by ion-exchange chromatography (HiTrap Q HP, GE Healthcare) with a gradient of 10–1,000 mM NaCl, followed by two rounds of size-exclusion chromatography (Superdex 200 16/60, GE Healthcare) in buffer A.

For SNX1[BAR] purification, the chromatographic steps were the same as for SNX1[FL] but with slightly different buffer compositions. For lysis and Ni-NTA affinity, buffer B was used, complemented with 10 mM imidazole. Bound material was eluted with 250 mM imidazole in buffer B. Subsequent dialysis of the sample was carried in buffer A, followed by size-exclusion chromatography. The size-exclusion chromatography was repeated for three cycles to ensure a completely homogeneous sample.

For SNX1[BAR†] purification, the cellular pellet was lysed by high-pressure homogenization (25 kpsi) in buffer B supplemented with 0.5 mM PMSF and 5 mM benzamidine. After clearing the bacterial lysates by centrifugation for 45 min at 50,000 × *g*, the soluble fraction was incubated in batch for 2 h with glutathione Sepharose 4B (GE Healthcare). Protein was released from the resin by overnight cleavage of the N-terminal GST tag with TEV protease. Next, ion-exchange chromatography (HiTrap Q HP) was performed using a gradient of 15–1,000 mM NaCl, followed by three cycles of size-exclusion chromatography (Superdex 75 16/60) in buffer A.

Purification of SNX5[FL], SNX5 (4 mut)[‡] and SNX5 (6 mut)[§] followed the same procedures for lysis and affinity chromatography as for SNX1[BAR†], but with buffer E (1 M NaCl, 2 mM DTT, 50 mM Tris-HCl pH 8.0) complemented with 1 mM EDTA. Cleavage of the GST tag was performed with TEV protease in batch mode for 2 h at 4 °C. Then, the supernatant was filtered and loaded into size-exclusion chromatography (Superdex 200 16/60) using buffer E.

For SNX1[FL]–SNX5[FL] purification, individually overexpressed SNX1 and SNX5 cell pellets were mixed and lysed together. Lysis and affinity chromatography with glutathione Sepharose 4B were done as for SNX1[BAR] with buffer C. Then, SNX1[FL]–SNX5[FL] was further purified with Ni-NTA beads following the procedure described for SNX1[FL] and SNX1[BAR†] but with buffer A. The bound protein was eluted with 250 mM imidazole and dialyzed overnight with SENP2. Next, the protein sample was subjected to ion-exchange chromatography (HiTrap Q HP) using a linear gradient elution of 15–1,000 mM NaCl, followed by size-exclusion chromatography (Superdex 200 16/60) in buffer A.

Purification of other heterodimers used in this study such as SNX1[BAR]–SNX5[FL], SNX1[BAR]–SNX5[BAR], SNX1-SAH[3A]—SNX5-SAH[3A] and SNX1-BT*–SNX5-BT* followed the same purification procedure described for SNX1[FL]–SNX5[FL].

For the CI-MPR constructs, the purification followed the same protocol as for SNX1[FL] but the last size-exclusion chromatography step was performed on Superdex 75 16/60. Constructs of CI-MPR expressed with a double tag followed the same protocol as before but included an initial affinity chromatography step with glutathione Sepharose 4B and a tag cleavage step in solution similar to what has been described previously for SNX1[BAR†].

For purification of the SNX5–SNX5* and SNX5–SNX5** complexes, individually overexpressed SNX5 and SNX5* or SNX5** cell pellets were mixed and lysed together. Lysis and affinity chromatography with glutathione Sepharose 4B were done as for SNX1[BAR†] with buffer A. Complexes were further purified with Ni-NTA beads as previously described for SNX1[FL] in buffer A. These complexes displayed significantly lower yields and increased aggregation tendency compared with SNX1–SNX5.

Retromer complex and SNX3 protein were purified following the procedure described previously[29]. Protein concentration for all

proteins used in this work was calculated using their theoretical extinction coefficient.

## Protein crystallization, data collection and structure determination

The SNX1$^{BAR}$–SNX5$^{BAR}$ complex was crystallized using vapor diffusion methods. Upon evaluation of numerous crystallization screens from various commercial sources, diamond-shaped crystals were obtained after 2–3 days at 18 °C, in the MIDAS condition 1-17 (Molecular Dimensions). This crystallization condition was further refined using the sitting-drop method and larger volumes. Good crystals were obtained by mixing 1 µl of the heterodimer at 8 mg ml$^{-1}$ with 1 µl of the precipitant solution containing 100 mM HEPES pH 7, 11% (w/v) polyvinyl alcohol and 10% (v/v) 1-propanol. However, the best diffracting crystals were obtained after several microseeding steps. Native crystals were cryoprotected by quick-soaking into mother liquor supplemented with 20% (v/v) glycerol before being flash-frozen in liquid nitrogen. Suitable derivatives were obtained by 10 min of soaking in the cryoprotectant solution supplemented with 10 mM K$_2$PtBr$_4$.

Crystallographic native and derivative datasets were collected with the software MxCuBE at XALOC beamline in the ALBA synchrotron facility (Cerdanyola del Valles, Spain) using a Pilatus 6 M detector. Diffraction images were indexed, integrated and scaled using XDS[43] or MOSFLM/SCALA[44]. The SNX1$^{BAR}$–SNX5$^{BAR}$ heterodimer structure was solved by multiple anomalous dispersion (MAD) using three datasets of the Pt derivative (Peak, Inflection and Remote) collected from different regions of the same crystal to avoid radiation damage. Heavy atom positions were identified using SHELXC/D as implemented in autoSHARP[45]. After phasing, subsequent density modification using SOLOMON[46] gave a starting map into which 14 chains with 702 residues were automatically built with Buccaneer[47]. Next, iterative refinement with Phenix and manual building in Coot[48] yielded a final model with two heterodimers in the asymmetric unit (4 chains with 792 residues). This model was used as a template for molecular replacement with the native dataset that diffracted to a resolution of 2.8 Å. The final structure has an $R_{factor}$ and an $R_{free}$ of 22.4% and 27.8%, respectively. Data collection statistics for each dataset are shown in Table 1.

## Computation of per-residue docking energy

We estimated the residue contribution to the binding energy with the resEnergy pyDock module[14]. Taking the structure of a complex as input, the module computes pyDock docking energy partitioned at the residue level, giving a much more detailed description of the energetic landscape of the interaction. We calculated the residue contribution of the SNX1–SNX1 homodimer (PDB 4FZS) and the SNX1$^{BAR}$–SNX5$^{BAR}$ heterodimer (this work). Additionally, we computed the mean residue docking energy of an ensemble of 20 SNX1–SNX1 homodimer models. We generated the ensemble with the homology modeling package Modeller v.9.17 (ref. 49), configured with default parameters, and using the SNX1$^{BAR}$–SNX5$^{BAR}$ heterodimer structure as a template (this work).

## Size-exclusion chromatography coupled with multiangle light scattering

Protein oligomerization states were determined by SEC–MALS. For each assay, 100 µl of protein sample at 1 mg ml$^{-1}$ in 150 mM NaCl, 0.5 mM TCEP, 10 mM HEPES pH 7.5 buffer were analyzed with high-performance liquid chromatography (HPLC; Shimadzu) coupled with a Superdex 200 Increase 10/300 GL column (Cytiva). The process was monitored using DAWN HELEOS II and Optilab rEX detectors (Wyatt Technology) to measure the multiangle light scattering and refractive index of the sample, respectively. The results were analyzed using ASTRA v.6 software (Wyatt Technology). Bovine serum albumin (BSA) was used as the calibration standard.

## Isothermal titration calorimetry

ITC measurements were carried out at 25 °C on a MicroCal PEAQ-ITC titration microcalorimeter (Malvern Panalytical). All proteins and peptides used in this work were dialyzed overnight at 4 °C against 300 mM NaCl, 0.5 mM TCEP, 25 mM HEPES pH 7.5 buffer. Before titration, samples were tempered at 25 °C and degassed for 5 min in a Thermo Vac. The titration sequence consisted of an initial 0.4 µl injection to prevent artifacts (not used in data fitting), followed by 24 or 18 injections of 2 s and 1.2 µl or 2 µl with a spacing of 150 s between them. Heat of dilution used to correct the experimental data was performed under the same conditions. Results were fitted and integrated to a one-site model using the MicroCal PEAQ-ITC software (Malvern Panalytical). Final graphs were prepared using Origin ITC software (MicroCal). Values for the binding constant ($K_a$, $K_d = 1/K_a$), the molar binding stoichiometry, binding enthalpy, free energy and entropy of binding were obtained after data analysis. For ITC analysis of the interaction between cargo CI-MPR and heterodimer (SNX1–SNX5), 426 µM SUMO-CI-MPR $_{(2330-2491)}$ or 249 µM CI-MPR$_{2347-2375}$ peptide was titrated into 18–19 µM SNX1–SNX5 complex. For studying the interaction of CI-MPR with individual molecules of the heterodimer, 398 µM SUMO-CI-MPR$_{2330-2491}$ or 242 µM CI-MPR$_{2347-2375}$ peptide was titrated into 18 µM SNX5 or 19.6 µM SNX1, respectively. To reveal the subdomains important for the interaction with cargo, 234 µM CI-MPR$_{2347-2375}$ peptide was titrated into 8.42 µM SNX5$^{PX}$, 746 µM SNX1$^{PX}$ was titrated into 16 µM CI-MPR$_{2347-2375}$ peptide and 805 µM CI-MPR$_{2347-2375}$ was titrated into 21 µM SNX1$^{BAR}$–SNX5$^{BAR}$ complex. Finally, to identify the cooperative interaction between subdomains, two heterodimer combinations were tried, 445 µM CI-MPR$_{2347-2375}$ peptide was titrated into 100 µM SNX1$^{BAR}$ and 20 µM SNX5$^{PX}$ mix, and 236 µM CI-MPR$_{2347-2375}$ was titrated into 15 µM SNX1$^{BAR}$–SNX5 partial heterodimer. Data are the mean ± s.d. of a minimum of two replicate titrations for each assay.

## Circular dichroism

Proteins were dialyzed overnight at 4 °C against PBS. Circular dichroism spectra were acquired at 25 °C using a Jasco J-710/J-810 spectropolarimeter. Data points were collected from 200 nm to 250 nm using a cuvette with a path length of 0.1 cm. Proteins were measured at a concentration of 1 µM. Ellipticity was converted to mean residue ellipticity.

## Ni-NTA-tagged pull down

For analysis of the SNX1–SNX5 interface, 5 µM His-SUMO-SNX1 or mutant His-SUMO-SNX1 (2× mut)$^†$ was incubated with 10 µM SNX5, SNX5$^‡$ or SNX5$^§$ for 1 h at 4 °C in 150 mM NaCl, 10 mM imidazole, 0.5 mM TCEP, 10 mM HEPES pH 7.5. Then, 20 µl of equilibrated Protino Ni-NTA beads was added to 100 µl of the protein mixture and incubated with gentle agitation for 1 h at 4 °C. Beads were washed three times with 500 µl of previously described buffer, followed by 5 min of centrifugation at 4 °C and 1,000 × $g$. Bound proteins were eluted with 2X Laemmli buffer, boiled, resolved by SDS–polyacrylamide gel electrophoresis (SDS–PAGE) and visualized by Coomassie blue staining.

## Liposome formulation

Stock solutions of lipids were prepared at 10 mg ml$^{-1}$ in chloroform for 1,2-dioleoyl-*sn*-glycero-3-phosphocholine (DOPC), 1,2-dioleoyl-*sn*-glycero-3-phosphoethanolamine (DOPE), 1,2-dioleoyl-*sn*-glycero-3-phospho-1-serine (DOPS) and 1,2-dioleoyl-*sn*-glycero-3-phosphoethanolamine-*N*-(lissamine rhodamine B sulfonyl) (Liss Rhod PE) (Avanti Polar Lipids), or dissolved in chloroform:methanol:water (20:9:1) at 0.2 mg ml$^{-1}$ for 1,2-dioleoyl-*sn*-glycero-3-phospho-(1′-myo-inositol) (PtdIns) (Avanti Polar Lipids), 1,2-dioleoyl-*sn*-glycero-3-phospho-(1′-myo-inositol-3′-phosphate) (PtdIns(3)P) (Avanti Polar Lipids), 1,2-dioleoyl-*sn*-glycero-3-phospho-(1′-myo-inositol-4′-phosphate) (PtdIns(4)P) (Avanti Polar Lipids), 1,2-dioleoyl-*sn*-glycero-3-phospho-(1′-myo-inositol-5′-phosphate) (PtdIns(5)P) (Avanti Polar Lipids), dipalmitoyl phosphatidylinositol 3,4-bisphosphate

(PtdIns(3,4)P2) (Echelon Biosciences), dipalmitoyl phosphatidylinositol 3,5-bisphosphate (PtdIns(3,5)P2) (Echelon Biosciences), 1,2-dioleoyl-*sn*-glycero-3-phospho-(1′-myo-inositol-4′,5′-bisphosphate) (PtdIns(4,5)P2) (Avanti Polar Lipids) and 1,2-dioleoyl-*sn*-glycero-3-phospho-(1′-myo-inositol-3′,4′,5′-trisphosphate) (PtdIns(3,4,5)P3) (Avanti Polar Lipids). Distearoyl-*sn*-glycero-3-phosphoethanolamine maleimide (DSPE-Mal; Nanosoft Biotechnology) was dissolved in chloroform:methanol (9:1).

Lipid molar ratios used for each experiment were as follows: for tubulation or cryo-EM assays, 45% DOPC, 30% DOPE, 20% DOPS, 5% PtdInsP; for flotation assays, 40–46% DOPC, 23–29% DOPE, 15–21% DOPS, 2–20% PtdInsP, 2% Liss Rhod PE; and for cargo-cross-linked flotation assays, 45–47% DOPC, 27–29% DOPE, 20–22% DOPS, 2% Liss Rhod PE, 1–6% DSPE-Mal. Lipid mixtures were kept under argon for 30 min at 37 °C to obtain a homogeneous mix. The organic solvent was removed in a stream of nitrogen, and residual traces were removed in a vacuum chamber for at least 1 h to generate a lipid film.

**Liposome flotation**

Large unilamellar vesicles (LUVs) were prepared by thin-film hydration in buffer F (120 mM NaCl, 0.5 mM TCEP, 10 mM HEPES pH 7.5) complemented with 10% sucrose, followed by ten freeze-thaw-vortexing cycles and extruded through a 200-nm pore-diameter polycarbonate membrane until the mixture became clear. When liposomes were to be cross-linked with CI-MPR, the hydration was done in buffer G (50 mM NaCl, 0.5 mM TCEP, 10% sucrose, 25 mM MES pH 6.5). For CI-MPR cross-linking, 100 μl of liposomes (1 mM) was mixed with 10 μM, 20 μM, 40 μM or 60 μM CI-MPR$_{2330–2491}$, matching the same maleimide molar ratios on liposomes. Samples were incubated overnight with argon at 4 °C. Then, the reaction was blocked with 2 mM DTT. Liposomes were centrifuged for 30 min at 20,000 × $g$, resuspended in 100 μl of buffer G and incubated 1 h at 4 °C with the corresponding SNXs. Samples were then gently mixed with 80% sucrose in buffer G to reach a final concentration of 30%, which was placed at the bottom of an Ultra-Clear tube (Beckman Coulter, 344057) and overlaid with 300 μl of 25% sucrose in buffer G, with another layer on top containing 50 μl of buffer G without sucrose. Tubes were ultracentrifuged at ~235,000 × $g$ for 1 h at 4 °C. The floated fraction (pink color) was collected and used for SDS–PAGE analysis. Samples were normalized relative to their absorbance at 573 nm associated with the Liss Rhod PE lipid. Liposome flotation assays functionalized with 5% of distinct PtdIns followed the same scheme described above with the omission of the cross-linking process. All proteins were incubated with liposomes at a final concentration of 25 μM.

**Liposome tubulation and cryo-electron microscopy data acquisition**

LUVs were prepared by thin-film hydration in buffer F as described above but using a 400-nm pore-diameter polycarbonate membrane for the extrusion step. For tubulation assays, 300 μM of liposomes was incubated with 15 μM of the corresponding SNXs and incubated overnight at 4 °C. For tubulation assays with cargo, 30 μM CI-MPR was incubated with 15 μM SNX1–SNX5 for 1 h at 4 °C and then incubated with liposomes as before. After overnight incubation, the mixture was centrifuged at 20,000 × $g$ for 1 h and resuspended in buffer F before being placed in glow-discharged Quantifoil R 2/2 300 mesh copper grids. Vitrification was performed on a Leica EM GP2 automatic plunger. Grids were frontside-blotted for 2 s at 90% humidity and 8 °C. Grids were then plunged into a liquid ethane batch and stored under L2 until visualization. Routinely, cryo-TEM images for evaluation of tubulation activity were acquired on a JEM-2200FS/CR transmission electron microscope equipped with an in-column omega energy filter and a K2 direct detector (Gatan) using DigitalMicrograph software (Gatan).

Sample preparation for cryo-ET followed the same procedure as before but in buffer G (100 mM NaCl, 0.5 mM TCEP, 10 mM HEPES pH 7.5). In this particular case, the CI-MPR–SNX1–SNX5 preincubated sample was added to 600 μM liposomes in buffer G. Before sample vitrification, BSA gold tracers (6 nm; Electron Microscopy Sciences) were added to the tubulation reaction in a 1:8 fiducials:reaction volume ratio. Four microliters of sample was incubated on a fresh glow-discharge grid for 30 s and blotted with filter paper before plunge-freezing in liquid ethane. Sample vitrification was performed on Vitrobot Mark II (FEI Company) at 8 °C and with a relative humidity close to saturation (90%). The high-resolution cryo-ET dataset was obtained in an FEI Titan Krios G3 microscope, coupled with a Gatan K2 Summit direct detector operated by SerialEM v.4.0 software, at the cryo-EM facility of Leicester University. Fifty-nine tomograms were acquired using a dose-symmetric scheme[50] with tilt range ±60°, 3° angular increment and defocus values between −2 μm and −5 μm. For each tilt angle, ten frames were recorded with a dose rate of 0.29 e$^-$ Å$^{-2}$ per frame and a total tomogram dose of 120 e$^-$ Å$^{-2}$. Acquisition magnification was ×53,000, rendering a pixel size of 2.73 Å.

**Tomogram reconstruction and subtomogram averaging**
**Preprocessing and contrast transfer function estimation and correction.** The preprocessing of the data consisted of two steps: motion correction and dose weighting. Each movie was individually motion corrected using MotionCor2 (ref. 51). Each resulting micrograph was then weighted by the cumulative radiation dose using a MATLAB (MathWorks) implementation of the algorithm from ref. 52. The value used for the accumulated dose per micrograph was taken from the last frame of the movie of the corresponding tilt and was then reduced by 20% to be more conservative. The contrast transfer function (CTF) was estimated using CTFFIND4 (ref. 53), and the CTF correction was done using the ctfphaseflip command from IMOD[54,55]. The CTF at each tilt was estimated prior to dose weighting. The estimation was done using only the micrograph area that corresponds to a projection of the imaged area at the zero tilt. This strategy was used in ref. 56 and proved to lead to more precise and robust defocus estimations. The estimated defocus and astigmatism were later used to correct the aligned tilt series by phase flipping. To ensure the correct hand of the tomograms (that is, the correct sign of the tilt angles), we estimated the defocuses of the left and right sides of each micrograph of a tilt series. With this, it was determined whether the left or right side of a tilt is closer to focus. This allowed us to set the correct sign of the tilt angle based on the conventions of the used processing software. A set of customizable MATLAB scripts and functions were written to automate all of the preprocessing steps, the CTF estimation and the setup of the data structure. The scripts can be found in the GitHub repository at https://github.com/C-CINA/TomographyTools.

**Tilt series alignment and tomogram reconstruction.** Tilt series alignment and tomogram reconstruction were done using IMOD. For the fiducial model, we used 6–30 gold beads per micrograph. The alignment parameter options were set to 'group rotation', 'group magnifications' and 'fixed tilt angles'. This parameter choice gave the best alignment quality while still being conservative enough to avoid overfitting. The alignment quality was assessed by examining the symmetry of the gold beads in the 3D reconstruction and by the mean residual error, which was between 0.5 pixels and 1 pixel throughout the whole dataset.

Two types of tomograms were reconstructed: binned by factor 2 tomograms using the fake SIRT-like filter from IMOD (equivalent to 50 iterations), and full-sized tomograms using the default weighted back projection parameters. For simplicity, these two tomogram types were called SIRT tomograms and WBP tomograms, respectively. The SIRT tomograms were mainly used for particle localization and the first rough subtomogram alignments. The rest of the subtomogram averaging was performed using the WBP tomograms. During some of the alignment steps using the WBP tomograms, the subvolumes themselves were binned further down.

**Subtomogram averaging.** Using the Dynamo software for subtomogram averaging[57,58], all intact tubes with a minimum length of ~160 nm and no extreme curvatures were manually traced along their center in all tomograms. The coordinates were saved in a database for further processing. In total, 180 tubes were used for further processing. The average tube length was ~460 nm, and the maximum length was ~1,500 nm (exact values: mean, 466 ± 224 nm; minimum, 157 nm; maximum, 1,552 nm). All of the tubes were accommodated within the *xy* plane of the ice. The tubes showed no preferred orientations. To locate the initial particle coordinates, we performed the following steps for each tube individually. (1) Tube average: based on the previously defined tube coordinates, the tubes were oversampled by extracting subtomograms (box size, $224^3$ pixels) every 36 pixels along the tube using the WBP tomograms. The size of the subtomograms was large enough to contain a full segment of the tube. The particles were aligned (initial reference, raw average of particles; iterations, 6; lowpass range, 60–40 Å) and averaged. The resulting tube average revealed helices around the tubes with a different number of helical starts depending on the tubes. (2) Subboxing: to extract smaller subvolumes along the helix of the tube average (subboxing), the helix first had to be parametrized. This was done by computing the radius, the helical pitch and the number of helical starts. These parameters were computed automatically using a set of self-written MATLAB scripts and functions. Subboxing was then performed using the SIRT tomogram by extracting subvolumes with a size of $64^3$ pixels every 16 pixels along the helix. The azimuth angle of each subvolume was corrected by the value of the helical pitch to force the *x* axis of the subvolumes to point in the direction of the helix. Alignment and averaging of the new subvolumes (starting reference, C2 results from one tube; iterations, 6; lowpass range, 50–30 Å) using a mask that excluded laterally neighboring particles led to an average that included three clearly visible particles. (3) Individual subboxing: the coordinates of the individual particles were determined by further subboxing the previously obtained average. The coordinates of the three visible particles for each tube were manually marked. Obtained coordinates with similar positions (within a range of 8 pixels) were reduced to the coordinate with the highest cross-correlation value with the reference of the previous alignment. Using these new coordinates, the particles were extracted into smaller subvolumes (box size, $48^3$ pixels) from the SIRT tomogram, aligned and averaged (initial reference, C2 result from one tube; mask, excluding lateral neighbors; iterations, 13; lowpass range, 50–24 Å). (4) Outlier exclusion: subvolumes that fulfilled at least one of the following criteria were excluded from further processing: (a) extreme radius: subvolume coordinate too close or too far from the tube center; (b) extreme angle: normal vector of subvolume differs too much from normal vector of tube surface; (c) missing neighbor: subvolume has no neighboring particles on either of its tips; and (d) low cross-correlation: subvolume has a low cross-correlation to the reference. Coordinates with similar positions (within a range of 17 pixels) were then again reduced to the coordinate with the highest cross-correlation value with the reference of the previous alignment. After outlier exclusion, a total of 77,436 particles were left for further processing. The complete outlier exclusion was automated using self-written MATLAB functions, reducing the initial set of 77,436 particles down to a total of 15,116. The resulting final coordinates were considered as trusted particles and were used for further processing. All final tube averages were aligned to one C2 low-resolution reference, and the coordinates of the corresponding particles were adjusted accordingly. This recentering was done to ensure that the particles across all tubes shared the same center. The final coordinates were used to extract subvolumes from the WBP tomograms with a box size of $96^3$ pixels.

## Model building and fitting into cryo-electron microscopy maps
The structure of the full-length heterodimer was built into the density using Coot. First, crystal structures of the SNX1$^{BAR}$–SNX5$^{BAR}$ domains

(PDB 8A1G, present work), SNX1$^{PX}$ domain (PDB 2I4K), SNX5$^{PX}$ domain in complex with the CI-MPR peptide (PDB 6N5Y) and the linker regions derived from the AlphaFold model were manually fitted into the density map. The amphipathic helix regions in SNX1 (amino acids 168–206) and in SNX5 (amino acids 271–306) were regularized in Coot. Once the amphipathic helix regions exhibited proper geometry, they were idealized in Phenix using the geometry minimization protocol. Finally, the whole SNX1–SNX5 composite structure was refined with the phenix. real_space tool implemented in Phenix using rigid-body and morphing with secondary-structure restraints.

## Cell culture and transfection
HT1080 (American Type Culture Collection (ATCC), CCL-121) and HEK293T (ATCC, CRL-3216) cells were maintained in Dulbecco's Modified Eagle's Medium (DMEM) containing 4 mM L-glutamine and 1.5 g l$^{-1}$ sodium bicarbonate (Quality Biological, 112-319-101) supplemented with 10% fetal bovine serum (Corning, 35-011-CV), 50 U ml$^{-1}$ penicillin and 50 µg ml$^{-1}$ streptomycin (Corning, 30002-CL) and incubated in 5% $CO_2$ and 37 °C. Lipofectamine 2000 (Thermo Fisher, 11668019) was used for transfections according to the manufacturer's protocol. Cells were fixed or imaged ~24 h after transfection. Recombinant proteins were expressed in *E. coli* BL21 (DE3) or Rosetta (DE3) cells.

## CRISPR-Cas9 knockout
Double SNX1-2 KO and double SNX5-6 KO HT1080 cells were generated using CRISPR-Cas9 (ref. 59). The targeting sequences for human SNX1 (5′-GGCCGGGGGATCAGAACCCG-3′), human SNX2 (5′-CCGTGATCTTTGATAGATCC/TCCAGATCCTCAAAGTCGGT-3′), human SNX5 (5′-GCTCTGAAACGTGGGCAGTG/AGAAACTGGG AGAAGGTGAA-3′) and human SNX6 (5′-GATGTGCTGCCACACGAC AC-3′)[7] were cloned separately into pSpCas9 (BB)-2A-GFP plasmid (Addgene, 48138, deposited by F. Zhang, Massachusetts Institute of Technology). HT1080 cells were transfected with three plasmids containing the different targeting sequences for the same gene. GFP-positive cells were isolated by flow cytofluorometry after 24 h and single-cell cloned in 96-well plates. Knockout of each clone was confirmed by immunoblotting using specific antibodies.

## Retroviral transduction
To generate SNX1 rescue and SNX5 rescue HT1080 cells, retrovirus particles were prepared by transfecting HEK293T cells with pQCXIP-HA-SNX1 (WT, BT*, SAH* or BT-SAH*) or pQCXIP-HA-SNX5 (WT, BT*, SAH* or BT-SAH*) and retrovirus-packaging plasmids pCMV-Gag-Pol (Cell Biolabs, RV-111) and pCMV-VSV-G (Cell Biolabs, RV-110) using Lipofectamine 3000 (Thermo Fisher Scientific, L3000015) according to the manufacturer's instructions. Medium was collected 24 h after transfection and centrifuged for 10 min at 1,000 × *g* to remove debris. The double SNX1-2 KO or double SNX5-6 KO HT1080 cells were immediately infected with the corresponding virus, and stably transduced cells were selected with 2 µg ml$^{-1}$ puromycin.

## Antibodies
The following primary antibodies were used for immunoblotting and/ or immunofluorescence microscopy: rabbit anti-SNX1 (Atlas Antibodies, HPA047373), rabbit anti-SNX2 (Atlas Antibodies, HPA037400), rabbit anti-SNX5 (Abcam, ab180520), rabbit anti-SNX6 (Atlas Antibodies, HPA049374), rabbit anti-EEA1 (Cell Signaling Technology, C45B10), mouse anti-CI-MPR (Abcam, 2G11), chicken anti-GFP (Thermo Fisher Scientific, A10262), mouse HRP-conjugated anti-α-tubulin (Santa Cruz Biotechnology, DM1A), rat anti-HA epitope (Roche, 3F10). HRP-conjugated goat anti-rabbit IgG (Jackson ImmunoResearch, AB_2313567), Alexa Fluor 594 donkey anti-rat IgG (Thermo Fisher Scientific, A21209), Alexa Fluor 555 donkey anti-mouse IgG (Thermo Fisher Scientific, A31570), Alexa Fluor 488 donkey anti-rabbit IgG (Thermo

Fisher Scientific, A21206) and Alexa Fluor 488 goat anti-chicken IgG (Thermo Fisher Scientific, A11039).

## Immunoblotting

Cells were trypsinized and collected into 1.5-ml tubes, washed with PBS once and lysed in 1% Triton X-100, 50 mM Tris-HCl pH 7.4, 150 mM NaCl and cOmplete EDTA-free Protease Inhibitor (Roche, 11873580001). Laemmli SDS–PAGE sample buffer (Bio-Rad, 161-0747) containing 2.5% 2-mercaptoethanol was added to the lysate and incubated for 5 min at 98 °C. Immunoblotting was performed using SDS–PAGE separation and subsequent transfer to nitrocellulose membranes. Membranes were blocked for 0.5–1 h with 3% nonfat milk (Bio-Rad, 1706404) in TBS-T (TBS supplemented with 0.05% Tween 20; Sigma-Aldrich, P9416-100ML) before being incubated overnight with primary antibody diluted in TBS-T with 3% nonfat milk. Membranes were washed three times for 20 min in TBS-T and incubated for 2–3 h in HRP-conjugated secondary antibody (1:5,000) diluted in TBS-T with 3% nonfat milk. Membranes were washed three times in TBS-T and visualized using Clarity ECL western blot substrate (Bio-Rad, 1705061).

## Fluorescence microscopy

For live-cell imaging, HT1080 cells were seeded on eight-well imaging chambers (Cellvis, C8-1.5H-N) precoated with collagen (Gibco, A1064401) 24 h prior to transfection.

Live-cell imaging was performed in a controlled-environment chamber (37 °C and 5% $CO_2$). Images were acquired on a Zeiss LSM780 or Zeiss LSM880 inverted confocal laser scanning microscope fitted with a Plan-Apochromat 63× (NA = 1.4) objective (Zeiss). z-stacks were obtained, and maximal intensity projections were generated. Microcopy images were acquired with Zeiss ZEN Black v.2.3 software. Images were further processed in ImageJ/Fiji (https://fiji.sc).

For immunofluorescence microscopy, cells were plated onto cover glasses, transfected with Lipofectamine 2000 according to the manufacturer's instructions and fixed with 4% paraformaldehyde in PBS. Cells were then permeabilized and blocked at the same time with 0.1% saponin (Sigma-Aldrich) and 1% BSA in PBS for 30 min at 25 °C. Primary antibodies were diluted in the same buffer and incubated on cells for 1.5 h at room temperature. Alexa Fluor secondary antibodies were diluted in the same buffer containing 4,6-diamidino-2-phenylindole (DAPI), and cells were incubated for 1 h at room temperature. The coverslips were mounted on glass slides using ProLong Gold Antifade (Thermo Fisher Scientific, P36934), and the cells were imaged on a confocal microscope (Zeiss LSM710 or Zeiss LSM880) with an oil-immersion 63×/1.40 NA Plan-Apochromat Oil DIC M27 objective lens (Zeiss). Image settings (that is, gain, laser power and pinhole) were kept constant for comparison. Images were acquired using Zeiss ZEN Black v.2.3 and processed with Fiji, including brightness adjustment, contrast adjustment, channel merging and cropping.

## Quantification of SNX membrane association

To estimate the efficiency of SNX recruitment to punctate intracellular membranes, we used the Find Maxima function of ImageJ/Fiji. Maximum intensity projections of z-stack images acquired from GFP-SNX1-WT-expressing or GFP-SNX5-WT-expressing HT1080 cells were used to set tolerance values for the Find Maxima function, and similarly applied to mutant constructs within each dataset. The number of local maxima for each cell was normalized to the average number of local maxima in GFP-SNX1-WT-expressing or GFP-SNX5-WT-expressing HT1080 cells and compared between mutant constructs. Fewer local maxima were identified in cells with increased cytosolic GFP signal.

## Colocalization analysis

To measure the Pearson's correlation coefficient (PCC) between green and red channels, we used the PSC colocalization plug-in of ImageJ/Fiji. Maximum intensity projections of z-stack images were acquired from each condition, and each cell was selected by drawing a region of interest encircling the surface of each cell with the selection brush tool. Then, the PCC was measured for each cell.

## Quantification and statistical analysis

Quantification of fluorescence microscopy data is presented as Super-Plots[60]. Statistical significance was calculated by one-way analysis of variance (ANOVA), followed by multiple comparisons using Dunnett's or Tukey's test (GraphPad Prism v.9 for macOS). All graphs were drawn using Prism v.9, and P values are indicated in each graph. Statistical analyses were performed using Prism.

## Reporting summary

Further information on research design is available in the Nature Portfolio Reporting Summary linked to this article.

## Data availability

Atomic coordinates and structure factors of the crystallographic complexes are available in the Protein Data Bank (PDB) under accession codes 8A1G and 8ABQ (Table 1). Cryo-ET structures and representative tomograms have been deposited in the Electron Microscopy Data Bank (EMDB) under accession code EMD-15413, and the associated PDB accession code 8AFZ (Table 2). Dose-weighted tilt series are available in the Electron Microscopy Public Image Archive (EMPIAR) under accession code 11484. Additional data that support the findings of this study are available from the corresponding authors upon request. The MATLAB scripts used to compute the neighborhood analysis have been implemented in Dynamo v.4.9 (freely available for download at http://dynamo-em.org), and its functionalities can be accessed through the command dpktbl.neighborhood.analize. Source data are provided with this paper.

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

## Acknowledgements

We thank A. Marina and M. Martinez from the Hierro Lab for technical assistance in protein purification. We also thank R. Coray from the Castaño-Díez Lab for assistance in tomographic data analysis.

This work was funded by MCIN/AEI/10.13039/501100011033 (PID2020-119132GB-I00, CEX2021-001136-S) (to A.H.), the Intramural Program of NICHD, NIH (ZIA HD001607 to J.S.B.), the Swiss National Science Foundation grant 205321 179041 (to D.C.-D.), the Human Frontiers Science Program grant RGP0017/2020 (to D.C.-D.) and the PID2021-127309NB-I00 funded by AEI/10.13039/501100011033/ FEDER, UE (to D.C.-D.). This study made use of the Diamond Light Source proposal MX20113, ALBA synchrotron beamline BL13-XALOC, the cryo-EM facilities at the UK Electron Bio-Imaging Centre, proposals EM17171-6 and EM17171, and the Midlands Regional Cryo-EM Facility at the Leicester Institute of Structural and Chemical Biology (LISCB). We thank C. Savva (LISCB, University of Leicester) for his help in cryo-EM data collection.

## Author contributions

C.L.-R. performed cloning, purification, crystallization, structure solution, cryo-EM and cryo-TEM; S.S. performed cryo-TEM, subtomogram averaging and structure analysis; E.N.A.-S. performed cloning, purification, tubulation and coflotation assays; M.I. and C.D.W. performed cellular studies; S.B.-M. and D.G.-C. performed cryo-EM and cryo-TEM imaging; M.R. and J.F.-R. performed computational analysis of binding free energies; A.V. performed cloning, purification and ITC analysis; and A.L.R. performed X-ray crystal structure solution, cryo-EM and cryo-TEM. J.S.B., D.C.-D. and A.H. designed the research, analyzed the data and wrote the manuscript with help from E.N.A.-S. and S.S.

## Competing interests

The authors declare no competing interests.

## Additional information

**Extended data** is available for this paper at https://doi.org/10.1038/s41594-023-01014-7.

**Correspondence and requests for materials** should be addressed to Juan S. Bonifacino, Daniel Castaño-Díez or Aitor Hierro.

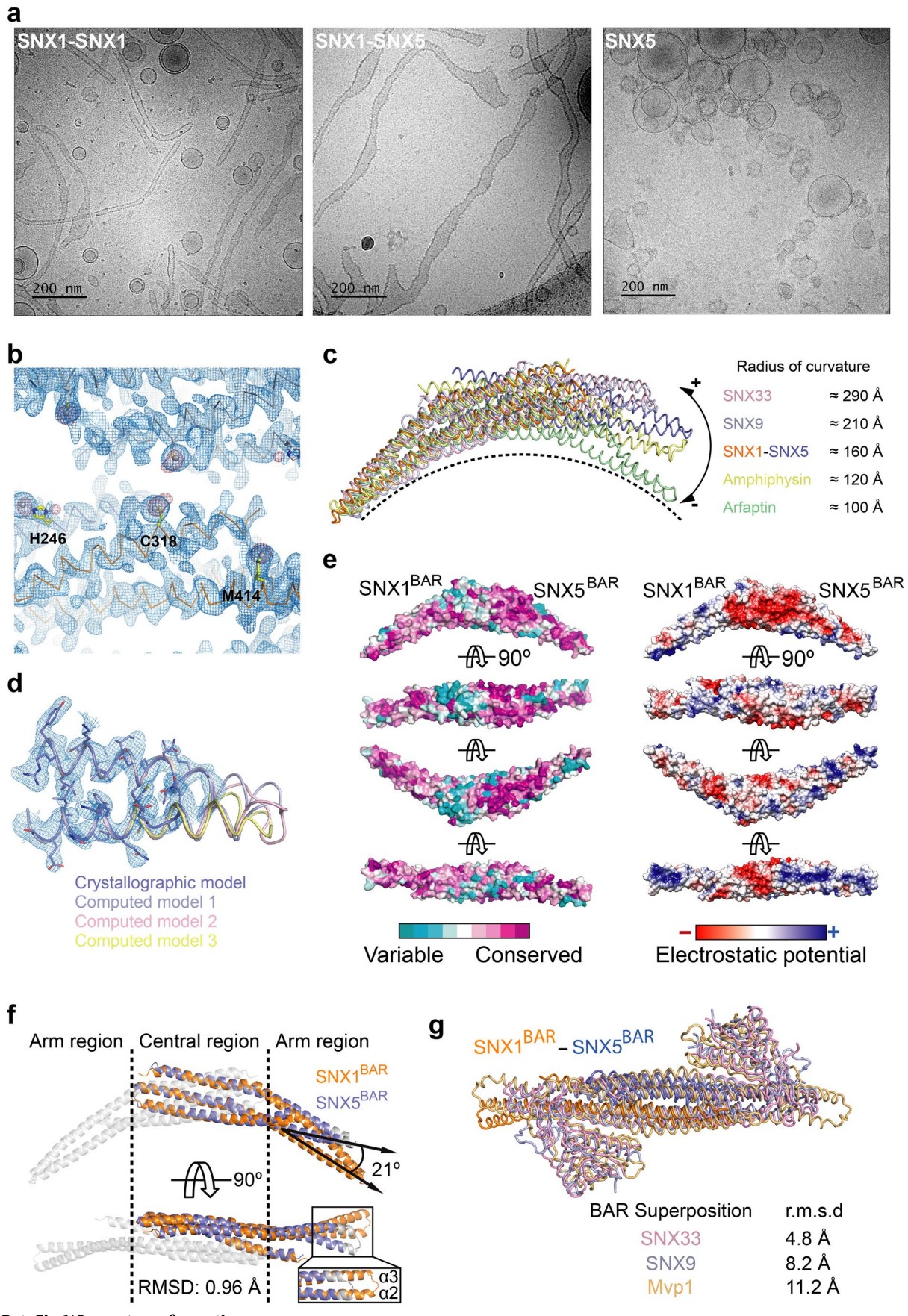

**Extended Data Fig. 1 | See next page for caption.**

**Extended Data Fig. 1 | Structural comparison of BAR domains and PX-BAR proteins.** (**a**) Liposome tubulation ability. Representative cryo-TEM images of liposomes prepared with a defined lipid composition of DOPC/DOPE/DOPS/PtdIns(3)P (45:30:20:5 molar ratio) and incubated with SNX1 homodimers, SNX1-SNX5 heterodimer or monomeric SNX5. (**b**) MAD density map (blue) contoured at 1.5σ and Pt anomalous difference map (magenta) contoured at 4.0σ superimposed on the refined structure. Sidechains of H246, C318 and M414 are highlighted in yellow as examples of platinum binders. (**c**) Comparison of the curvature of SNX1$^{BAR}$-SNX5$^{BAR}$ heterodimer with other BAR domains. To evidence differences in curvature, the structures were compared by superimposing SNX1$^{BAR}$ with one subunit from each dimer. (**d**) 2Fo-Fc electron density map (contour 1.0 σ) at the tip of the SNX5$^{BAR}$ domain. The main chain is shown as a tube (slate color) and side chains are shown as sticks. Predicted structures by the DaReUS-Loop web server are superimposed over the crystal structure. Model 1 represents the structure with the lowest statistical potential as determined by KORP[61]. (**e**) Left side ConSurf analysis[62] showing surface conservation of amino-acid residues within the heterodimeric SNX1$^{BAR}$-SNX5$^{BAR}$. Right side illustrates electrostatic surface potential viewed in the same orientations as in the left side. The scale ranges from −5 kT e$^{-1}$ (red) to 5 kT e$^{-1}$ (blue). (**f**) SNX1$^{BAR}$ superposed with SNX5$^{BAR}$ through the central region highlighting the structural variations between the distal arms. (**g**) Superposition of known PX-BAR structures (SNX33, PDB 4AKV [to be published]; SNX9, PDB 2RAI[63]; Mvp1, PDB 6Q0X[64] over the SNX1$^{BAR}$-SNX5$^{BAR}$ heterodimer. Data in **a** are representative of three independent experiments.

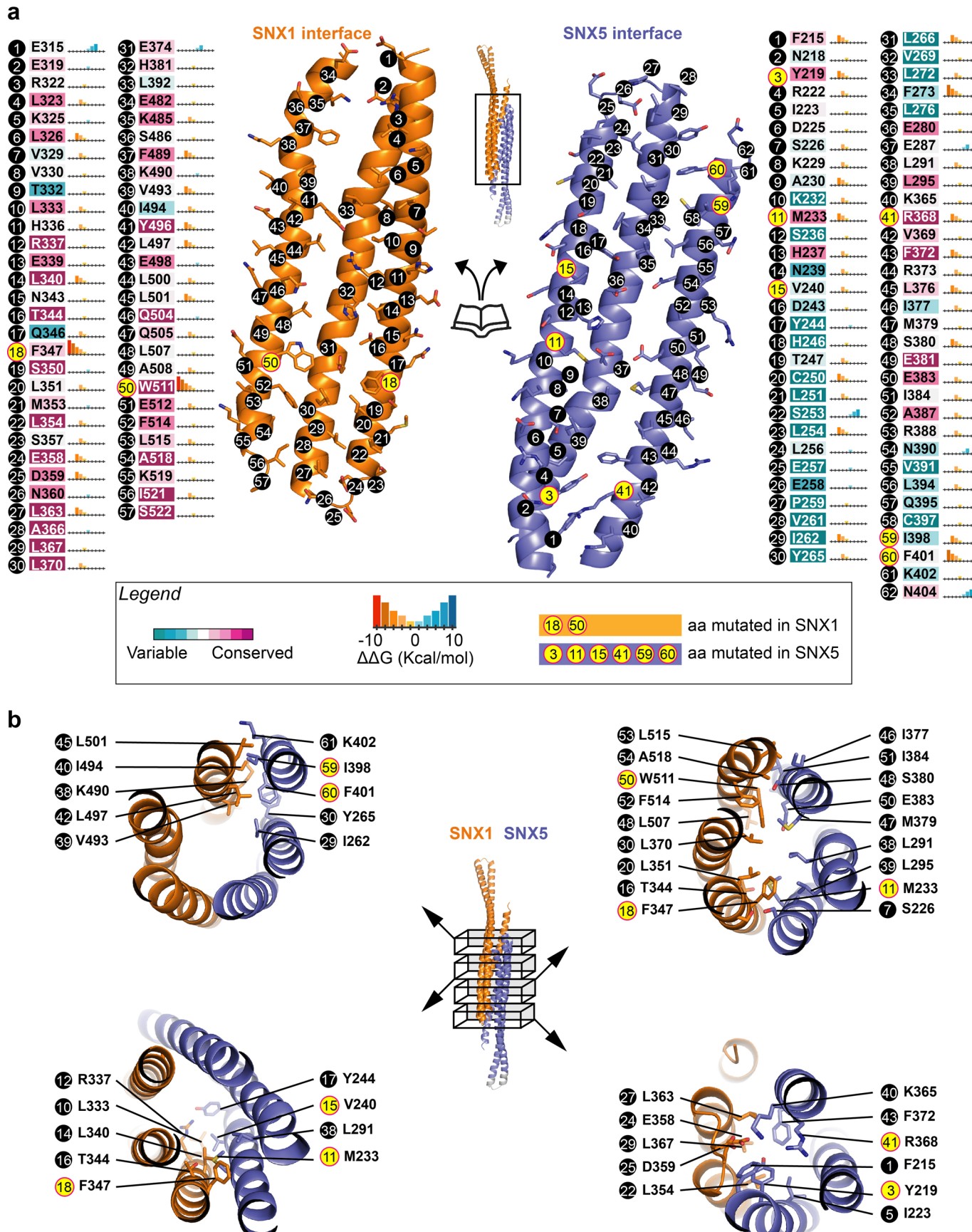

**Extended Data Fig. 2 | Interface characterization. (a)** Detailed per-residue conservation and energetic analysis of the SNX1$^{BAR}$-SNX5$^{BAR}$ interface. Mutated residues that affect dimer formation are highlighted in yellow. **(b)** Detailed view of neighboring sites of amino acids that were mutated at the proximal and central regions of the BAR domains to interfere with dimerization.

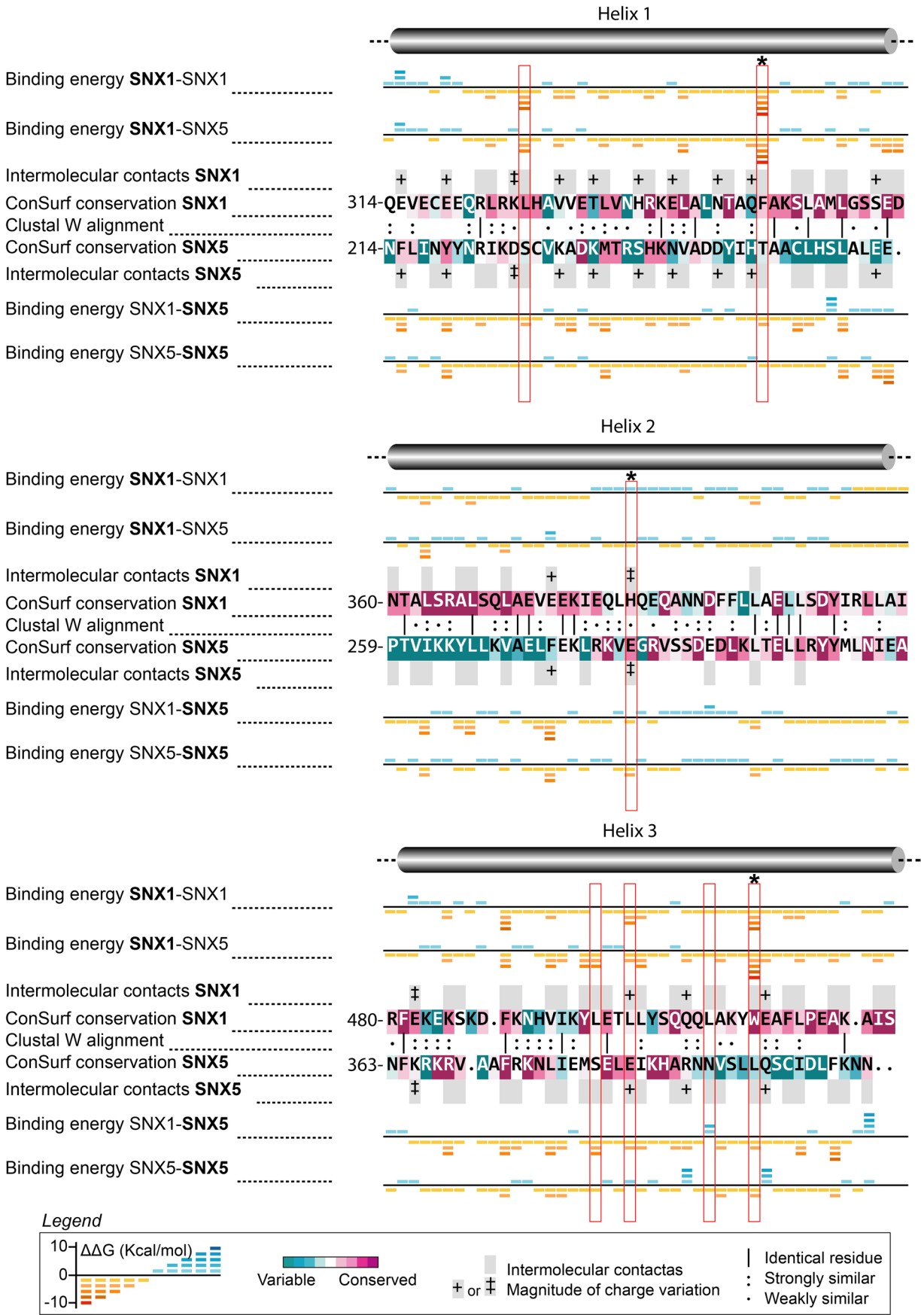

**Extended Data Fig. 3 | See next page for caption.**

**Extended Data Fig. 3 | Pairwise comparison of interfaces.** Alignment generated from the structural superposition of SNX1BAR and SNX5BAR central regions. Alignments also include per-residue energetic contribution in theoretical SNX1 and SNX5 homodimers generated by homology modeling using our SNX1BAR-SNX5BAR crystal structure as template. Energetic values in each row correspond to the molecule highlighted in bold within the respective complex. Red boxes mark residues that were mutated to interfere with dimerization. Red boxes with an asterisk indicate residues that were mutated to promote SNX5 dimerization.

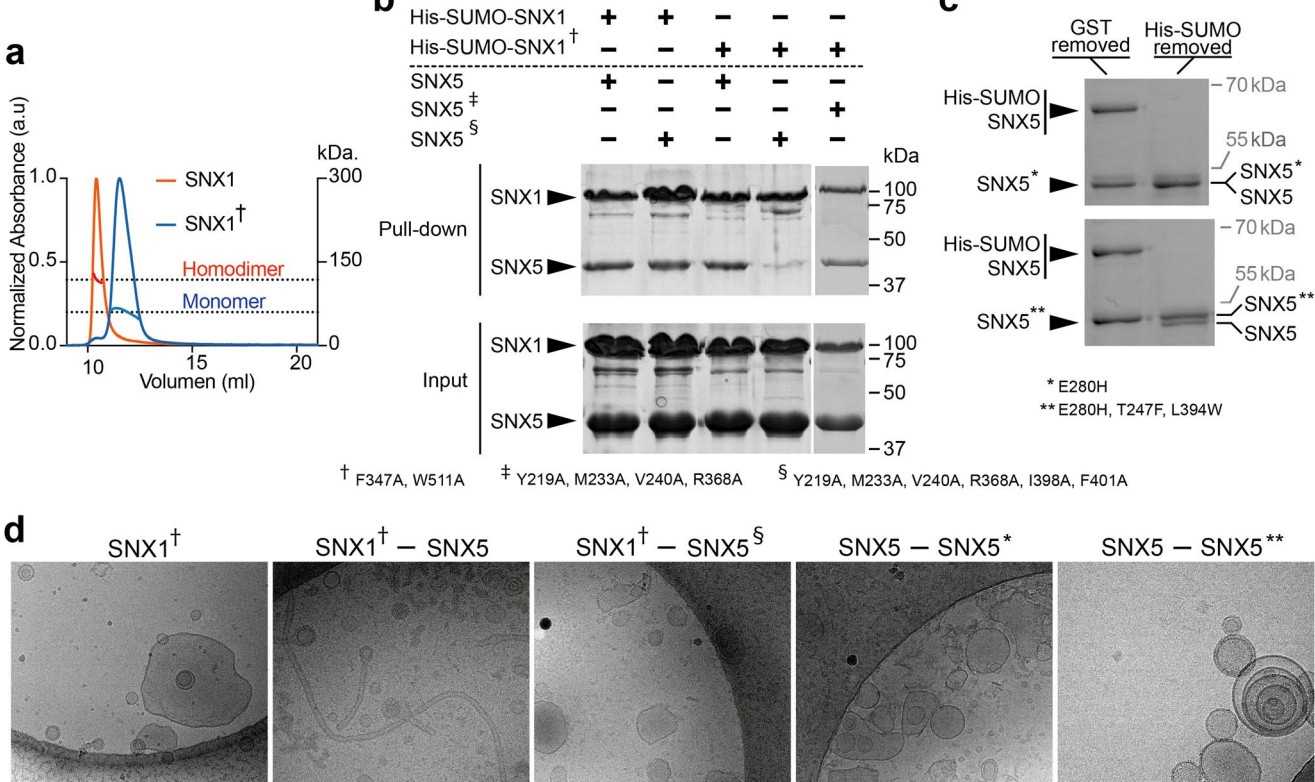

**Extended Data Fig. 4 | SNX1^BAR-SNX5^BAR interface validation.** (**a**) SEC-MALS analysis of F347A, W511A double mutant on the SNX1 (SNX1†) interface that precludes its homodimerization. (**b**) Affinity pull down assay for mutants influencing SNX1 heterodimerization. (**c**) Tandem affinity purification of SNX5 induced homodimers. SNX5^WT was tagged with His-SUMO, and single (*) or triple (**) SNX5 mutants were tagged with GST. The complex was purified from combined cell lysates. Left lane shows the dimer with the His-SUMO tag still on SNX5. (**d**) Liposome tubulation ability of SNX1 and SNX5 proteins with mutations at interfaces affecting their dimerization capacity. Results in **a-d** are representative of two independent experiments.

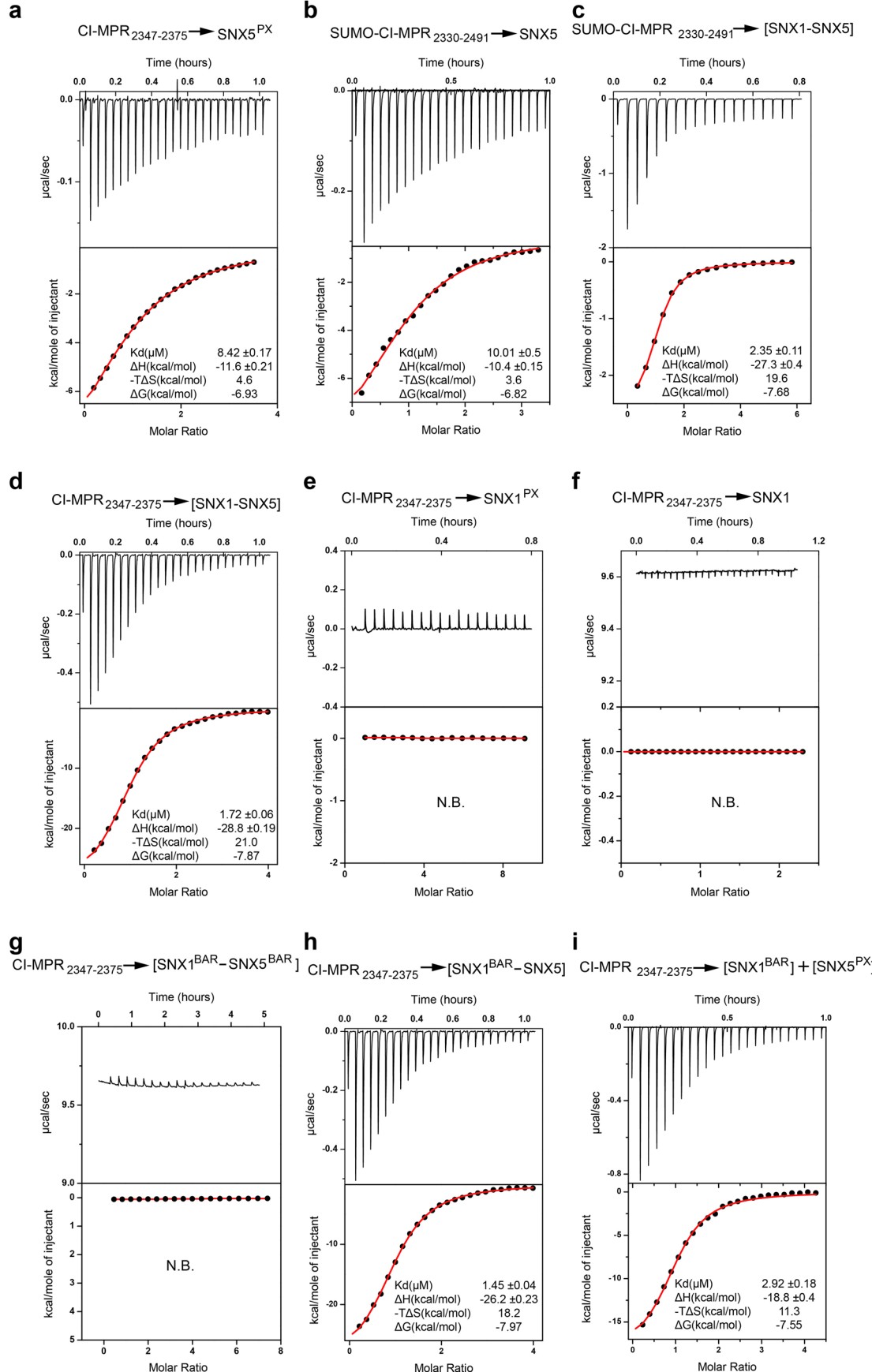

**Extended Data Fig. 5 | SNX1^BAR domain enhances the interaction between the PX domain of SNX5 and CI-MPR.** Representative ITC experiments for the binding of the cytosolic region of CI-MPR (amino acids 2330-2491) (panels **b**, **c**), or the bipartite sorting motif (amino acids 2347-2375) (panels **a**, **d-i**) titrated into ESCPE-1 or selected subdomains. Top panels show the raw data and bottom panels represent the integrated and normalized data fit with a 1:1 binding model. Data are representative of at least two independent experiments.

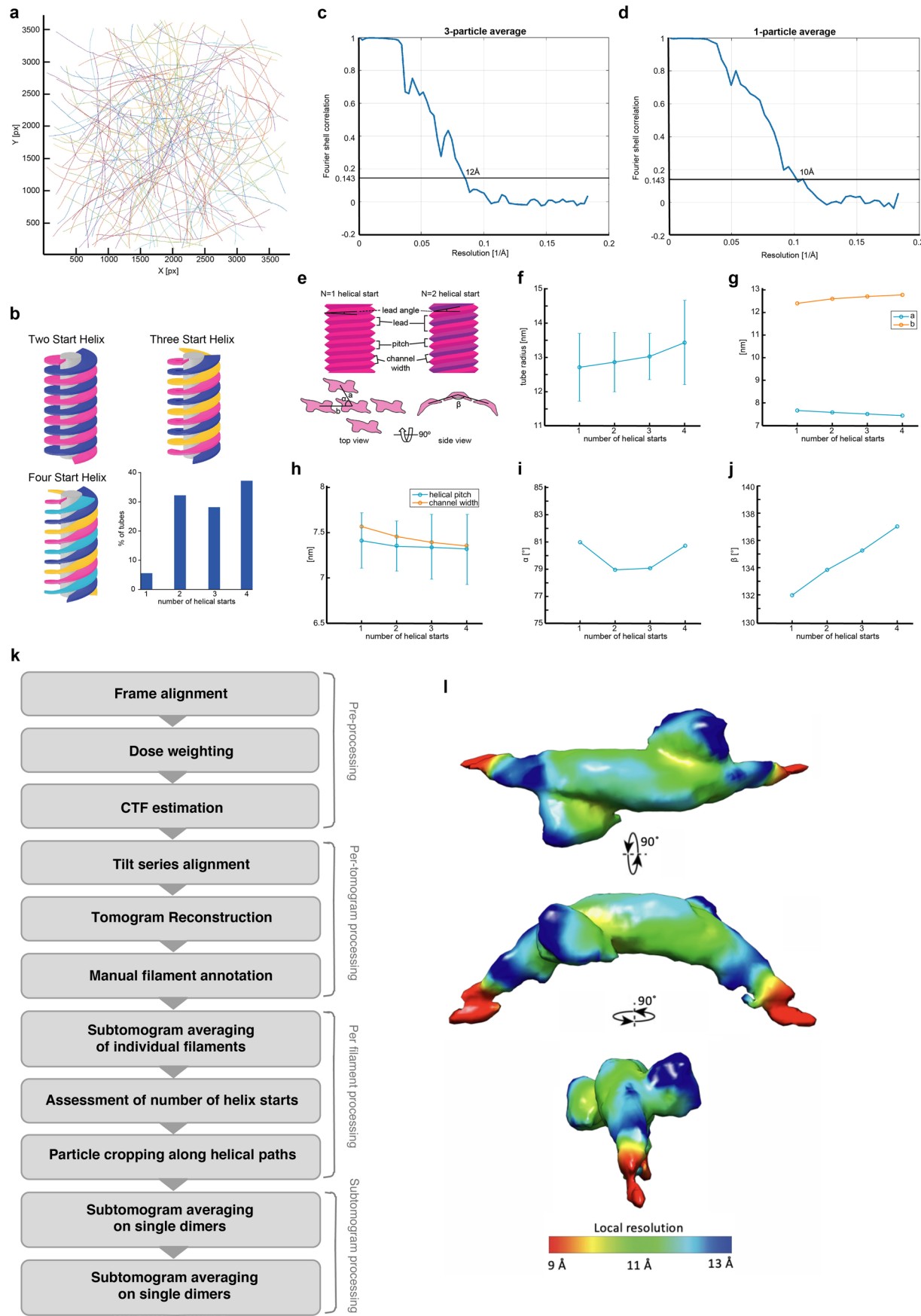

**Extended Data Fig. 6 | See next page for caption.**

**Extended Data Fig. 6 | Lattice geometry and cryo-ET data processing workflow.** (**a**) Overlap of projections of coordinates from all selected tubes. No preferred orientation can be seen. (**b**) Distribution of number of helical starts of all analyzed tubes. (**c**) Fourier shell correlation (FSC) curve for the final 3-particle map. (**d**) Fourier shell correlation (FSC) curve for the final 1-particle map. (**e**) Definition of the analyzed helix parameters (pitch, lead, lead angle, channel width) and relations to neighboring particles (a, b, alpha, beta). (**f**) Tube radius (mean +/− SD) related to number of helical starts N. Mean and standard deviation have been computed on the available number of filaments detected for each N: 10 filaments for N = 1, 57 filaments for N = 2, 49 filaments for N = 3 and 65 filaments for N = 4. (**g**) Distances between the centers of the particles related to N. (**h**) Helical pitch and channel width related to N. Helical pitch is presented as mean values +/− SD. (**i**) Angle alpha related to N. (**j**) Angle beta related to N. (**k**) Cryo-ET data processing workflow. (**l**) Average of subtomograms from individual particles displaying local resolution (Å) coloured from highest resolution (red) to lowest resolution (dark blue).

**a**

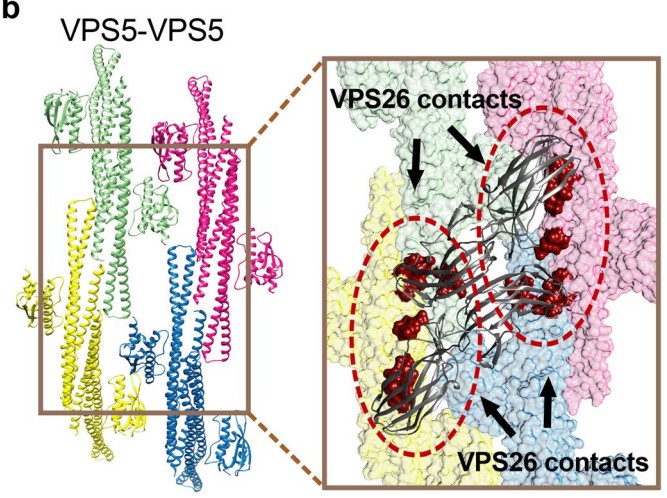

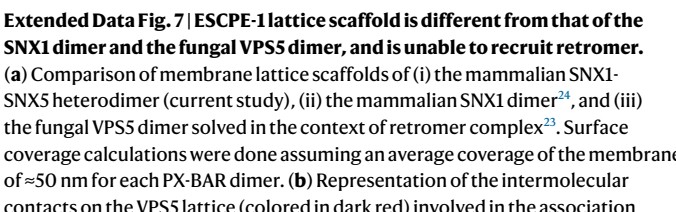

| | Mammals | | Fungi |
|---|---|---|---|
| | SNX1-SNX5 | SNX1-SNX1 | VPS5-VPS5 |
| Areal density | ≈10,368 molecules/µm² | ≈12,219 molecules/µm² | ≈16,526 molecules/µm² |
| Surface coverage | ≈52% | ≈61% | ≈82% |

**b** VPS5-VPS5

**c**

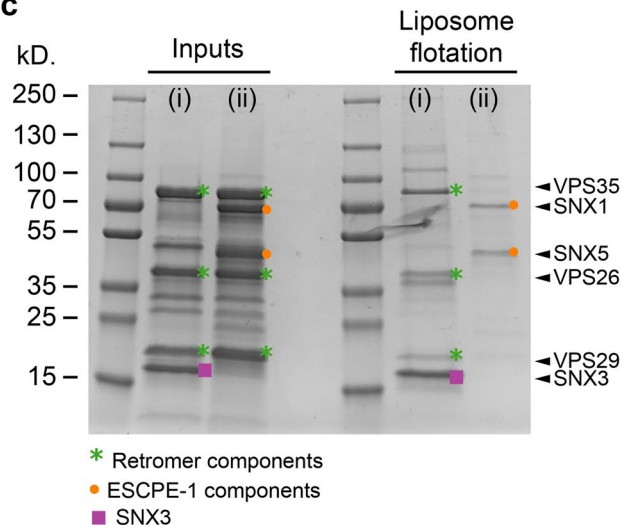

\* Retromer components
● ESCPE-1 components
■ SNX3

**Extended Data Fig. 7 | ESCPE-1 lattice scaffold is different from that of the SNX1 dimer and the fungal VPS5 dimer, and is unable to recruit retromer.** (**a**) Comparison of membrane lattice scaffolds of (i) the mammalian SNX1-SNX5 heterodimer (current study), (ii) the mammalian SNX1 dimer[24], and (iii) the fungal VPS5 dimer solved in the context of retromer complex[23]. Surface coverage calculations were done assuming an average coverage of the membrane of ≈50 nm for each PX-BAR dimer. (**b**) Representation of the intermolecular contacts on the VPS5 lattice (colored in dark red) involved in the association with the VPS26 subunit of the retromer complex. Note that the distribution of contacts on two adjacent BAR domains (green and yellow, or pink and blue) from separate dimers is not conserved in the SNX1 or SNX1-SNX5 lattices. (**c**) In flotation assays, (i) retromer (VPS35-VPS29-VPS26 subunits) was recruited by SNX3 and the DMT1-II$_{550-568}$ sorting motif to liposomes (DOPC/DOPE/DOPS/PtdIns(3)P/Liss Rhod-PE 45:28:20:5:2 molar ratio) whereas (ii) retromer was not recruited by SNX1-SNX5 and the CI-MPR cargo. Results in **c** are representative of three independent experiments.

**a**

| Injectant | Cell | Kd (µM) ±SD |
|---|---|---|
| CI-MPR $_{2347-2375}$ | [SNX1-SAH $^{3A}$ – SNX5-SAH$^{3A}$] | 5.06 ± 0.4 |
| CI-MPR $_{2347-2375}$ | [SNX1-BT$^{*}$– SNX5-BT$^{*}$ ] | 10.4 ± 0.52 |

**b**

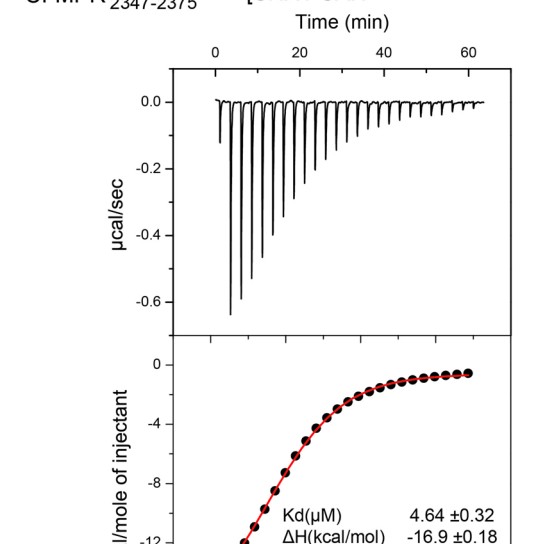

**c**

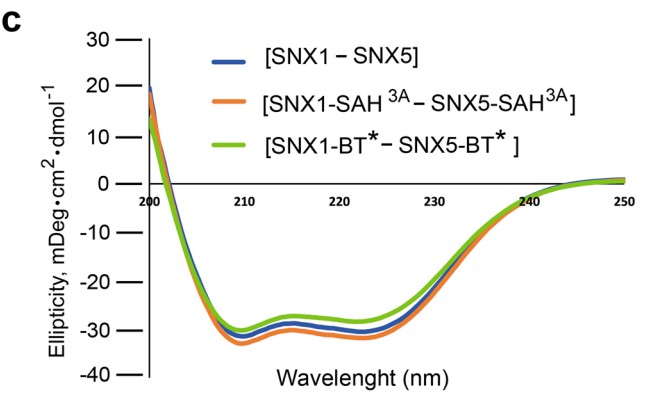

**d**

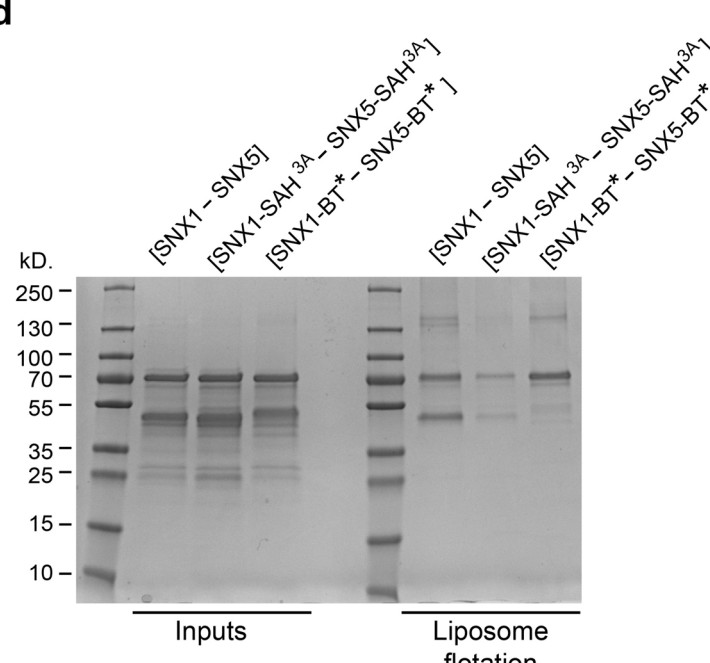

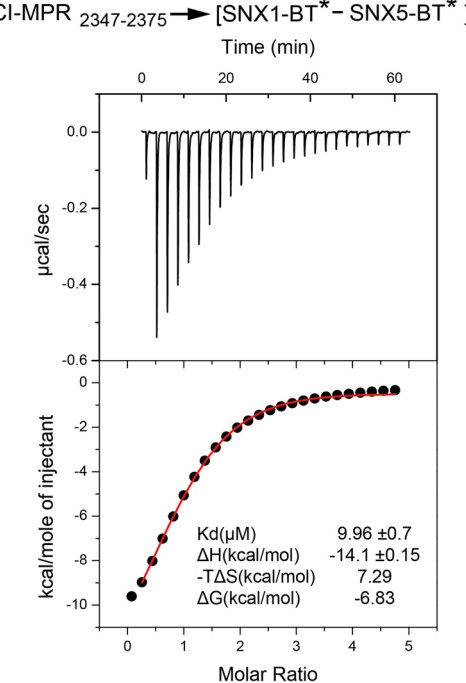

**Extended Data Fig. 8 | Mutations within the SAH regions, or within the BAR-TIP regions in ESCPE-1 affect cargo binding and interfere with membrane association. (a)** Summary of Kds between the CI-MPR bipartite sorting motif (amino acids 2347-2375) titrated into the SNX1-SAH³ᴬ:SNX5-SAH3A mutant or the SNX1- BT*:SNX5- BT* mutant. Values are the mean and standard deviation (SD) from two independent experiments. **(b)** Representative ITC experiments for the binding of the previous SAH3A and BT* mutants. Top panels show the raw data and bottom panels represent the integrated and normalized data fit with a 1:1 binding model. **(c)** Circular dichroism (CD) spectra of SNX1ᵂᵀ-SNX5ᵂᵀ, the SNX1-SAH³ᴬ:SNX5-SAH3A mutant, and the SNX1- BT*:SNX5- BT* mutant. **(d)** Liposome flotation assay of SNX1ᵂᵀ-SNX5ᵂᵀ, the SNX1-SAH³ᴬ:SNX5-SAH3A mutant, and the SNX1-BT*:SNX5-BT* mutant. The liposome composition was: DOPC/DOPE/DOPS/PtdIns(3)P/Liss Rhod-PE 45:28:20:5:2 molar ratio. Data are representative of two **(a-c)**, or three **(d)** independent experiments.

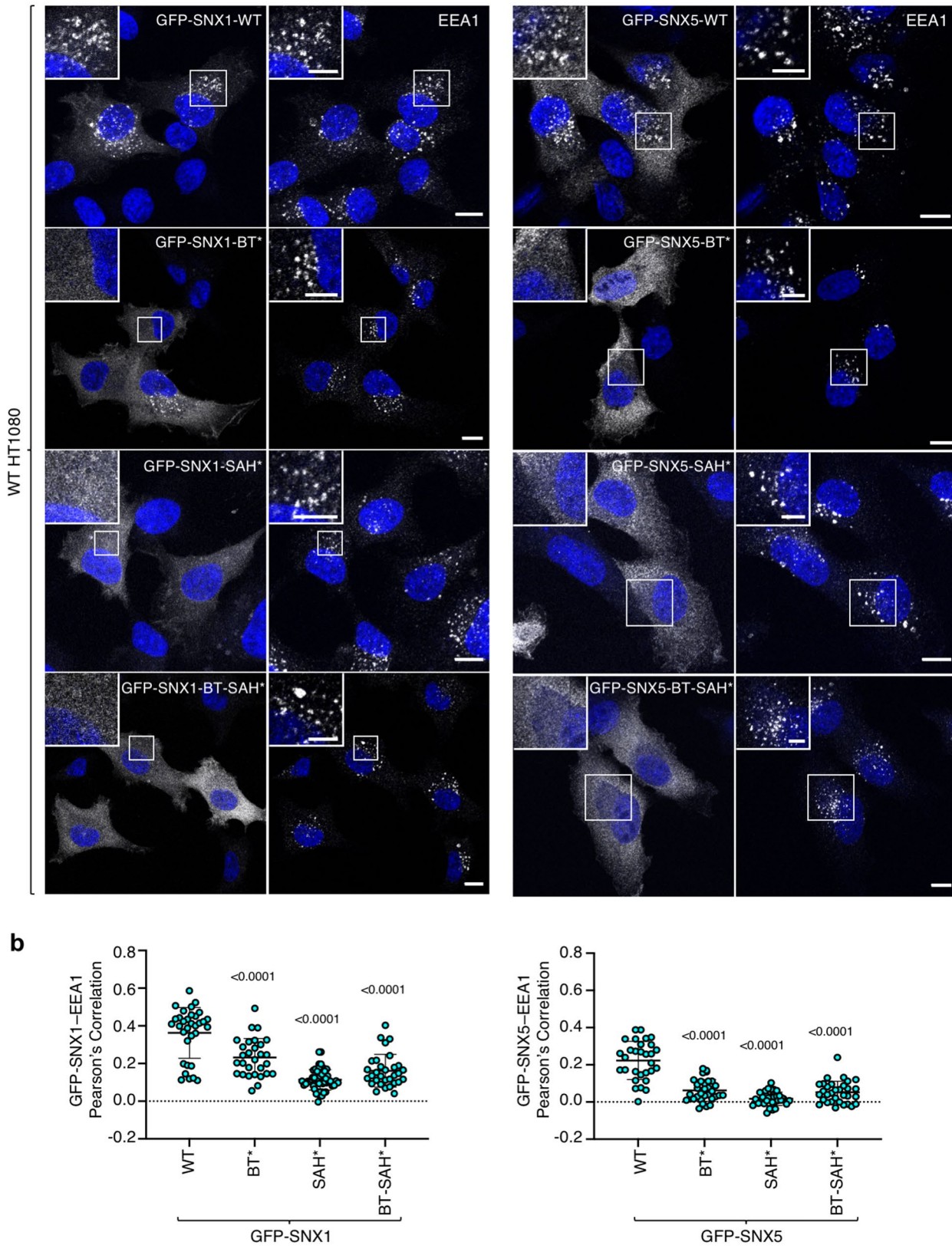

**Extended Data Fig. 9 | See next page for caption.**

**Extended Data Fig. 9 | Requirement of BAR$^{tip}$-to-BAR$^{tip}$ and BAR$^{tip}$-to-PX interactions for endosomal association of SNX1 and SNX5 in WT cells.** (**a**) Immunofluorescence microscopy of fixed-permeabilized WT HT1080 cells transiently transfected with plasmids encoding GFP-tagged WT and mutant SNX1 or SNX5 constructs, and stained for early endosomes (EEA1; red), and nuclei (DAPI; blue). Because of the low expression levels of GFP-SNX5 constructs, the GFP-SNX5 signal was enhanced by immunostaining with antibody to GFP. Scale bars: 10 μm. Insets are magnified views of the boxed areas. Scale bars: 5 μm. (**b**) Graphs showing the Pearson's correlation coefficient (PCC) between GFP-tagged proteins and EEA1 calculated from following number of cells in the experiment shown in panel A. For GFP-SNX1, n = 33 in WT, n = 29 in BT, n = 51 in SAH and n = 31 in BT-SAH. For GFP-SNX5, n = 30 in WT, n = 31 in BT, n = 33 in SAH, n = 32 in BT-SAH. The graphs show the individual data points and the mean ± SD of the data. Statistical significance was calculated by one-way ANOVA with multiple comparisons to the SNX WT control using Dunnett's test with the number of cells indicated above. *p*-values are indicated on the plots.

**Extended Data Table 1 | Glossary of protein symbols used through the text**

| Protein symbol | Description | UniProt |
|---|---|---|
| SNX1$^{FL}$ | Full length SNX1 | Q13596 |
| SNX1$^†$ | Full length SNX1 with F347A+W511A mutants | |
| SNX1$^{PX}$ | PX domain of SNX1, aa 93 - 282 | |
| SNX1$^{BAR}$ | BAR domain of SNX1, aa 301 - 522 | |
| SNX1$^{BAR†}$ | BAR domain of SNX1, aa 301 - 522, with F347A+W511A mutants | |
| SNX1-SAH$^{3A}$ | Full length SNX1 with L271A+P272A+R273A mutants in the SAH region | |
| SNX1-BT* | Full length SNX1 with L437G+L438G+W439G+A440G+N441S+K442G+P443G+D444G+K445G mutants in the BAR TIP region. | |
| SNX1-BT-SAH* | Full length SNX1 with L437G+L438G+W439G+A440G+N441S+K442G+P443G+D444G+K445G mutants in the BAR TIP region, and L270S+P271G+R272G+A273G+V274G+G275S+T276G+Q277G+T278G+L279G mutants in the SAH region | |
| SNX5$^{FL}$ | Full length SNX5 | Q9Y5X3 |
| SNX5$^{PX}$ | PX domain of SNX5, aa 1-183 | |
| SNX5$^{BAR}$ | BAR domain of SNX5, aa 195 - 404 | |
| SNX5$^‡$ | Full length SNX5 with Y219A+M233A+V240A+R368A mutants | |
| SNX5$^§$ | Full length SNX5 with Y219A+M233A+V240A+R368A+I398A+F401A mutants | |
| SNX5* | Full length SNX5 with E280H mutant | |
| SNX5** | Full length SNX5 with E280H+T247F+L394W mutant | |
| SNX5-SAH$^{3A}$ | Full length SNX5 with R175A+K177A+N178A in the SAH region | |
| SNX5-BT* | Full length SNX5 with A329G+R330G+L331G+K332G+K334G+D335G+V336G+K337G mutants in the BAR TIP region | |
| SNX5-SAH* | Full length SNX5 with V174G+R175G+R176G+K177G+N178S+T179G+K180G+E181G+M182G+F183S mutants in the SAH region | |
| SNX5-BT-SAH* | Full length SNX5 with A329G+R330G+L331G+K332G+K334G+D335G+V336G+K337G mutants in the BAR TIP region, and V174G+R175G+R176G+K177G+N178S+T179G+K180G+E181G+M182G+F183S mutants in the SAH region | |
| SNX3$^{FL}$ | Full length SNX3 | O60493 |
| VPS35 | Full length VPS35 | Q96QK1 |
| VPS29 | Full length VPS29 | Q9UBQ0 |
| VPS26 | Full length VPS26 | O75436 |
| CI-MPR$_{2330-2491}$ | cytosolic region of CI-MPR, aa 2330 – 2491, with C2342A+C2343A mutants | P11717 |
| CI-MPR$_{2347-2375}$ | CI-MPR bipartite sorting motif, aa 2347 - 2375 | P11717 |
| DMT1-II$_{550-568}$ | DMT1-II sorting motif, aa 550 - 568 | P49281 |

Aitor Hierro
Daniel Castaño-Díez
Juan S. Bonifacino

# Reporting Summary

## Statistics

For all statistical analyses, confirm that the following items are present in the figure legend, table legend, main text, or Methods section.

| n/a | Confirmed | |
|---|---|---|
| ☐ | ☒ | The exact sample size (*n*) for each experimental group/condition, given as a discrete number and unit of measurement |
| ☐ | ☒ | A statement on whether measurements were taken from distinct samples or whether the same sample was measured repeatedly |
| ☐ | ☒ | The statistical test(s) used AND whether they are one- or two-sided *Only common tests should be described solely by name; describe more complex techniques in the Methods section.* |
| ☒ | ☐ | A description of all covariates tested |
| ☒ | ☐ | A description of any assumptions or corrections, such as tests of normality and adjustment for multiple comparisons |
| ☐ | ☒ | A full description of the statistical parameters including central tendency (e.g. means) or other basic estimates (e.g. regression coefficient) AND variation (e.g. standard deviation) or associated estimates of uncertainty (e.g. confidence intervals) |
| ☐ | ☒ | For null hypothesis testing, the test statistic (e.g. *F*, *t*, *r*) with confidence intervals, effect sizes, degrees of freedom and *P* value noted *Give P values as exact values whenever suitable.* |
| ☒ | ☐ | For Bayesian analysis, information on the choice of priors and Markov chain Monte Carlo settings |
| ☒ | ☐ | For hierarchical and complex designs, identification of the appropriate level for tests and full reporting of outcomes |
| ☒ | ☐ | Estimates of effect sizes (e.g. Cohen's *d*, Pearson's *r*), indicating how they were calculated |

*Our web collection on statistics for biologists contains articles on many of the points above.*

## Software and code

Policy information about availability of computer code

| Data collection | • Crystallographic native and derivative data sets were collected with the software MxCuBE at XALOC beamline in ALBA (Cerdanyola del Valles, Spain) using a Pilatus 6M detector.<br>• High-resolution cryoET data was obtained with Serial-EM 4.0 software in an FEI Titan Krios G3 microscope, coupled with Gatan K2 Summit direct detector.<br>• For live-cell imaging, images were acquired with Zeiss ZEN Black 2.3 software on a Zeiss LSM780 or Zeiss LSM880 inverted confocal laser scanning microscope fitted with a Plan-Apochromat 63X (NA=1.4) objective (Carl Zeiss).<br>• For immunofluorescence microscopy, images were acquired using Zeiss ZEN Black 2.3 software (Carl Zeiss) on a confocal microscope (LSM710 or LSM880; Carl Zeiss) with an oil-immersion 63×/1.40 NA Plan-Apochromat Oil DIC M27 objective lens (Carl Zeiss). |
|---|---|
| Data analysis | • X-ray diffraction images were indexed, integrated and scaled using XDS (BUILT=20170923) or MOSFLM/SCALA 7.2.2. Heavy atom positions were identified using SHELXC/D as implemented in autoSHARP 3.10.8. Density modification was done using SOLOMON as implemented in autoSHARP 3.10.8. Refinement was done with PHENIX 1.18 and manual model building with COOT 0.8.9.<br>• Cryo-tomography tilt series alignment and tomogram reconstruction was done using IMOD 4.5.0. Sub-tomogram averaging was done with the Dynamo software.<br>• ITC data was fitted and integrated employing the MicroCal PEAQ-ITC software VPViewer2000, from Malvern Panalytical. Final graphs were prepared using Origin 7 ITC software (MicroCal).<br>• Binding energy per residue was estimated with resEnergy pyDock.<br>• Fluorescence microscopy images were processed in ImageJ/Fiji 1.52k (https://fiji.sc). Statistical analyses were performed using Prism Software 9.3.1 (GraphPad). |

For manuscripts utilizing custom algorithms or software that are central to the research but not yet described in published literature, software must be made available to editors and reviewers. We strongly encourage code deposition in a community repository (e.g. GitHub). See the Nature Portfolio guidelines for submitting code & software for further information.

# Data

Policy information about availability of data

All manuscripts must include a data availability statement. This statement should provide the following information, where applicable:

- Accession codes, unique identifiers, or web links for publicly available datasets
- A description of any restrictions on data availability
- For clinical datasets or third party data, please ensure that the statement adheres to our policy

Atomic coordinates and structure factors of the crystallographic complexes are available in the Protein Data Bank (PDB) with accession codes 8A1G and 8ABQ (Table 1). Cryo-ET structures and representative tomograms have been deposited in the Electron Microscopy Data Bank (EMDB) with accession code EMD-15413, and the associated PDB 8AFZ (Table 2). Dose-weighted tilt series are available in the Electron Microscopy Public Image Archive (EMPIAR) under the accession code 11484. Additional data that support the findings of this study are available from the corresponding authors on request.
The matlab scripts used to compute the neighborhood analysis have been implemented in Dynamo 4.9 (freely available for download at dynamo-em.org) and its functionalities can be accessed through the command dpktbl.neighborhood.analize.
Validation reports are included in Supplementary Information. Additional data that support the findings of this study are available from the corresponding authors on request.

# Human research participants

Policy information about studies involving human research participants and Sex and Gender in Research.

| | |
|---|---|
| Reporting on sex and gender | No human subjects involved in this study |
| Population characteristics | Not applicable |
| Recruitment | Not applicable |
| Ethics oversight | Not applicable |

Note that full information on the approval of the study protocol must also be provided in the manuscript.

# Field-specific reporting

Please select the one below that is the best fit for your research. If you are not sure, read the appropriate sections before making your selection.

☒ Life sciences      ☐ Behavioural & social sciences      ☐ Ecological, evolutionary & environmental sciences

For a reference copy of the document with all sections, see nature.com/documents/nr-reporting-summary-flat.pdf

# Life sciences study design

All studies must disclose on these points even when the disclosure is negative.

| | |
|---|---|
| Sample size | For sub-tomogram averaging, all intact tubes with a minimum length of ~160 nm and no extreme curvatures were manually traced along their center in all tomograms. In total, 180 tubes were used for further processing. Sample size was was predetermined by the microscope allocation time. Data was sufficient to achieve the reported resolutions and build the molecular models. The number of replicates for the functional assays are stated in the figure legends. |
| Data exclusions | Sub-tomogram outlier exclusion: Sub-volumes that fulfilled at least one of the following criteria were excluded from further processing: (a) Extreme radius: Sub-volume coordinate too close or too far from the tube center. (b) Extreme angle: Normal vector of sub-volume differs too much from normal vector of tube surface. (c) Missing neighbor: Sub-volume has no neighboring particles on either of its tips. (d) Low cross correlation: Sub-volume has a low cross-correlation to the reference. |
| Replication | Liposome tubulation, liposome flotation and ITC assays were repeated at least twice and all attempts at replication were successful. 'In cellulo' functional studies were repeated as indicated in the figure legends. |
| Randomization | For immunofluorescence analysis, images were randomly acquired at the confocal microscope for each set of experiments. For cryo-ET, the most promising areas for data collection were selected from low magnification images. |
| Blinding | Imaging of liposome tubulation reactions was performed by someone blind to the composition of the reactions. All light microscopy experiments have been measured by blinded personnel when required manual intervention. |

# Reporting for specific materials, systems and methods

We require information from authors about some types of materials, experimental systems and methods used in many studies. Here, indicate whether each material, system or method listed is relevant to your study. If you are not sure if a list item applies to your research, read the appropriate section before selecting a response.

## Materials & experimental systems

| n/a | Involved in the study |
|---|---|
| ☐ | ☒ Antibodies |
| ☐ | ☒ Eukaryotic cell lines |
| ☒ | ☐ Palaeontology and archaeology |
| ☒ | ☐ Animals and other organisms |
| ☒ | ☐ Clinical data |
| ☒ | ☐ Dual use research of concern |

## Methods

| n/a | Involved in the study |
|---|---|
| ☒ | ☐ ChIP-seq |
| ☒ | ☐ Flow cytometry |
| ☒ | ☐ MRI-based neuroimaging |

## Antibodies

| | |
|---|---|
| Antibodies used | The following primary antibodies were used for immunoblotting and/or immunofluorescence microscopy: rabbit anti-SNX1 (1:2,000 dilution) (Atlas Antibodies, HPA047373), rabbit anti-SNX2 (1:2,000 dilution) (Atlas Antibodies, HPA037400), rabbit anti-SNX5 (1:2,000 dilution) (Abcam, ab180520), rabbit anti-SNX6 (1:2,000 dilution) (Atlas Antibodies, HPA049374), rabbit anti-EEA1 (1:100 dilution) (Cell Signaling Technology, C45B10), mouse anti-CI-MPR (1:200 dilution) (Abcam, 2G11), chicken anti-GFP (1:500 dilution) (Thermo Fisher Scientific, A10262), mouse HRP-conjugated anti-α-tubulin (1:10,000) (Santa Cruz Biotechnology, DM1A), rat anti-HA epitope (1:10,000) (Roche, 3F10). HRP-conjugated goat anti-rabbit (1:5,000 dilution) (Jackson ImmunoResearch, AB_2313567), Alexa Fluor 594 donkey anti-rat IgG (1:1,000 dilution) (Thermo Fisher Scientific, A21209), Alexa Fluor 555 donkey anti-mouse IgG (1:1,000 dilution) (Thermo Fisher Scientific, A31570), Alexa Fluor 488 donkey anti-rabbit IgG (1:1,000 dilution) (Thermo Fisher Scientific, A21206) and Alexa Fluor 488 goat anti-chicken IgG (1:1,000 dilution) (Thermo Fisher Scientific, A11039). |
| Validation | All primary antibodies were validaded by the vendor for immunoblotting and/or immunofluorescence. The validation statement can be found on the corresponding manufacture's website. |

## Eukaryotic cell lines

Policy information about cell lines and Sex and Gender in Research

| | |
|---|---|
| Cell line source(s) | • HT1080, American Type Culture Collection CCL-121<br>• HEK293T, American Type Culture Collection CRL-3216 |
| Authentication | The expression (or lack thereof) of target proteins in WT and KO HT1080 KO cells was confirmed by immunoblotting with specific antibodies. |
| Mycoplasma contamination | The cell lines were not tested for mycoplasma contamination. |
| Commonly misidentified lines<br>(See ICLAC register) | The HT1080 cells used in this study have not been reported to be contaminated by any other cells according to https://iclac.org/databases/cross-contaminations/. No commonly misidentified cell lines were used in the study. |

