## [Peer Review File · Nature Structural & Molecular Biology]

Peer Review Information

Manuscript Title: Architecture of the ESCPE-1 membrane coat

Corresponding author name(s): Juan S. Bonifacino, Daniel Castaño-Díez, Aitor Hierro

Reviewer Comments & Decisions:

Decision Letter, initial version:

Message: 23rd Nov 2022

Dear Dr. Hierro,

Thank you again for submitting your manuscript "Architecture of the ESCPE-1 membrane coat". We now have comments (below) from the 3 reviewers who evaluated your paper. In light of those reports, we remain interested in your study and would like to see your response to the comments of the referees, in the form of a revised manuscript.

You will see that all reviewers are generally positive about the findings, however they do raise concerns which we would expect to be addressed during revision. Specifically, reviewers #1 and #2 agree that further analysis is required to clarify phosphoinositide binding. In line with reviewer #1 comments, we would expect the context of the findings in wider literature discussed more thoroughly, and replicates of experiments (eg. ITC) provided. Moreover, reviewer #2 brings up a similar issue with analysis of cargo binding affinity. Reviewer #3 points out minor points, most of which relate to methodology reporting.

Please be sure to address/respond to all concerns of the referees in full in a point-by-point response and highlight all changes in the revised manuscript text file. If you have comments that are intended for editors only, please include those in a separate cover letter.

We expect to see your revised manuscript within 6 weeks. If you cannot send it within this time, please contact us to discuss an extension; we would still consider your revision, provided that no similar work has been accepted for publication at NSMB or published elsewhere.

As you already know, we put great emphasis on ensuring that the methods and statistics

reported in our papers are correct and accurate. As such, if there are any changes that should be reported, please submit an updated version of the Reporting Summary along with your revision.

Reporting Summary:

Please note that all key data shown in the main figures as cropped gels or blots should be presented in uncropped form, with molecular weight markers. These data can be aggregated into a single supplementary figure item. While these data can be displayed in a relatively informal style, they must refer back to the relevant figures. These data should be submitted with the final revision, as source data, prior to acceptance, but you may want to start putting it together at this point.

Data availability: this journal strongly supports public availability of data. All data used in

accepted papers should be available via a public data repository, or alternatively, as Supplementary Information. If data can only be shared on request, please explain why in your Data Availability Statement, and also in the correspondence with your editor. Please note that for some data types, deposition in a public repository is mandatory - more information on our data deposition policies and available repositories can be found below: <https://www.nature.com/nature-research/editorial-policies/reporting-standards#availability-of-data>

[Redacted]

Sincerely,

Katarzyna Ciazynska
(she/her)
Associate Editor
Nature Structural & Molecular Biology
<https://orcid.org/0000-0002-9899-2428>

Referee expertise:

Referee #1: structural biology, cryo-ET, membrane trafficking

Referee #2: cell biology, membrane trafficking, structural biology

Referee #3: structural biology, cryo-ET, membrane trafficking

Reviewers' Comments:

Reviewer #1:

Remarks to the Author:

This study uses structural and biochemical approaches to answer the question of how the heterodimer formed by two BAR-containing proteins can function as an endosomal membrane coat. First, the authors identified the interface responsible for the formation homo- and heterodimerisation of BAR-proteins. Next, the authors follow up on how phospholipids and cargo might contribute to coat assembly, presenting at the end the architecture of the membrane-assembled heterodimer of the SNX-BAR complex, named ESCPE-1.

Overall the paper shows meticulously performed experiments that, however, do not lead to a clear point of how these results contribute to the new knowledge. Some results contradict the published data referred to in the manuscript. To my disappointment, these important discrepancies are not discussed. For example, line 149: 'Previous phosphoinositide interaction studies with the PX domain of SNX1 and SNX5 have shown weak-to-moderate binding, or even no interaction, with PtdIns(3)P, PtdIns(3,4)P2 and PtdIns(3,5)P2 (ref 15-21).' This statement is misleading as the PX domains of the two proteins have been reported to be clearly different. In all listed references PX domain of SNX1 demonstrates weak-to-moderate interactions with PtdInsPs, whereas the PX domain of SNX5, as pointed out in ref 21 cannot interact with PtdInsPs and PtdIns(3)P in particular due to the structure modification that allows cargo binding. Therefore results where the PX domain of SNX1 is unable to bind any PtdInsPs and in contrast, the PX domain of SNX5 binds with similar efficiency PtdIns(3)P, PtdIns(3,4)P2 and PtdIns(3,5)P2 directly contradict previous data and need to be explained. The other example, line 162: 'As expected, the presence of CI-MPR tail on liposomes lacking any PtdInsP not only triggered the recruitment of SNX5 alone but also augmented the recruitment of the SNX1-SNX5 heterodimer (Fig. 2c). This is fully consistent with previous data for the yeast retromer and the mammalian SNX1-SNX6 heterodimer, for which cargo facilitates their recruitment to the membrane (ref 8,22,23).' This is misleading. The references claim the cooperativity between the cargo and PIPs that was not explored in this manuscript: Ref 8 Fig 2. 'Cooperative binding of an SBM (SNX binding motif) within the CI-MPR tail and PtdIns(3)P recruits SNX-BAR to a membrane.' The cargo recognition by SNX1-SNX6 heterodimer is an insufficient condition for membrane recruitment; it needs cooperative binding to the lipid. Ref 22 says: 'These proteoliposomes were ineffective in recruiting substantial amounts of retromer (Fig.5 D), indicating that retromer does not avidly recognise cargo in this format.' As for ref 23, it studied how cargo facilitates membrane remodelling by retromer-SNX-Bar. On the same note, one of the claims the authors make in the discussion is that they can structurally confirm why metazoan retromer-SNX-Bar

assembly is impossible and SNX-BAR heterodimer is taking up on the endosomal function of retromer. However, there is no direct experimental proof, such as in vitro reconstitution (even simple retromer recruitment to the liposome in liposome flotation experiments as authors did for SNXs alone), or a model explicitly demonstrating why the retromer could not be docked on the top of a single-layered heterodimer of SNX-Bar. Despite the dynamic nature, incorporating retromer into the coat may provide additional constricting force for tubule formation, as was suggested by Zhang et al., 2020 and experimentally confirmed by Gopaldass et al., 2022 (bioRxiv). Both of these research need to be included in the discussion. Moreover, the direct comparison of two CryoEM structures: one presented in this paper and the second one - membrane-assembled SNX1 homodimer by Zhang et al., 2020 is essential for understanding.

Major issues:

1. The stability of homo- vs heterodimer must be clearly stated. Previously was shown that SNX5 S226E (Itai, N. et al., 2018) might prevent the formation of a heterodimer. Why was it not included in the set of mutants used in the study? Line 112: 'On the other hand, given that the SNX5 interface does not exhibit clear conserved hotspots, we initially introduced four-point mutations, Y219A, M233A, V240A and R368A....'
2. Clarify the discrepancy between previously published data and current results on PX domains binding and specificity and the cooperativity effect for membrane recruitment of assemblies (see above).
3. Fig 2d. The number of ITC repeats (n=1) is insufficient to determine the experimental error (provided sd values reflect the curve fitting error). Therefore, it remains unclear if SNX1-BAR promotes cargo binding to SNX5-PX ($\sim 3 \mu\text{M}$ vs $\sim 8 \mu\text{M}$ with unknown experimental error).
4. Line 237: 'In particular, VPS5 and SNX1 homodimers exhibit PX-to-BAR lateral interactions between adjacent rows, whereas SNX1-SNX5 heterodimers are characterised by intertwined BARTip-to-PX contacts along the helical row (Fig. 3h). and Line 246: 'BARTip-to-BARTip contacts occur between the side-tips of the $\alpha 2$ helices from each BAR domain, resembling an SNX1 coat (ref 27), whereas BARTip-to-PX contacts involve the tip of the $\alpha 3$ helix from each BAR domain with the start of the AH (SAH) in the adjacent molecule (Fig. 4d).' Better illustration is needed that demonstrates (1) a close-up view at the discussed elements (Bar's $\alpha 2$ and $\alpha 3$, PX) and (2) all structures need to be aligned to a bar dimer to highlight contact difference. Authors should note that the Bar-tip-to-PX interface is also present in the VPS5 structure. The current Fig3.h is confusing as it demonstrates different tilting angles and inconsistent colouring. So, if we ignore tilting and compare side-by-side structures based on contacts/relations: yellow and red dimers in SNX1-SNX5 that have tip-to-tip connections correspond to blue and red in SNX1 and VPS5 homodimers. Therefore, the relations BAR-tip (yellow)-PX (green) in SNX1-SNX5 corresponds to blue (BAR-tip) and green (PX) in VPS5.
5. Line 220: 'The lack of high resolution precluded the distinction between the two possible helical directions (Fig. 3e).' The authors should clarify that SNX1-SNX1 and SNX5-SNX5 BARTip-BARTip contact cannot be ruled out at this resolution moreover C2 symmetry was applied to an asymmetrical molecule.
6. Line 252 'The spatial proximity between these elements, the increment of affinity for

CI-MPR in presence of SNX1, and the induction of more homogeneous tubulation in presence of cargo, suggest that cargo recognition, coat assembly and membrane deformation are integrated through cooperative interactions. Indeed, introducing three Ala mutations within the SAH regions of SNX1 and SNX5 (SAH3A), or replacing the BARTip regions by Gly-Ser linkers (BT*) impaired liposomal tubulation (Fig. 4f-h).¹ As the authors pointed out, there is a spatial proximity between these structural elements but no direct evidence. These mutants can be tested in ITC to prove the point of increment of affinity for CIMPR in the presence of SNX1. However, at first, the structural integrity of all mutant proteins (at least by SEC, better by CD) and their ability to associate with membranes (liposome flotation) must be demonstrated. Fig 4 f-h needs to be clarified; what combinations of mutants were tested, e.g. SNX1(SAH3A)-SNX5(SAH3A), SNX1-SNX5(SAH3) etc. Why mutants (SNX1-BT-SAH* and SNX5-BT-SAH*) that were late tested in cells, were not tested in vitro? In addition, the replacement of the BARTip regions by Gly-Ser linkers would also affect tip-to-tip interactions that could contribute to the impairment of liposome tubulation as well as mutations of AH. Please note the results of SNX1-homodimer by Zhang et al., 2020.

Minor comments:

1. Please provide Uniprot numbers for all proteins used in this research.
2. Fig1.a, please add the Coomassie SDS PAGE of proteins with a ladder, a molecular weight of monomers, and corresponding dimers next to the SEC-MALS profile.
3. Fig. 2.a Please state in the legend if samples used in inputs and liposome flotation SDS PAGE were normalised. Have you tried PtdIns(3,4)P2 and PtdIns(3,5)P2? In your follow-up experiments with PX-domains, these phospholipids seem to be as efficient as PtdIns(3)P, at least for SNX5. Why choose to use PtdIns(3)P for reconstitution? Please be consistent with labelling, for example, in Line 153: 'PtdIns(3,4)P2 and PtdIns(3,5)P2 recruited SNX5, but none recruited the SNX1PX domain (Fig. 2b).'¹ Here, it is not clear if you compare SNX5 full length to PX of SNX1 or you compare two PX domains. The same applies to Fig2. b.
4. Fig2. c SDS PAGE needs to have a ladder. It needs to be clarified what 1%, 2% etc., represent.
5. All uncut/unmodified gels need to be shown in the supplementary material.
6. The lattice map with multiple starts needs to be demonstrated in addition to the current one (Fig 3.d). Could authors also comment on lattice completeness and defects? Please provide additional information on what 'advance helical subboxing' means and provide explicit geometrical parameters if applicable.
7. Map visualisations of the EMDB entry EMD-15413: could the authors explain what modifies the signal around the central PX-Bar dimer, resulting in the high contrast area with sharp edges?
8. Extended Data Table 2 must show the number of subtomograms at each processing step and what kind of operation was done for particle removal.
9. Line 321: 'The ESCPE-1 coat is also different to that of F-BAR domains ref 32 and N-

BAR domains ref33. This marked difference in the ESCPE-1 lattice organisation allows larger exposed protein surfaces along the tube...' Please clarify and elaborate on this statement.

10. Line 353: '... when cargo is no longer available, the coat composition might switch to SNX1 homodimers that have a significant bending capacity ref13...' In the paper, the authors compared the diameter of tubules formed by VPS5-retromer, homodimer SNX1 and SNX1-SNX5, with the heterodimer-cargo assembly being the smallest by diameter. Why here do authors suggest that SNX1 homodimer will start the neck with a much smaller diameter? And why would SNX1 homodimer appear if heterodimer is the more stable state?

Reviewer #2:

Remarks to the Author:

This study by Lopez-Robles and colleagues reports a new structural analysis of the complex of SNX1 and SNX5 (BAR domain sorting nexins that dimerise to form a complex referred to as ESCPE-1) using X-ray crystallography and cryoET of in vitro reconstituted membrane tubules. These proteins are essential for recycling a variety of transmembrane proteins from endosomal compartments through direct peptide engagement with the PX domain of SNX5, while generating membrane tubules for cargo packaging and endosomal escape. This paper nicely shows how these proteins dimerise, and provides a model for how they assemble into a polymeric array to generate the membrane tubule architecture.

The paper is well written and well presented, the methods are well described, and the data looks convincing, although the relatively low resolution of the tomographic reconstructions limits the interpretation of the data somewhat. Overall, I think the results should be of interest to the field, but I had some questions about how some of the data is interpreted, and thought that some of the discussion was overly speculative. For context I have expertise in the structural and cell biology of trafficking including the SNX proteins described here, but I am not a technical expert in the cryoEM and tomographic methods.

Questions and comments:

1. A puzzling finding for me was the reliance on PI3P for membrane interaction. The BAR domains would be expected to have little specificity beyond a general headgroup electronegativity, and the PX domains of SNX1 and SNX5 have been previously shown to not bind PI3P. From previous structural considerations, combined with liposome pelleting and biophysical binding experiments, there is expected to be no PIP binding by SNX5 (or SNX6), and SNX1 (or SNX2) has a distinct preference for PI(3,4)P₂ (Chandra et al., 2019). In the liposome flotation assays of Fig. 2B, it seems that the PX domain of SNX1 has no PIP-binding capacity at all, while the PX domain of SNX5 interacts with several PIPs non-specifically. Is this potentially due to technical differences between liposome flotation and pelleting methods, and have you tried the pelleting assay in comparison? Did the authors ever test the ability of the SNX1-SNX5 complex to tubulate membranes without PI3P, or using other PIPs instead?

2. I was also puzzled by the finding that the SNX1 BAR domain is proposed to enhance the affinity of the SNX5 PX domain for cargo peptides. Within the context of the full-length SNX1-SNX5 heterodimer complex, I could envisage that SNX1 interaction provides some stability to SNX5 that allosterically enhanced the peptide interaction, although there is no physical contact between any part of SNX1 and the SNX5 PX domain. Are the authors

proposing that the tip-PX interaction potentially observed in the cryoET reconstruction is also occurring between the isolated SNX1 BAR domain and SNX5 PX domains in solution and this is able to enhance the peptide affinity? This seems unlikely, or at least the interaction is likely to be of very low affinity in solution. On pg 11 and 12, it is mentioned that the SNX1 tip might contact the SNX5 SAH and this was consistent with the increased binding affinity of CI-MPR in the presence of SNX1. But as I understand the ITC experiments, the SAH sequence was not included in the SNX5 PX construct used for ITC? Also, I think there is a scale problem with Fig S5b and S5i, as the kcal/mol is extremely low and doesn't appear to correlate with the raw data above or with the values stated in the Fig. 2d table.

3. The main limitation to the tomographic structural analysis is that the low resolution and the imposing of C2 symmetry precludes identifying the specific orientation of the SNX1-SNX5 heterodimer. The authors acknowledge this and are careful to only discuss appropriate structural details such as the overall topology of the polymeric coat and contacts between adjacent dimers without referring to specific SNX subunits. However, it does restrict the interpretation of the structure, for example with respect to speculating about how tip interactions might occur with the SNX5 PX domain to stabilise cargo interaction. I don't think it is really possible to say this with the current structural data with any certainty. A minor point, with the modelling of the SNX1-SNX5 heterodimer into the tomogram density, it wasn't clear how a decision was made as to the orientation of the dimer, and some of the figures (such as Fig. 3e) I wasn't sure if this model docked into the tomogram was a heterodimer, or a homodimer?

4. A more general question regarding the tomographic reconstructions. Using the described methods, it appears that any heterogeneity in the coat (e.g. misaligned particles, or gaps in the lattice) would be explicitly excluded. But can the authors say anything about how homogeneous or heterogeneous the lattice is in the initial reconstructions? i.e.

5. In the final tomographic models, obviously it is not possible to identify bound lipids etc. But can the authors see whether the expected PIP binding site in SNX1 would be adjacent to the bilayer as expected? Similarly, does it seem as though the AH helices are penetrating into the bilayer as expected?

6. A minor question regarding SNX1, and SNX5 mutants discussed on pg 13. Do these have a dominant negative effect on the localisation of endogenous partner (SNX5, SNX1 respectively)?

7. I thought the final statements, and model of SNX1 homodimers and SNX1-SNX5 heterodimers shown in Fig. 7B, were overly speculative and not supported by any of the data in the paper. I'm not saying they don't exist, but is there really any evidence that SNX1 homodimers and heterodimers play different roles in the cell, or combine to affect membrane tubulation in a specific way?

Minor comments

1. I found I started to get a bit confused by the various annotations of SNX mutants throughout, with daggers, asterisks, and other symbols. I had to keep jumping back and forth to remind myself what each symbol meant. Preferably I would just list the mutations in the text, or else would it be possible to provide a supplementary table summarising the various mutant nomenclature.

2. Line 68. "process that guide cargo" should be "process that guides cargo".

3. Line 94 I think Fig. 2d should be noted as Fig. 1d.

4. Fig. 4 title. ESCAPE-1 should be ESCPE-1.

Reviewer #3:

Remarks to the Author:

Lopez-Robles provide the long-sought structure of the ESCPE-1 membrane coat involved in the retrieval of the CI-MPR and many other cellular receptors. This is an impressively thorough study spanning high resolution structure determination of the SNX1-5 protomer, cryo-ET and STA structure solution of the coat as assembled on lipids, and a complete functional validation of the role of structural interfaces in coat assembly in vitro and in cellulo, endosome recruitment, and CI-MPR retrieval. All in all, this is a seminal contribution to the structural biology of subcellular protein sorting, rigorously executed and clearly communicated.

Minor points:

Density from the Pt MAD Fourier synthesis should be shown and statistics of the MAD phasing should be provided in the crystallographic statistics table.

Line 98. Typo. 'establish' not 'stablish'

State the concentration of proteins used for liposome flotation assays

Why was the lipopeptide strategy used for flotation assays but not for the cryoET sample?

ITC suggests cooperative binding of cargo between SNX1-SNX5 heterodimers, i.e. as oligomers. cryoEM/ET samples were generated from pre-incubated high concentrations of cargo and SNX1-SNX5 before mixing with liposomes. Do oligomers of SNX1-SNX5 pre-form in these cases, i.e. in the absence of membrane?

The description of the flotation assays and ITC experiments reads as a dry catalog of data. Please guide the reader as to the motivation for the experiments and their relevance for interpreting the structure in the context of its biological function.

How did the authors determine the resolution of their final maps?

Was the data split into half-sets and independently processed as is generally done for gold-standard FSC calculations?

Provide an explicit description of how the final resolution of the STA map was determined, and a formal description of neighborhood map generation (Fig. 3f) and model fitting / model building in the methods section.

Were any specific scripts or internal versions of Dynamo written or modified specifically for this study, and if so, what is the availability of the code?

EMBD and PDB codes are provided, however, an explicit statement of coordinate deposition should still be provided.

Author Rebuttal to Initial comments

Editor Comments:

You will see that all reviewers are generally positive about the findings, however they do raise concerns which we would expect to be addressed during revision. Specifically, reviewers #1 and #2 agree that further analysis is required to clarify phosphoinositide binding. In line with reviewer #1 comments, we would expect the context of the findings in wider literature discussed more thoroughly, and replicates of experiments (eg. ITC) provided. Moreover, reviewer #2 brings up a similar issue with analysis of cargo binding affinity. Reviewer #3 points out minor points, most of which relate to methodology reporting.

We thank the reviewers for their thoughtful and constructive comments, which we have addressed in detail in the accompanying responses. The revised manuscript includes the results of several new experiments suggested by the reviewers. Additional results are also included for the reviewers' perusal. The following is a list of experiments and changes made in response to the reviewers:

- Liposome flotation assay with different SNX1^{PX} constructs to evaluate their PtdIns interaction
- Liposome pelleting assay with different SNX1^{PX} constructs to evaluate their PtdIns interaction
- Liposome flotation assay with full-length SNX5 to evaluate its PtdIns interaction
- Liposome flotation assay with full-length SNX1 to evaluate its PtdIns interaction
- Liposome flotation assay with ESCPE-1 and retromer to evaluate their potential association.
- Circular dichroism spectroscopy of the SNX1-SAH^{3A}:SNX5-SAH^{3A} mutant and the SNX1-BT*:SNX5-BT* mutant in comparison with SNX1^{WT}-SNX5^{WT}
- Extended ITC assays with CI-MPR and the SNX1-SAH^{3A}:SNX5-SAH^{3A} mutant and the SNX1-BT*:SNX5-BT* mutant
- Liposome flotation assay with the SNX1-SAH^{3A}:SNX5-SAH^{3A} mutant and the SNX1-BT*:SNX5-BT* mutant
- Extended comparative analysis between the tubular organization of SNX1-SNX5 heterodimers, SNX1 homodimers, and VPS5 homodimers
- Two additional videos highlighting various structural elements such as the BAR domains, helices $\alpha 2$ and $\alpha 3$, PX domains, amphipathic helices and the outer leaflet of the membrane.

To accommodate the new results, we have made a number of changes to the figures and accordingly edited the main text. We have also substantially re-organized the discussion to integrate the new results together with the reviewers' comments. We believe these changes have significantly improved our manuscript and hope that the reviewers will now find it suitable for publication.

Reviewers' Comments:

Reviewer #1:

Remarks to the Author:

This study uses structural and biochemical approaches to answer the question of how the heterodimer formed by two BAR-containing proteins can function as an endosomal membrane coat. First, the authors identified the interface responsible for the formation homo- and heterodimerisation of BAR-proteins. Next, the authors follow up on how phospholipids and cargo might contribute to coat assembly, presenting at the end the architecture of the membrane-assembled heterodimer of the SNX-BAR complex, named ESCPE-1.

Overall the paper shows meticulously performed experiments that, however, do not lead to a clear point of how these results contribute to the new knowledge. Some results contradict the published data referred to in the manuscript. To my disappointment, these important discrepancies are not discussed. For example, line 149: 'Previous phosphoinositide interaction studies with the PX domain of SNX1 and SNX5 have shown weak-to-moderate binding, or even no interaction, with PtdIns(3)P, PtdIns(3,4)P2 and PtdIns(3,5)P2 (ref 15-21).' This statement is misleading as the PX domains of the two proteins have been reported to be clearly different. In all listed references PX domain of SNX1 demonstrates weak-to-moderate interactions with PtdInsPs, whereas the PX domain of SNX5, as pointed out in ref 21 cannot interact with PtdInsPs and PtdIns(3)P in particular due to the structure modification that allows cargo binding. Therefore results where the PX domain of SNX1 is unable to bind any PtdInsPs and in contrast, the PX domain of SNX5 binds with similar efficiency PtdIns(3)P, PtdIns(3,4)P2 and PtdIns(3,5)P2 directly contradict previous data and need to be explained.

We thank the reviewer for the suggestion. Given the discrepancies with previous observations, we have extended the analysis of the phosphoinositide binding specificity for SNX1 and SNX5. To this end, we screened all eight phosphoinositides using liposome flotation assays. The results showed that SNX5 was slightly recruited to liposomes in a nonspecific manner (**Fig. 2b**). Thus, the previous observation of additive binding towards increasing concentrations of PtdIns(3)P, PtdIns(3,4)P2 and PtdIns(3,5)P2 (old Fig. 2b) might have been derived from nonspecific electrostatic association due to charge accumulation and not to specific binding. In the current version of the manuscript, we have replaced **Fig. 2b** with the new liposome flotation results and discussed the nonspecific minor binding of SNX5 which is in line with previous studies (Liu et al. 2006 and Chandra et al., 2019).

On the other hand, in an equivalent flotation assay with the PX domain of SNX1 we found no phosphoinositide association (**Fig. 2b**). Given that the SNX1^{PX} construct used in Chandra et al. (2019) encompassed aa142-269 and the one used in the present study encompassed aa142-282, which includes the SAH region, we wondered whether the SAH region affected the interaction with PtdInsPs. To test this, we repeated the flotation assay with the construct used in Chandra et al., 2019 but found no interaction with PtdIns(3)P or PtdIns(3,4)P2 (see **Fig. R1** for reviewers' perusal). At this point, and as noted by reviewer #2, we considered the possibility that the use of two different methods, flotation versus sedimentation, might have produced inconsistent results. Thus, we performed a sedimentation assay with both SNX1^{PX} constructs. Here too, neither SNX1^{PX142-282} nor SNX1^{PX142-269} associated with liposomes containing PtdIns(3)P or PtdIns(3,4)P2 (see **Fig. R2** for reviewers' perusal). In contrast, full length SNX1 showed strong association for PtdIns(3)P and a minor or very weak association to PtdIns(4,5)P2, PtdIns(3,5)P2 and PtdIns(3,4)P2 respectively (**Fig. 2b**). This observation is in line with previous findings such as the SNX1/SNX6 association to PtdIns(3)P observed by Yong, X. et al. 2020, and the binding of SNX1 to PtdIns(3)P and PtdIns(3,5)P2 reported by Cozier GE et al., 2002.

From these assays and previous literature, we interpret that the association of the PX domain of SNX1 with specific PtdIns is much weaker than the full-length SNX1 protein. On the other hand, the discrepancies observed in pelleting assays might have derived from the stringency of the buffer and/or the number of washing steps. Likewise, in pelleting assays, proteins that have tendency to aggregate or oligomerize might lead to overestimation of the interaction.

According to these results we have now included the liposome flotation assay of SNX1, SNX5 and SNX1^{PX}-phosphoinositide interactions as part of **Fig 2b**, and included the observation that, under our experimental conditions, the PX domain of SNX1 was unable to interact specifically with PtdIns.

For convenience, we also have placed below the new **Fig. 2b**.

Figure R1. Liposome flotation assay showing that neither SNX1^{PX142-282}, nor SNX1^{PX142-269} constructs associate with PtdIns(3)P or PtdIns(3,4)P2.

Figure R2. Liposome pelleting assay showing that neither SNX1^{PX142-282}, nor SNX1^{PX142-269} constructs associate with PtdIns(3)P or PtdIns(3,4)P2.

Figure 2b. Liposome flotation analyses to characterize the binding of SNX5, SNX1 and SNX1^{PX} to specific phosphoinositides. Note that only full length SNX1 interacts specifically with PtdIns(3)P, and to a minor extent with PtdIns(4,5)P2, PtdIns(3,5)P2, and PtdIns(3,4)P2.

The other example, line 162: 'As expected, the presence of CI-MPR tail on liposomes lacking any PtdInsP not only triggered the recruitment of SNX5 alone but also augmented the recruitment of the SNX1-SNX5 heterodimer (Fig. 2c). This is fully consistent with previous data for the yeast retromer and the mammalian SNX1-SNX6 heterodimer, for which cargo facilitates their recruitment to the membrane (ref 8,22,23).' This is misleading. The references claim the cooperativity between the cargo and PIPs that was not explored in this manuscript: Ref 8 Fig 2. 'Cooperative binding of an SBM (SNX binding motif) within the CI-MPR tail and PtdIns(3)P recruits SNX-BAR to a membrane.' The cargo recognition by SNX1-SNX6 heterodimer is an insufficient condition for membrane recruitment; it needs cooperative binding to the lipid. Ref 22 says: 'These proteoliposomes were ineffective in recruiting substantial amounts of retromer (Fig.5 D), indicating that retromer does not avidly recognise cargo in this format.' As for ref 23, it studied how cargo facilitates membrane remodelling by retromer-SNX-Bar. Unfortunate that it is confusing

We do agree with the referee that we did not explore cooperativity between cargo and PIPs for SNX1-SNX5 membrane recruitment. Instead, we focused on the observation that cargo (CIMPR) recruited more SNX1-SNX5 complex than SNX5 alone in absence of PtdIns(3)P, which suggested a cooperative action of SNX1 (Fig. 2c). To avoid potential misinterpretation on this aspect we have reworded the aforementioned sentence in a wider context: *“As expected, the presence of CI-MPR tail on liposomes lacking any PtdInsP not only triggered the recruitment of SNX5 alone but also augmented the recruitment of the SNX1-SNX5 heterodimer which suggested a cooperative action of SNX1 (Fig. 2c). This observation is in line with other cooperative effects mediated by CIMPR and PtdIns(3)P for the recruitment of SNX1-SNX6 to model membranes (Yong et al., 2020), and the positive cooperative effect of the DMT1-II cargo in the recruitment of SNX3-retromer (Harrison et al., 2013). Also, in a similar way, the yeast Vps10 cargo enhanced local clustering of Vps5-Vps17-retromer in membrane microdomains (Purushothaman et al., 2018)”*

On the same note, one of the claims the authors make in the discussion is that they can structurally confirm why metazoan retromer-SNX-Bar assembly is impossible and SNX-BAR heterodimer is taking up on the endosomal function of retromer. However, there is no direct experimental proof, such as in vitro reconstitution (even simple retromer recruitment to the liposome in liposome flotation experiments as authors did for SNXs alone), or a model explicitly demonstrating why the retromer could not be docked on the top of a single-layered heterodimer of SNX-Bar. Despite the dynamic nature, incorporating retromer into the coat may provide additional constricting force for tubule formation, as was suggested by Zhang et al., 2020 and experimentally confirmed by Gopaldass et al., 2022 (bioRxiv). Both of these research need to be included in the discussion. Moreover, the direct comparison of two

CryoEM structures: one presented in this paper and the second one - membrane-assembled SNX1 homodimer by Zhang et al., 2020 is essential for understanding.

We thank the reviewer for this comment as it addresses important aspects that might have not been illustrated and/or discussed properly. The discussion was not intended to create the impression that the ‘metazoan retromer-SNX-Bar assembly is impossible’. Indeed, recent studies have demonstrated that ESCPE-1 can engage SNX27-Retromer through the interaction between SNX1 and the SNX27-FERM domain to promote recycling of certain cargoes (Yong X et al. 2020; Yong X et al. 2021; Chandra M et al. 2022). However, this SNX27:Retromer:ESCPE-1 ‘supercomplex’ has been proposed to be of a transient nature at the emerging membrane bud from where cargo is handed to ESCPE-1 (Simonetti B et al. 2022). Given that tubular and planar membranes impose distinct spatial restrictions, we do not exclude the possibility that, for retrieval of certain cargos, retromer and other factors could associate with ESCPE-1 in pseudo-planar membranes through a different lattice organization. Nonetheless, the observation by Gopaldass et al., 2022 (bioRxiv) that retromer subunits VPS35-VPS26-VPS29 are incorporated into the VPS5-VPS17 coat to provide additional constriction force is in good agreement with previous data from several labs indicating that yeast retromer forms a stable pentameric complex. However, in higher metazoans, it is well established that retromer does not form stable complexes with SNX1/2-SNX5/6. In this regard, following the reviewer suggestion, we co-incubated retromer with SNX1-SNX5 and CIMPR, and performed a liposome flotation. The results showed that SNX1-SNX5 was unable to recruit retromer (Extended Data Fig. 7c). In contrast, co-incubation of retromer with SNX3 and the DMT1-II cargo under the same experimental conditions resulted in retromer recruitment to the membrane (Extended Data Fig. 7c) which is consistent with their direct association (Lucas M et al. 2016). In addition, we have included an image of retromer contacts across the VPS5 lattice in *Chaetomium thermophilum* (Extended Data Fig. 7b). The image shows contact patches that involve adjacent BAR domains from different dimers. The geometrical distribution of these patches is not conserved in the SNX1 lattice and neither is conserved in the SNX1-SNX5 lattice indicating that retromer would not be able to dock in the same configuration as in *Chaetomium thermophilum*. These results confirm that mammalian retromer is not recruited by SNX1-SNX5 as it is by SNX3, and support the notion of functional diversification between the mammalian and yeast retromer. These results are now discussed in the revised version of the manuscript.

Extended Data Fig. 7 is also shown below for convenience.

Extended Data Figure 7. ESCPE-1 lattice scaffold is different from that of the SNX1 dimer and the fungal VPS5 dimer, and is unable to recruit retromer. **(a)** Comparison of membrane lattice scaffolds of (i) the mammalian SNX1-SNX5 heterodimer (current study), (ii) the mammalian SNX1 dimer [Zhang, Y. et al. 2021], and (iii) the fungal VPS5 dimer solved in the context of retromer complex [Kovtun, O. et al. 2018]. Surface coverage calculations were done assuming an average coverage of the membrane of ≈ 50 nm for each PX-BAR dimer. **(b)** Representation of the intermolecular contacts on the VPS5 lattice (colored in dark red) involved in the association with the VPS26 subunit of the retromer complex. Note that the distribution of contacts on two adjacent BAR domains (green and yellow, or pink and blue) from separate dimers is not conserved in the SNX1 or SNX1-SNX5 lattices. **(c)** In flotation assays, (i) retromer (VPS35-VPS29-VPS26 subunits) was recruited by SNX3 and the DMT1-II cargo to liposomes

(DOPC/DOPE/DOPS/PtdIns(3)P/Liss Rhod-PE 45:28:20:5:2 molar ratio) whereas (ii) retromer was not recruited by SNX1-SNX5 and the CI-MPR cargo.

Major issues:

1. The stability of homo- vs heterodimer must be clearly stated. Previously was shown that SNX5 S226E (Itai, N. et al., 2018) might prevent the formation of a heterodimer. Why was it not included in the set of mutants used in the study? Line 112: 'On the other hand, given that the SNX5 interface does not exhibit clear conserved hotspots, we initially introduced four-point mutations, Y219A, M233A, V240A and R368A...

In the cited reference (Itai, N. et al., 2018), the S226E mutant on SNX5 was evaluated to mimic the effect of phosphorylation on S226 showing that it was able to block the association with SNX1 or SNX2 whereas S226A exhibited a wildtype phenotype. Despite this interesting regulatory finding by a post-translational modification, our efforts were centered on a different aspect: how the energetic distribution at the interfaces contributes to specificity between SNXs. Our analysis did not identify S226 as an energetically 'hot' residue for the interaction; indeed the S226A mutant behaved as SNX5WT (Itai, N. et al., 2018). Instead, we selected the nearby Y219 which showed better conservation and energetic scores for interface stabilization.

2. Clarify the discrepancy between previously published data and current results on PX domains binding and specificity and the cooperativity effect for membrane recruitment of assemblies (see above).

Please, refer to our response above where we discuss this point.

3. Fig 2d. The number of ITC repeats (n=1) is insufficient to determine the experimental error (provided sd values reflect the curve fitting error). Therefore, it remains unclear if SNX1-BAR promotes cargo binding to SNX5-PX (~3 μ M vs ~8 μ M with unknown experimental error).

We believe this is a misunderstanding since all the ITC experiments were performed at least in duplicate under similar conditions to confirm the reproducibility of the results. The n=1 described in the ITC table is the fit to a standard single-site binding model. In the revised version of the manuscript, we have edited **Fig 2d** to show the

average and standard deviation K_d values, and placed the representative data as **Extended Data Fig. 5**. Since this comment is partially related to point #2 from reviewer_2, we would like to refer this reviewer to our additional comments below.

4. Line 237: 'In particular, VPS5 and SNX1 homodimers exhibit PX-to-BAR lateral interactions between adjacent rows, whereas SNX1-SNX5 heterodimers are characterised by intertwined BARTip-to-PX contacts along the helical row (Fig. 3h). and Line 246: 'BARTip-to-BARTip contacts occur between the side-tips of the α_2 helices from each BAR domain, resembling an SNX1 coat (ref 27), whereas BARTip-to-PX contacts involve the tip of the α_3 helix from each BAR domain with the start of the AH (SAH) in the adjacent molecule (Fig. 4d).' Better illustration is needed that demonstrates (1) a close-up view at the discussed elements (Bar's α_2 and α_3 , PX) and (2) all structures need to be aligned to a bar dimer to highlight contact difference. Authors should note that the Bar-tip-to-PX interface is also present in the VPS5 structure. The current Fig3.h is confusing as it demonstrates different tilting angles and inconsistent colouring. So, if we ignore tilting and compare side-by-side structures based on contacts/relations: yellow and red dimers in SNX1-SNX5 that have tip-to-tip connections correspond to blue and red in SNX1 and VPS5 homodimers. Therefore, the relations BAR-tip (yellow)-PX (green) in SNX1-SNX5 corresponds to blue (BAR-tip) and green (PX) in VPS5.

We appreciate this comment from the reviewer. As suggested, in the revised manuscript, we have included two videos (**Extended Data Video 1** and **Extended Data Video 2**) that highlight the discussed elements (BAR domains, helices α_2 and α_3 , PX domains, and amphipathic helices). We think that this visual aid improves the description of the lattice contacts. Also, we have updated Fig. 3 with a different view of SNX1-SNX5 neighboring molecules showing two tip-to-tip contacts that are equivalent to the SNX1 and VPS5 representations. Regarding the tilting, we prefer to maintain the orientation of the molecules relative to the longitudinal axis of the tube as it gives a wider perspective. However, we agree with the referee that a side-by-side comparison might help in a better interpretation of the differences between coats. For this reason, we have included a new figure with this information (see **Extended Data Fig. 7**)

5. Line 220: 'The lack of high resolution precluded the distinction between the two possible helical directions (Fig. 3e).' The authors should clarify that SNX1-SNX1 and SNX5-SNX5 BARTip-BARTip contact cannot be ruled out at this resolution moreover C2 symmetry was applied to an asymmetrical molecule.

We acknowledge that at the current resolution it would be possible to fit BARTip-to-BARTip contacts between identical protomers holding alternate orientations. However, considering that cargo binding is enhanced through SNX1^{BAR} and SNX5^{PX} contacts, only successive SNX1-SNX5 heterodimers would enable Tip-to-PX contacts between SNX1^{BAR} and SNX5^{PX}. Thus, although we could not exclude orientations with contacts between identical protomers, we considered the head-to-tail interlinkage the most plausible scaffold in presence of cargo. As suggested, we have rephrased our original argument to clarify this aspect.

6. Line 252 'The spatial proximity between these elements, the increment of affinity for CI-MPR in presence of SNX1, and the induction of more homogeneous tubulation in presence of cargo, suggest that cargo recognition, coat assembly and membrane deformation are integrated through cooperative interactions. Indeed, introducing three Ala mutations within the SAH regions of SNX1 and SNX5 (SAH3A), or replacing the BARTip regions by Gly-Ser linkers (BT*) impaired liposomal tubulation (Fig. 4f-h).' As the authors pointed out, there is a spatial proximity between these structural elements but no direct evidence. These mutants can be tested in ITC to prove the point of increment of affinity for CIMPR in the presence of SNX1. However, at first, the structural integrity of all mutant proteins (at least by SEC, better by CD) and their ability to associate with membranes (liposome flotation) must be demonstrated. Fig 4 f-h needs to be clarified; what combinations of mutants were tested, e.g. SNX1(SAH3A)-SNX5(SAH3A), SNX1-SNX5(SAH3) etc. Why mutants (SNX1-BT-SAH* and SNX5-BT-SAH*) that were late tested in cells, were not tested in vitro? In addition, the replacement of the BARTip regions by Gly-Ser linkers would also affect tip-to-tip interactions that could contribute to the impairment of liposome tubulation as well as mutations of AH. Please note the results of SNX1-homodimer by Zhang et al., 2020.

We apologize for the ambiguity on Fig 4 f-h. The SAH^{3A} mutant involves three Ala mutations within the SAH regions in both SNX1 and SNX5, whereas the BT* mutant involves the replacement of the BAR^{tip} regions by Gly-Ser linkers in both SNX1 and SNX5. For clarity we have indicated the mutants as SNX1-SAH^{3A}:SNX5-SAH^{3A}, and SNX1- BT*:SNX5- BT* in the text and in Fig 4 g,h.

Regarding the SAH* mutants, we were unable express these proteins in E. coli. despite exploring several strategies as codon optimization, lowering the expression temperature or using other strains which ultimately precluded the *in vitro* analysis. Nonetheless, in line with the valuable suggestions from this reviewer we have extended the ITC analysis on CI-MPR binding with the SNX1-SAH^{3A}:SNX5-SAH^{3A} mutant and the SNX1-BT*:SNX5-BT* mutant. Both mutants diminished the binding affinity for CIMPR, thus providing additional evidence that the Tip-to-PX contacts contribute to enhance the affinity for CIMPR (see **Extended Data Fig. 8a,b**). Although these mutants displayed similar CD spectra indicating that the secondary structures remained mostly unaffected (see **Extended**

Data Fig. 8c), we noticed that the SAH^{3A} and BT* mutants exhibited lower association with synthetic liposomes (see **Extended Data Fig. 8d**). In particular, the BT* mutants displayed a major loss in the recruitment of SNX5. The fact that SNX1-BT* displaced SNX5-BT* on a flotation assay was unexpected and difficult to interpret as both proteins were purified as a complex and none of the mutants involved interface residues. Although we have not determined a mechanistic explanation for the SNX5-BT* unexpected behavior, the BT* mutants clearly affected coat assembly and the binding correlation between protomers in synthetic liposomes. These results are now discussed in the revised version of the manuscript.

Extended Data Fig. 8 is also shown below for convenience.

Extended Data Fig. 8: Mutations within the SAH regions, or within the BAR-TIP regions in SNX1 and SNX5 affect cargo binding and interfere with membrane association. (a) Summary of Kds between the CI-MPR bipartite sorting motif (amino acids 2347-2375) titrated into the SNX1-SAH^{3A}:SNX5-SAH^{3A} mutant or the SNX1-

BT*:SNX5- BT* mutant. Values are the mean \pm standard deviation (SD) from two independent experiments. **(b)** Representative ITC experiments for the binding of the previous SAH3A and BT* mutants. Top panels show the raw data and bottom panels represent the integrated and normalized data fit with a 1:1 binding model. **(c)** Circular dichroism (CD) spectra of wild-type SNX1-SNX5, the SNX1-SAH^{3A}:SNX5-SAH3A mutant, and the SNX1-BT*:SNX5- BT* mutant. **(d)** Liposome flotation assay of wild-type SNX1-SNX5, the SNX1-SAH^{3A}:SNX5-SAH3A mutant, and the SNX1-BT*:SNX5-BT* mutant. The liposome composition was: DOPC/DOPE/DOPS/PtdIns(3)P/Liss_Rhod-PE 45:28:20:5:2 molar ratio.

Minor comments:

1.Please provide Uniprot numbers for all proteins used in this research.

In the revised manuscript we have included a table as **Extended Data Table 1** with all proteins and constructs used in this study together with their corresponding symbols employed throughout the text and Uniprot numbers.

2.Fig1.a, please add the Coomassie SDS PAGE of proteins with a ladder, a molecular weight of monomers, and corresponding dimers next to the SEC-MALS profile.

We acknowledge these recommendations which have now been included in **Fig. 1a** of the revised manuscript.

Fig. 1a is also shown below for convenience.

Fig. 1a: SDS-PAGE and SEC-MALS analysis of full length SNX1, SNX5 and SNX1-SNX5 showing the molecular weight difference between species.

3. Fig. 2.a Please state in the legend if samples used in inputs and liposome flotation SDS PAGE were normalised. Have you tried PtdIns(3,4)P2 and PtdIns(3,5)P2? In your follow-up experiments with PX-domains, these phospholipids seem to be as efficient as PtdIns(3)P, at least for SNX5. Why choose to use PtdIns(3)P for reconstitution? Please be consistent with labelling, for example, in Line 153: 'PtdIns(3,4)P2 and PtdIns(3,5)P2 recruited SNX5, but none recruited the SNX1PX domain (Fig. 2b).' Here, it is not clear if you compare SNX5 full length to PX of SNX1 or you compare two PX domains. The same applies to Fig2. b.

We thank the reviewer for these valuable comments. All SDS PAGE samples originated from flotation assays were normalized relative to their absorbance at 573 nm associated with the Liss Rhodamine-PE content. This has been updated in the figure legends.

Regarding the selection of PtdIns(3)P in our reconstitution experiments, this was initially based on previous studies where SNX1 promoted membrane tubulation in presence of this phospholipid [van Weering, J.R. et al. 2012; Zhang et al., 2020]. Our follow-up flotation assays mentioned above confirmed the binding preference of SNX1 for PtdIns(3)P, but also showed decreasing affinities for PtdIns(4,5)P2, PtdIns(3,5)P2, and PtdIns(3,4)P2 respectively. In response to this reviewer and related to point 1.3 from reviewer #2, we co-incubated SNX1-SNX5 and CI-MPR with liposomes containing either PtdIns(3,4)P2, or lacking any PtdIns. We observed that in presence of PtdIns(3,4)P2 there was tubulation activity, albeit to a lesser extent than with PtdIns(3)P which is consistent with their binding affinities, whereas in absence of any PtdIns there was no tubulation (see **Fig. 2f**). These results have been included as **Fig. 2f**, and discussed in the revised manuscript.

4. Fig2. c SDS PAGE needs to have a ladder. It needs to be clarified what 1%, 2% etc., represent.

We acknowledge these recommendations which have now been included in **Fig. 2c** of the revised manuscript.

5. All uncut/unmodified gels need to be shown in the supplementary material.

We have now included all unprocessed gel images as part of **supplementary material**.

6. The lattice map with multiple starts needs to be demonstrated in addition to the current one (Fig 3.d).

For demonstrative purposes, we provide here an example of the maps used for the characterization of the multiple starts. In this map, an averaged filament segment is expressed in cylindrical coordinates, then projected in the radial direction.

Figure R3. Radial projection map of a tube

Our methodology to state the multiple starts proceeds by first visually inspecting this "cylindrical projection map" on each tube, then moving to a numerical characterization of the apparent lattice.

The inset shows (a zoom of) the Fourier Transform of the image, which is used in the determination of the lattice parameters of each tube.

Could authors also comment on lattice completeness and defects?

This question has been raised and extended by reviewer #2. We provide there our perspective on this question in a more complete context.

Please provide additional information on what 'advance helical subboxing' means and provide explicit geometrical parameters if applicable.

Our mention to "advanced helical subboxing" in paragraph "Structure of the membrane-assembled SNX1-SNX5 coat" of the "Results" section was meant to make a reference to the procedure explained later in the Method Details section. In order to clarify that the used approach will indeed be detailed in a later point, in the main text we have replaced the sentence:

"By further subtomogram averaging using advanced helical sub-boxing techniques, we generated two different averages"

With:

"Following the methodology described in the 'Subtomogram Averaging' paragraph in the Method Details, we characterized the helical behavior of the coating of each filament to perform a particle extraction guided by the lattice geometry determined in each case, leading to two different averages"

As explained before, geometrical parameters of the lattice were determined for each filament separately. Those parameters were used merely to drive the extraction of particles on each tube.

7. Map visualisations of the EMDB entry EMD-15413: could the authors explain what modifies the signal around the central PX-Bar dimer, resulting in the high contrast area with sharp edges?

The high contrast area is due to the mask used during subtomogram averaging focused on the central PX-Bar dimer, a procedure intended to isolate the effect of neighboring intensities and of the tube itself. As discussed in our response, the short-range ordering of the coating allows for the identification of the underlying lattice, but not for a solidary refinement of a set of neighboring PX-bar dimers.

8. Extended Data Table 2 must show the number of subtomograms at each processing step and what kind of operation was done for particle removal.

We thank the reviewers for leading us to notice the mismatch between the number indicated in the methods section of the main text (which reflected the raw number of subtomograms before the application of the thresholding step), and the final number reflected in Extended Data Table 2. We have clearly stated in both locations that the initial number of subtomograms is 77,436 and the final, after application of the multicriterion thresholding as a single step is 15,116.

9. Line 321: 'The ESCPE-1 coat is also different to that of F-BAR domains ref 32 and N-BAR domains ref33. This marked difference in the ESCPE-1 lattice organisation allows larger exposed protein surfaces along the tube...' Please clarify and elaborate on this statement.

The comparison of SNX1-SNX5 with F-BAR and N-BAR domains was potentially misleading as no PX domains were contributing to membrane scaffolding. Thus, we have centered our analysis on the SNX1 and VPS5 scaffolds and their differences in aerial density and surface coverage (see **Extended Data Fig. 7a**).

10. Line 353: '... when cargo is no longer available, the coat composition might switch to SNX1 homodimers that have a significant bending capacity ref13...' In the paper, the authors compared the diameter of tubules formed by VPS5-retromer, homodimer SNX1 and SNX1-SNX5, with the heterodimer-cargo assembly being the smallest by diameter. Why here do authors suggest that SNX1 homodimer will start the neck with a much smaller diameter? And why would SNX1 homodimer appear if heterodimer is the more stable state?

We agree with the reviewer that the reconstructed cryo-EM tubular coat of SNX1 homodimer exhibits slightly larger diameter than the SNX1-SNX5 coat. However, as observed by Zhang, Y. et al. 2021, and us (**Extended Data Fig. 1a**) SNX1 can make tubes of variable diameters ranging from 30 to 50 nm. Yet, our speculative hypothesis is not that the fission mechanism could be driven by constriction, but rather by the creation of frictional stress to the underlying membrane during the transition from SNX1-SNX5 heterodimers to SNX1 homodimers. The transition between coats might be ultimately guided by the lack of cargo in the sorting tube. Indeed, the finding that SNX1-BT* displaced SNX5-BT* from the membrane on a flotation assay (**Extended Data Fig. 8d**) suggest that the heterodimer can be disassembled in the presence of membranes through alteration of lattice contacts. Since this topic is partially related to point #7 from reviewer_2, we would like to refer this reviewer to our additional comments below.

Reviewer #2:

Remarks to the Author:

This study by Lopez-Robles and colleagues reports a new structural analysis of the complex of SNX1 and SNX5 (BAR domain sorting nexins that dimerise to form a complex referred to as ESCPE-1) using X-ray crystallography and cryoET of in vitro reconstituted membrane tubules. These proteins are essential for recycling a variety of transmembrane proteins from endosomal compartments through direct peptide engagement with the PX domain of SNX5, while generating membrane tubules for cargo packaging and endosomal escape. This paper nicely shows how these protein dimerise, and provides a model for how they assemble into a polymeric array to generate the membrane tubule architecture.

The paper is well written and well presented, the methods are well described, and the data looks convincing, although the relatively low resolution of the tomographic reconstructions limits the interpretation of the data somewhat. Overall, I think the results should be of interest to the field, but I had some questions about how some of the data is interpreted, and thought that some of the discussion was overly speculative. For context I have expertise in the structural and cell biology of trafficking including the SNX proteins described here, but I am not a technical expert in the cryoEM and tomographic methods.

Questions and comments:

1. A puzzling finding for me was the reliance on PI3P for membrane interaction. The BAR domains would be expected to have little specificity beyond a general headgroup electronegativity, and the PX domains of SNX1 and SNX5 have been previously shown to not bind PI3P. From previous structural considerations, combined with liposome pelleting and biophysical binding experiments, there is expected to be no PIP binding by SNX5 (or SNX6), and SNX1 (or SNX2) has a distinct preference for PI(3,4)P2 (Chandra et al., 2019). In the liposome flotation assays of Fig. 2B, it seems that the PX domain of SNX1 has no PIP-binding capacity at all, while the PX domain of SNX5 interacts with several PIPs non-specifically. Is this potentially due to technical differences between liposome flotation and pelleting methods, and have you tried the pelleting assay in comparison? Did the authors ever test the ability of the SNX1-SNX5 complex to tubulate membranes without PI3P, or using other PIPs instead?

We thank the reviewer for these insightful comments. Given the similarity to the comments made by reviewer #1 on this issue, we refer this reviewer to our response above.

2. I was also puzzled by the finding that the SNX1 BAR domain is proposed to enhance the affinity of the SNX5 PX domain for cargo peptides. Within the context of the full-length SNX1-SNX5 heterodimer complex, I could envisage that SNX1 interaction provides some stability to SNX5 that allosterically enhanced the peptide interaction, although there is no physical contact between any part of SNX1 and the SNX5 PX domain. Are the authors proposing that the tip-PX interaction potentially observed in the cryoET reconstruction is also occurring between the isolated SNX1 BAR domain and SNX5 PX domains in solution and this is able to enhance the peptide affinity? This seems unlikely, or at least the interaction is likely to be of very low affinity in solution. On pg 11 and 12, it is mentioned that the SNX1 tip might contact the SNX5 SAH and this was consistent with the increased binding affinity of CI-MPR in the presence of SNX1. But as I understand the ITC experiments, the SAH sequence was not included in the SNX5 PX construct used for ITC? Also, I think there is a scale problem with Fig S5b and S5i, as the kcal/mol is extremely low and doesn't appear to correlate with the raw data above or with the values stated in the Fig. 2d table.

We agree with the reviewer that the interaction between the SNX5^{PX} domain and the SNX1^{BAR} domain is of very low affinity but still sufficient to generate an allosteric effect for cargo binding. In this regard, as suggested by reviewer #1 and related to these comments, we extended the ITC analysis on CIMPR binding with the SNX1-SAH^{3A}:SNX5-SAH^{3A} mutant and the SNX1-BT*:SNX5-BT* mutant. In the case of the SNX1-BT*:SNX5-BT* mutant the K_d reverted to values similar to that of the SNX5^{PX} domain alone, whereas for the SNX1-SAH^{3A}:SNX5-SAH^{3A} mutant, the K_d was slightly higher than the wild type but reproducible in repetitive assays (see **Extended Data Fig. S8**). In the case of the SNX1-SAH^{3A}:SNX5-SAH^{3A} mutant, the partial increment of the K_d values might have derived from an incomplete inhibition of the interaction between the SNX1^{PX} domain and the SNX1^{BAR} domain, probably due to the presence of additional side chain and/or mainchain contacts. Unfortunately, the low resolution of the map precludes such detailed analysis. In any case, both mutants diminished the binding affinity for CIMPR which is in good agreement with the allosteric effect between the SNX5^{PX} domain and the SNX1^{BAR} domain for cargo binding. It should be noted as well that this allosteric behavior can be highly reinforced by the cumulative effects of multivalent interactions during coat oligomerization.

The SNX5^{PX} construct used in this study encompassed aa 1-183 with includes the SAH sequence. This has been included in the **Extended Data Table 1**.

We apologize for the typo in the scale values stated in the Extended Data Fig. 5b and 5i. This has been corrected in the revised manuscript.

3. The main limitation to the tomographic structural analysis is that the low resolution and the imposing of C2 symmetry precludes identifying the specific orientation of the SNX1-SNX5 heterodimer. The authors acknowledge this and are careful to only discuss appropriate structural details such as the overall topology of the polymeric coat and contacts between adjacent dimers without referring to specific SNX subunits. However, it does restrict the interpretation of the structure, for example with respect to speculating about how tip interactions might occur with the SNX5 PX domain to stabilise cargo interaction. I don't think it is really possible to say this with the current structural data with any certainty. A minor point, with the modelling of the SNX1-SNX5 heterodimer into the tomogram density, it wasn't clear how a decision was made as to the orientation of the dimer, and some of the figures (such as Fig. 3e) I wasn't sure if this model docked into the tomogram was a heterodimer, or a homodimer?

We agree with the reviewer that the low-resolution model derived from the cryo-ET and STA analysis precludes detailed visualization of lattice contacts between heterodimers. For the same reason, the orientation of the SNX1-SNX5 heterodimers around the tube remains undefined. However, as described above, the structure-guided mutations (SAH^{3A}:SNX5-SAH^{3A} and SNX1- BT*:SNX5- BT*) and the ITC analysis support the organization of successive heterodimers where the SNX1^{BAR} and SNX5^{PX} domains contact each other to stabilize cargo interaction. In Fig. 3e, we tried to show the two possible orientations of heterodimers within the helical density denoted as (I) and (II) but regrettably we did not include labels. In our view, option (II) might suit better the twist around helical turns due to length differences between BAR domains (see **Extended Data Fig. 1f**) but given the lack of additional data to support any specific orientation of the heterodimer around the tube we prefer to maintain both options. In the revised manuscript we acknowledge that we cannot exclude other orientations of the heterodimer but consider that the head-to-tail interlinkage is the most plausible scaffold in the presence of cargo.

4. A more general question regarding the tomographic reconstructions. Using the described methods, it appears that any heterogeneity in the coat (e.g. misaligned particles, or gaps in the lattice) would be explicitly excluded. But can the authors say anything about how homogeneous or heterogeneous the lattice is in the initial reconstructions? i.e.

The reviewer is right: the analysis doesn't allow a sound assessment of the global regularity of the tube coats. Our characterization of the lattice has been driven towards its use as a guide to ensure the highest possible quality of the subtomograms used for the final reconstruction. On a qualitative level, several hints point to an overall conservation of a short-range ordering, although with ubiquitous and strong deviations from perfect symmetry. The short-range ordering is apparent in the neighborhood maps provided in the text, and its approximate conservation is inferred

from the visual inspection of radial projection maps and from the fact that extraction of particles along the characterized helical paths does produce a repeated signal that leads to a cohered average. However, the strong deviations from lattice regularity are evidenced by the high number of subtomograms eliminated from the initial sampling.

The attempt to assess if the failure of a putative particle contained in a subtomogram to adapt to a lattice mode is due to a local failure of the lattice model or to a false positive can swiftly fall into a circular reasoning. Sharing with the reviewers the same curiosity, we have devised different approaches aimed at breaking this circularity. We have attempted to identify patterns in the spots where deviations from lattice homogeneity are most apparent, then tried to correlate such spots with the local morphology of the tubes (bends, areas of high curvature, difference between convex and concave areas). Such approaches were not able to produce a reliable quantitative assessment on the tubes' homogeneity.

5. In the final tomographic models, obviously it is not possible to identify bound lipids etc. But can the authors see whether the expected PIP binding site in SNX1 would be adjacent to the bilayer as expected? Similarly, does it seem as though the AH helices are penetrating into the bilayer as expected?

In response to the reviewer's questions, we have included a video (**Extended Data Video 1**) that shows how both PX domains lay adjacent to the outer leaflet of the membrane and how the AH helices are partially embedded and aligned along the longitudinal axis of the tube.

6. A minor question regarding SNX1, and SNX5 mutants discussed on pg 13. Do these have a dominant negative effect on the localisation of endogenous partner (SNX5, SNX1 respectively)?

We agree that this experiment would be interesting, but it would require identification of conditions for high overexpression and of antibodies that stain the endogenous proteins – something that we cannot do within the timeframe for resubmission. Furthermore, we do not think these experiments are critical to support the conclusions of the study.

7. I thought the final statements, and model of SNX1 homodimers and SNX1-SNX5 heterodimers shown in Fig. 7B, were overly speculative and not supported by any of the data in the paper. I'm not saying they don't exist, but is

there really any evidence that SNX1 homodimers and heterodimers play different roles in the cell, or combine to affect membrane tubulation in a specific way?

We appreciate this comment and agree with the reviewer that the combined model of SNX1 and SNX1-SNX5 coats is speculative at this stage. Yet, we believe that there are some insights that point to this hypothesis. First, SNX1 homodimers might coexist with SNX1-SNX5 heterodimers given that, (I) phosphorylation of SNX5 precludes heterodimerization and (II) the number of SNX1 molecules is slightly higher than that of SNX5 molecules (5.7×10^4 versus 4×10^4) in HeLa cells (Nagarjuna Nagaraj N et al. 2011). Second, in the absence of cargo SNX1 homodimers would be recruited to the neck of the tubule more efficiently than heterodimers given that SNX5 on its own does not associate with phospholipids. And third, and related to comment #10 from reviewer 2, alteration of lattice contacts can lead to the displacement of one protomer from the membrane which might be favored by the lack of cargo as it contributes to stabilize the lattice. The aim of the model is to speculate about the possibility that cargo could also regulate the transition from elongation to scission. In this sense, in the revised version of the manuscript we have rephrased the final statements to emphasize that this is a speculative model that needs to be confirmed by future studies.

Minor comments

1. I found I started to get a bit confused by the various annotations of SNX mutants throughout, with daggers, asterisks, and other symbols. I had to keep jumping back and forth to remind myself what each symbol meant. Preferably I would just list the mutations in the text, or else would it be possible to provide a supplementary table summarising the various mutant nomenclature.

We thank the reviewer for this helpful comment. In the new version of the manuscript, we have included a Table with detailed information about constructs and mutants, and their corresponding nomenclature used throughout the text (see Extended Data Table 1).

2. Line 68. “process that guide cargo” should be “process that guides cargo”.

We apologize for the typo in the sentence. This has been corrected.

3. Line 94 I think Fig. 2d should be noted as Fig. 1d.

We apologize for the mistake. This has been corrected.

4. Fig. 4 title. ESCAPE-1 should be ESCPE-1.

Sorry for the mistake. This has been corrected.

Reviewer #3:

Remarks to the Author:

Lopez-Robles provide the long-sought structure of the ESCPE-1 membrane coat involved in the retrieval of the CI-MPR and many other cellular receptors. This is an impressively thorough study spanning high resolution structure determination of the SNX1-5 protomer, cryo-ET and STA structure solution of the coat as assembled on lipids, and a complete functional validation of the role of structural interfaces in coat assembly in vitro and in cellulo, endosome recruitment, and CI-MPR retrieval. All in all, this is a seminal contribution to the structural biology of subcellular protein sorting, rigorously executed and clearly communicated.

We thank this reviewer for his/her highly positive remarks and greatly appreciate his/her enthusiasm for our study.

Minor points:

Density from the Pt MAD Fourier synthesis should be shown and statistics of the MAD phasing should be provided in the crystallographic statistics table.

This is an important point that we had overlooked. In the revised manuscript we include the MAD phased density map and the Pt anomalous difference map in the **Extended Data Fig. 1b** and included the MAD phasing statistics in **Extended Data Table. 2**

Extended Data Fig. 1b and phasing statistics are also shown below for convenience.

Extended Data Figure 1b: MAD density map (blue) contoured at 1.5σ and Pt anomalous difference map (magenta) contoured at 4.0σ superimposed on the refined structure. Sidechains of H246, C318 and M414 are highlighted in yellow as examples of platinum binders.

Phasing statistics

	SNX1-SNX5
	Pt_PEAk
Pt sites found/expected	14/42
FOM after SHARP	0.38
FOM after	0.55
SOLOMON	

Line 98. Typo. ‘establish’ not ‘stablish’

We apologize for the typo. This has been corrected

State the concentration of proteins used for liposome flotation assays

This is an important point that we had overlooked. The concentration of proteins in liposome flotation assays was 25 μM . During the revision, we also found a typo related with the CI-MPR concentration used in the tubulation assays which originally indicated 300 μM but it should be 30 μM . We have updated this information in the Methods section of the revised manuscript.

Why was the lipopeptide strategy used for flotation assays but not for the cryoET sample?

This an interesting point that we considered for the cryoET sample preparation but decided not to pursue because the efficiency of the reaction was about 70% and we were concerned of potential adverse effects resulting from the reaction residuals.

ITC suggests cooperative binding of cargo between SNX1-SNX5 heterodimers, i.e. as oligomers. cryoEM/ET samples were generated from pre-incubated high concentrations of cargo and SNX1-SNX5 before mixing with liposomes. Do oligomers of SNX1-SNX5 pre-form in these cases, i.e. in the absence of membrane?

As suggested by the reviewer, we have checked by cryo-EM a sample of preincubated CI-MPR with SNX1-SNX5 heterodimers but did not observe the formation of linear fibers or flat lattices. However, we did observe numerous highly flexible elongated particles with lengths of ≈ 15 and ≈ 30 nm, and occasionally even longer, which could be compatible with the presence of single ESCPE-1 particles and loosely assembled ESCPE-1 dimers and trimers (see **Fig. R4** for reviewers' perusal).

Figure R4. Cryo-EM image of SNX1-SNX5 incubated with the CI-MPR tail.

The description of the flotation assays and ITC experiments reads as a dry catalog of data. Please guide the reader as to the motivation for the experiments and their relevance for interpreting the structure in the context of its biological function.

We thank the reviewer for this suggestion to improve readability. Given the new liposome flotation results of SNX1 and SNX5 interaction with PtdIns, and the retromer flotation assays, we have rewritten a large part of the text corresponding to these results. Similarly, we have included the new ITC analysis on CIMPR binding to the SNX1-SAH^{3A}:SNX5-SAH^{3A} mutant and the SNX1-BT*:SNX5-BT* mutant, and rephrased the ITC experiments placing the rationale in a wider context.

How did the authors determine the resolution of their final maps?

See below

Was the data split into half-sets and independently processed as is generally done for gold-standard FSC calculations?

See below

Provide an explicit description of how the final resolution of the STA map was determined,...

The data was split into half sets and processed independently. Final resolution was computed through FSC computation with Relion 3.1, using a threshold value of 0.143.

... and a formal description of neighborhood map generation (Fig. 3f)...

The neighborhood map of an aligned data set can be interpreted as a histogram. The axes of the histogram correspond to spatial coordinates, and they match the axes of the average computed by averaging all the aligned particles.

The center of a bin represents a position in space relative to a particle located at the center of the histogram, and the value assigned to the bin reflects the number of particles in the data set that are found to lay with respect to some particle in a position comprised in the bin.

For a formal description, let us denote by $\vec{r}_i \in \mathbb{R}^3$ the three-dimensional position of a particle i in the data set, and by M_i the orthogonal matrix that codes the rotation that aligns that particle.

The algorithm locates for each particle i a set of indices N_i^ϵ , that corresponds to the particles \vec{r}_j in the data set whose distance to \vec{r}_i is below a given threshold, and that belong to the same tomogram (to avoid accidental interference between particles from different tomograms).

For each j in N_i^ϵ , the algorithm increases by the one the count of the bin whose center is closest to the point $M_i(\vec{r}_j - \vec{r}_i) \in \mathbb{R}^3$, which is the position of particle j relative to i after aligning particle i .

... and model fitting / model building in the methods section.

We apologize for overlooking this part. We have now included the steps for model fitting / model building in the methods section of the revised manuscript. For convenience we have included below the text:

“The structure of the full-length heterodimer was built into the density using COOT. First, crystal structures of the SNX1^{BAR}-SNX5^{BAR} domains (PDB: 8AIG, present work), SNX1^{PX} domain (PDB: 2I4K), SNX5^{PX} domain in complex with the CI-MPR peptide (PDB: 6N5Y), and the linker regions derived from the alpha fold model were manually fitted into the density map. The amphipathic helix (AH) regions in SNX1 (aa 168-206) and in SNX5 (aa 271-306)

were regularized in COOT. Once the AH regions exhibited proper geometry, they were idealized in PHENIX using the geometry minimization protocol. Finally, the whole SNX1-SNX5 composite structure was refined with the phenix.real_space tool implemented in PHENIX using rigid-body and morphing with secondary-structure restraints.”

Were any specific scripts or internal versions of Dynamo written or modified specifically for this study, and if so, what is the availability of the code?

We used the function `dpktbl.neighborhood.analyze` to compute the neighborhood map. An older version of this script (including functionalities mentioned in this report) is already accessible in the current public version of Dynamo, available at dynamo-em.org.

EMBD and PDB codes are provided, however, an explicit statement of coordinate deposition should still be provided.

This statement has been now included in the ‘Data and code availability’ section.

Decision Letter, first revision:

Message: Our ref: NSMB-A46942A

16th Feb 2023

Dear Dr. Hierro,

Thank you for submitting your revised manuscript "Architecture of the ESCPE-1 membrane coat" (NSMB-A46942A). It has now been seen by the original referees and their comments are below. The reviewers find that the paper has improved in revision, and therefore we'll be happy in principle to publish it in Nature Structural & Molecular Biology, pending minor revisions to satisfy the referees' final requests and to comply with our editorial and formatting guidelines.

We are now performing detailed checks on your paper and will send you a checklist detailing our editorial and formatting requirements in about a week. Please do not upload the final materials and make any revisions until you receive this additional information

from us.

Sincerely,
Kat

Katarzyna Ciazynska
(she/her)
Associate Editor
Nature Structural & Molecular Biology
<https://orcid.org/0000-0002-9899-2428>

Reviewer #1 (Remarks to the Author):

The authors satisfactorily addressed all earlier raised issues.

Reviewer #2 (Remarks to the Author):

The authors have made significant efforts to address queries of the three reviewers. I still find some of the findings a little puzzling, but I believe that the paper is suitable for publication. I have some minor comments but don't think these should prevent publication:

1. The authors state that new data in Fig. 2b suggests SNX5 has non-specific lipid binding. I don't think this can be claimed from the image, as the SNX5 band is extremely weak in all lanes no matter which lipid is used, and it's hard to know if this is just background precipitation rather lipid interaction.
2. The SNX1 FL interaction with PI3P in Fig 2b is quite clear, but this still puzzles me as it seems counter to previous studies, and I don't understand why the PX domain would not show any PI interaction. I can't see anything in the full-length structure that would impart the PI3P specificity except the PX domain itself. But the data seems clear, and I cannot argue with the author's result other than to say it puzzles me.
3. I still do not believe there is convincing evidence for the different stages of SNX1-SNX1 homodimers and SNX1-SNX5 heterodimers proposed in Figure 7. However, I realise this is speculative.

Brett Collins

Reviewer #3 (Remarks to the Author):

The authors did a comprehensive job with this revision, and it is essentially ready to publish. I appreciate their responses to the points raised by me and the other reviewers.

I found one more instance of "stablish", on line 199, which should be "established".

Author Rebuttal, first revision:

Reviewers' Comments:

Reviewer #1:

Remarks to the Author:

The authors satisfactorily addressed all earlier raised issues.

Reviewer #2:

Remarks to the Author:

The authors have made significant efforts to address queries of the three reviewers. I still find some of the findings a little puzzling, but I believe that the paper is suitable for publication. I have some minor comments but don't think these should prevent publication:

1. The authors state that new data in Fig. 2b suggests SNX5 has non-specific lipid binding. I don't think this can be claimed from the image, as the SNX5 band is extremely weak in all lanes no matter which lipid is used, and it's hard to know if this is just background precipitation rather lipid interaction.

Based on the results presented in Fig.2b, our statement was that: "...we found nonspecific minor binding of SNX5 to phosphoinositide-containing liposomes ...". We do agree with the referee that the SNX5 band is extremely weak in all lanes and is hard to claim some lipid interaction. In this sense, we have rephrased the aforementioned sentence as: "... we observed negligible levels of SNX5 association to phosphoinositide-containing liposomes ..."

2. The SNX1 FL interaction with PI3P in Fig 2b is quite clear, but this still puzzles me as it seems counter to previous studies, and I don't understand why the PX domain would not show any PI interaction. I can't see anything in the full-length structure that would impart the PI3P specificity except the PX domain itself. But the data seems clear, and I cannot argue with the author's result other than to say it puzzles me.

The finding that PX domain of SNX1 has very weak binding for PI3P compared to the full-length protein is an interesting result although we agree with the reviewer that the subjacent mechanism remains an intriguing question which will certainly motivate future work.

3. I still do not believe there is convincing evidence for the different stages of SNX1-SNX1 homodimers and SNX1-SNX5 heterodimers proposed in Figure 7. However, I realise this is speculative.

Given the highly speculative nature of the model proposed in Fig.7, in the revised version of the manuscript we have eliminated the transition drawing from heterodimers to homodimers. The updated cartoon only shows the SNX1-SNX5 tubular decoration and summarizes the present work. We also have removed the speculative hypothesis within the discussion.

Reviewer #3:

Remarks to the Author:

The authors did a comprehensive job with this revision, and it is essentially ready to publish. I appreciate their responses to the points raised by me and the other reviewers.

I found one more instance of "stablish", on line 199, which should be "established".

The spelling error has been corrected

Final Decision Letter:

Message 5th May 2023

:

Dear Dr. Hierro,

We are now happy to accept your revised paper "Architecture of the ESCPE-1 membrane coat" for publication as an Article in Nature Structural & Molecular Biology.

As soon as your article is published, you can generate your shareable link by entering the DOI of your article here: http://authors.springernature.com/share. Corresponding authors will also receive an automated email with the shareable link

Your paper will be published online soon after we receive proof corrections and will appear in print in the next available issue. You can find out your date of online publication by contacting the production team shortly after sending your proof corrections. Content is published online weekly on Mondays and Thursdays, and the embargo is set at 16:00 London time (GMT)/11:00 am US Eastern time (EST) on the day of publication. Now is the time to inform your Public Relations or Press Office about your paper, as they might be interested in promoting its publication. This will allow them time to prepare an accurate and satisfactory press release. Include your manuscript tracking number (NSMB-A46942B) and our journal name, which they will need when they contact our press office.

About one week before your paper is published online, we shall be distributing a press release to news organizations worldwide, which may very well include details of your work. We are happy for your institution or funding agency to prepare its own press release, but it must mention the embargo date and Nature Structural & Molecular Biology. If you or your Press Office have any enquiries in the meantime, please contact press@nature.com.

Please note that *Nature Structural & Molecular Biology* is a Transformative Journal (TJ). Authors may publish their research with us through the traditional subscription access route or make their paper immediately open access through payment of an article-processing charge (APC). Authors will not be required to make a final decision about access to their article until it has been accepted. <https://www.springernature.com/gp/open-research/transformative-journals> Find out more about Transformative Journals

Authors may need to take specific actions to achieve <https://www.springernature.com/gp/open-research/funding/policy-compliance-faqs> compliance with funder and institutional open access mandates. If your research is supported by a funder that requires immediate open access (e.g. according to <https://www.springernature.com/gp/open-research/plan-s-compliance> Plan S principles) then you should select the gold OA route, and we will direct you to the compliant route where possible. For authors selecting the subscription publication route, the journal's standard licensing terms will need to be accepted, including <https://www.springernature.com/gp/open-research/policies/journal-policies> self-archiving policies. Those licensing terms will supersede any other terms that the author or any third party may assert apply to any version of the manuscript.

Sincerely,

Katarzyna Ciazynska
(she/her)
Associate Editor
Nature Structural & Molecular Biology
<https://orcid.org/0000-0002-9899-2428>
